# Doubly Robust Conformalized Survival Analysis with Right-Censored Data

**Matteo Sesia** [1] [2]   **Vladimir Svetnik** [3]

## Abstract

We present a conformal inference method for constructing lower prediction bounds for survival times from right-censored data, extending recent approaches designed for more restrictive type-I censoring scenarios. The proposed method imputes unobserved censoring times using a machine learning model, and then analyzes the imputed data using a survival model calibrated via weighted conformal inference. This approach is theoretically supported by an asymptotic double robustness property. Empirical studies on simulated and real data demonstrate that our method leads to relatively informative predictive inferences and is especially robust in challenging settings where the survival model may be inaccurate.

## 1. Introduction

### 1.1. Background and Motivation

Survival analysis focuses on time-to-event data, with applications in many fields including clinical trials, engineering, and marketing. For example, in a clinical trial, researchers may aim to predict how long a cancer patient is likely to survive based on individual characteristics and treatments received. Two central goals are modeling the probability that an event will not occur before a given time and predicting the actual event time. These tasks are complicated by the fact that the data are *censored*—the exact event time may be unknown due to study limitations or participant withdrawal.

While traditional methods, such as the Kaplan-Meier estimator and parametric models like the Cox proportional hazards model (Cox, 1972), are valued for their interpretability, they struggle in high-dimensional settings or when their assumptions are violated. As a result, machine learning (ML)

approaches are gaining popularity (Ishwaran et al., 2008; Katzman et al., 2018), despite the difficulty of obtaining uncertainty estimates and statistical guarantees.

A promising approach to providing rigorous statistical inferences for ML models in survival analysis was recently introduced by Candès et al. (2023) and refined by Gui et al. (2024). Their *conformal inference* (Vovk et al., 2005; Lei & Wasserman, 2014) framework can use *any survival model* to compute a *lower prediction bound* (LPB) for an individual's survival time, supported by rigorous statistical guarantees.

LPBs indicate the time beyond which a patient is expected to survive with at least $(1 - \alpha)$ probability, for some fixed level $\alpha \in (0, 1)$. They can be useful in many applications, including for priorizing treatments under resource constraints. When the data are limited or the model overfits, LPBs tend to be conservatively low, reflecting greater uncertainty. By quantifying uncertainty on an individual basis, LPBs are potentially able to distinguish between patients with confidently high survival expectations and those with greater uncertainty. This can lead to actionable insights for high-stakes applications, mitigating the risks associated with over-reliance on potentially inaccurate ML predictions.

### 1.2. Main Challenges and Contributions

A limitation of the methods proposed by Candès et al. (2023) and Gui et al. (2024) is their focus on *type-I* censoring, a scenario not representative of many practical cases. In type-I censoring, all censoring times need to be observed, including for individuals who experienced an event. Observations are represented as $(\tilde{T}, C)$, where $\tilde{T} = \min(T, C)$, with $T$ as the event time and $C$ as the censoring time. For example, in a clinical trial with a fixed end date, $C$ is the time from enrollment to the trial's end, and patients are either censored $(T > C)$ or experience the event $(T < C)$.

These methods, however, do not extend to situations where the censoring times are unobserved for individuals who experience the event—a more common practical scenario known as *right-censoring*. Under right-censoring, we only observe $(\tilde{T}, \mathbb{I}[T < C])$, where $\mathbb{I}[T < C]$ indicates whether the event occurred before censoring. If $T < C$, the censoring time $C$ is unknown. For example, in a survival study, $T$ is the time until death, and $C$ is the time until withdrawal or the study's end. If a patient dies $(T < C)$, we observe

[1]Department of Data Sciences and Operations, University of Southern California, Los Angeles, CA, USA. [2]Thomas Lord Department of Computer Science, University of Southern California, Los Angeles, CA, USA. [3]Merck&Co., Inc., Rahway, NJ, USA. Correspondence to: Matteo Sesia <sesia@marshall.usc.edu>.

*Proceedings of the 42nd International Conference on Machine Learning*, Vancouver, Canada. PMLR 267, 2025. Copyright 2025 by the author(s).

$\tilde{T} = T$ but not $C$. Conversely, for censored individuals ($T > C$), we observe $\tilde{T} = C$ but not $T$. This incomplete information complicates conformal inference, requiring a novel approach.

We address this challenge by introducing a method for constructing informative LPBs from right-censored data, applicable to any survival model. This extends the approaches of Candès et al. (2023) and Gui et al. (2024) using a two-step process. First, we fit a *censoring model* to estimate the conditional probability of censoring and use it to impute unobserved censoring times, transforming the right-censored dataset into a semi-synthetic type-I censored dataset. Second, we fit a *survival model* to estimate survival probabilities and construct LPBs using the imputed dataset.

This method works well in practice and is *doubly robust* (Bang & Robins, 2005) in theory, ensuring asymptotically valid LPBs if either the censoring model or the survival model is consistently estimated, even if the other is not.

### 1.3. Related Work

This is not first work extending conformal inference to right-censored data. Qi et al. (2024) tackled this challenge by using the Kaplan–Meier estimate to impute the latent *survival* times, operating under the assumption that—despite its lack of covariate adjustment—it can approximate the conditional survival function reasonably well; see Appendix A1.3. However, this assumption is not always easy to justify. Further, unlike ours, their approach is not doubly robust as it relies entirely on having a good approximation of the conditional survival distribution—the very quantity we are trying to infer. As we shall see, our method performs similarly to that of Qi et al. (2024) in easier settings where the survival model already yields approximately valid LPBs, and is able to offer more robust coverage in harder scenarios.

This work contributes to a growing literature on conformal inference beyond exchangeability (Barber et al., 2023), particularly for handling incomplete data. Other works focused on unobserved counterfactuals (Lei & Candès, 2021), missing covariates (Zaffran et al., 2023), weak supervision (Cauchois et al., 2024), and label noise (Feldman et al., 2023; Sesia et al., 2024; Clarkson et al., 2024).

Borrowing from Candès et al. (2023) and Gui et al. (2024), we use weighted conformal inference techniques (Tibshirani et al., 2019) to address the challenge that, under right-censoring, the missing data may not be missing at random.

## 2. Methods

### 2.1. Problem Setup and Assumptions

We consider a sample of $n$ individuals, indexed by $[n] := \{1, \ldots, n\}$, drawn i.i.d. from some unknown population.

For each $i \in [n]$, let $X_i \in \mathcal{X} \subseteq \mathbb{R}^p$ represent a vector of $p$ features, $T_i > 0$ the survival time, and $C_i > 0$ the censoring time. Define the event indicator $E_i = \mathbb{I}(T_i < C_i) \in \{0, 1\}$. The observed time for each individual is $\tilde{T}_i = \min(T_i, C_i)$.

Candès et al. (2023) and Gui et al. (2024) study a type-I censoring scenario, where the available data are $\mathcal{D}^{\text{t1-c}} := \{(X_i, \tilde{T}_i, C_i)\}_{i=1}^n$, which includes both $C_i$ and $\tilde{T}_i$ for all $n$ individuals, along with their features $X_i$. In that setting, they construct an LPB for the survival time $T_{n+1}$ of a new random individual, with features $X_{n+1}$, from the same population. Their methods are reviewed in Appendix A1.

In the right-censoring scenario considered in this paper, the data are $\mathcal{D}^{\text{r-c}} := \{(X_i, \tilde{T}_i, E_i)\}_{i=1}^n$. These data are generally less informative than those found in a type-I censoring setting, because they only include the true censoring times for individuals who did not experience an event. This makes existing methods not directly applicable.

Given a right-censored data set $\mathcal{D}^{\text{r-c}}$, our goal is to predict the survival time $T_{n+1}$ of a new *test* individual with covariates $X_{n+1}$, sampled from the same distribution as the previous data points. Ideally, we would like to construct an LPB for $T_{n+1}$, denoted by $\hat{L}(X_{n+1}; \mathcal{D}^{\text{r-c}})$, that provides $(1 - \alpha)$ *marginal coverage* at the desired level $\alpha \in (0, 1)$:

$$\mathbb{P}\left[T_{n+1} \geq \hat{L}(X_{n+1}; \mathcal{D}^{\text{r-c}})\right] \geq 1 - \alpha. \qquad (1)$$

However, since exact finite-sample guarantees for survival analysis are generally unachievable without strong assumptions, we instead aim to construct LPBs that are *approximately* valid in finite samples. These LPBs are designed to satisfy a relaxed, asymptotic version of (1), which can be understood as a form of *double robustness*: they are asymptotically valid as long as at least one of two key population quantities is estimated consistently, even if the other is not.

We will formally define this double robustness property in Section 3, but for now, we focus on presenting our method, starting with its main underlying assumption.

**Assumption 2.1.** The observed data set $\mathcal{D}^{\text{r-c}} := \{(X_i, \tilde{T}_i, E_i)\}_{i=1}^n$ is generated by applying right-censoring to a latent data set $\{(X_i, T_i, C_i)\}_{i=1}^{n+1}$, consisting of covariates, survival times, and censoring times for $n + 1$ individuals, sampled i.i.d. from some joint distribution $P_{X,T,C} = P_X \cdot P_{T|X} P_{C|X}$, where $T \perp\!\!\!\perp C \mid X$. The features and true survival time of the test data point, $\{X_{n+1}, T_{n+1}\}$, are sampled independently from $P_X \cdot P_{T|X}$.

This setup can be summarized as follows:

$$\begin{aligned}
(X_i, T_i, C_i) &\overset{\text{i.i.d.}}{\sim} P_X \cdot P_{T|X} \cdot P_{C|X}, \qquad \forall i \in [n], \\
E_i &= \mathbb{I}(T_i < C_i), \qquad \tilde{T}_i = \min(T_i, C_i), \\
(X_{n+1}, T_{n+1}) &\overset{\text{ind.}}{\sim} P_X \cdot P_{T|X}.
\end{aligned} \qquad (2)$$

Above, $P_X$, $P_{T|X}$, and $P_{C|X}$ are arbitrary and unknown. The assumption $T \perp\!\!\!\perp C \mid X$, known as "conditional independent censoring", states that $T$ and $C$ are independent given $X$. This standard assumption in survival analysis is also made by Candès et al. (2023) and Gui et al. (2024).

In addition to the *calibration* dataset $\mathcal{D}^{\text{r-c}}$, our method assumes access to an independent *training* dataset $\mathcal{D}^{\text{r-c}}_{\text{train}}$, consisting of analogous observations $(X, \tilde{T}, E)$ from a separate group of individuals. These individuals are typically expected to come from the same population, though this is not strictly required for the theoretical results in this paper.

Following a *split-conformal* inference approach, we use $\mathcal{D}^{\text{r-c}}_{\text{train}}$ to train censoring and survival models, and $\mathcal{D}^{\text{r-c}}$ to transform their outputs into survival LPBs.

## 2.2. DR-COSARC

### 2.2.1. METHOD OVERVIEW

We name our method DR-COSARC, which stands for *Doubly Robust COnformalized Survival Analysis under Right Censoring*. This method uses two ML models: a *survival* model $\hat{\mathcal{M}}^{\text{surv}}$ approximating $P_{T|X}$ and a *censoring* model $\hat{\mathcal{M}}^{\text{cens}}$ approximating $P_{C|X}$. As detailed below, $\hat{\mathcal{M}}^{\text{surv}}$ guides the construction of a *candidate* LPB for $T_{n+1}$ as a function of $X_{n+1}$. This candidate LPB can be adjusted either lower or higher via a scalar *tuning parameter*, calibrated using $\mathcal{D}^{\text{r-c}}$ to (approximately) achieve the desired coverage level $1 - \alpha$. The censoring model $\hat{\mathcal{M}}^{\text{cens}}$ plays two key roles. First, it is used to impute the latent censoring times, simulating a type-I censoring scenario. Second, it helps account for covariate shift in the calibration data, similar to Candès et al. (2023) and Gui et al. (2024).

### 2.2.2. TRAINING SURVIVAL AND CENSORING MODELS

The models $\hat{\mathcal{M}}^{\text{surv}}$ and $\hat{\mathcal{M}}^{\text{cens}}$ can be trained using any survival analysis technique, like the standard Cox proportional hazards model, or more sophisticated ML approaches, including random survival forests (Ishwaran et al., 2008).

Both models can be trained on the same right-censored training set $\mathcal{D}^{\text{r-c}}_{\text{train}}$. For $\hat{\mathcal{M}}^{\text{surv}}$, the event indicator is defined as usual, with a value of 1 indicating that $T < C$. For $\hat{\mathcal{M}}^{\text{cens}}$, the same techniques can be applied after flipping the event indicator: a value of 1 now indicates that the event did not occur before the censoring time ($C < T$).

### 2.2.3. IMPUTING THE MISSING CENSORING TIMES

To simplify the notation, and without much loss of generality, we assume the conditional distribution of $C$ given $X = x$ has a continuous density, denoted by $f_{C|X}(c \mid x)$, with respect to the Lebesgue measure, for any $x \in \mathcal{X}$. Then, we let $F_{C|X}(c \mid x)$ denote the corresponding cumulative dis-

tribution function, defined as $F_{C|X}(c \mid x) = \int_0^c f_{C|X}(c' \mid x)dc'$. Additionally, we assume the pre-trained censoring model $\hat{\mathcal{M}}^{\text{cens}}$ provides empirical estimates $\hat{f}_{C|X}$ of $f_{C|X}$ and $\hat{F}_{C|X}$ of $F_{C|X}$, as it is typically the case in practice.

For example, if $\hat{\mathcal{M}}^{\text{cens}}$ is a Cox proportional hazards model, it estimates the hazard function $h_C(c \mid x) = \hat{f}_{C|X}(x \mid x)/[1 - \hat{F}_{C|X}(c \mid x)]$ using a simple formula, from which $\hat{f}_{C|X}$ can be derived. Alternatively, if $\hat{\mathcal{M}}^{\text{cens}}$ is a *random survival forest*, standard implementations provide a non-parametric estimate of the conditional survival function $1 - \hat{F}_{C|X}(c \mid x)$. This estimate can be smoothly interpolated and differentiated to obtain $\hat{f}_{C|X}$. For further details on computing $\hat{f}_{C|X}$ using standard survival analysis models, see the software repository accompanying this paper.

Next, leveraging our estimate $\hat{f}_{C|X}$ of $f_{C|X}$, we will transform the right-censored dataset $\mathcal{D}^{\text{r-c}}$ into a *synthetic* dataset $\tilde{\mathcal{D}}^{\text{tI-c}}$, designed to mimic the (unobserved) dataset $\mathcal{D}^{\text{tI-c}}$ that would have been collected under a type-I censoring scenario.

For any $i \in [n]$, consider a right-censored random sample $(X_i, \tilde{T}_i, E_i)$ from (2). Since $C_i$ is latent, we replace it with a synthetic "imputed" time, $\hat{C}_i$, computed as follows. If $E_i = 0$, we know that $C_i < T_i$. In this case, the true value of $C_i$ is observed and equal to $\tilde{T}_i$, so we can directly set $\hat{C}_i = \tilde{T}_i$. Otherwise, if $E_i = 1$, we know that $C_i > T_i = \tilde{T}_i$, but the true value of $C_i$ remains unknown. Fortunately, however, we have two key pieces of information that allow us to obtain a sensible "guess" $\hat{C}_i$ of $C_i$: the property $T \perp\!\!\!\perp C \mid X$ in Assumption 2.1 and the estimate $\hat{f}_{C|X}$ of $f_{C|X}$.

Concretely, if $E_i = 1$, we sample $\hat{C}_i$ from the distribution of $C \mid X = X_i, \tilde{T}_i, C > \tilde{T}_i$, independent of everything else. Thanks to the assumption that $T \perp\!\!\!\perp C \mid X$, the probability density of $\hat{C}_i$, as a function of the dummy variable $c \in \mathbb{R}$, can be written as:

$$\hat{C}_i \sim \frac{\hat{f}_{C|X}(c \mid X_i)}{\hat{A}(X_i, \tilde{T}_i)} \mathbb{I}[c > \tilde{T}_i], \tag{3}$$

where

$$\hat{A}(X_i, \tilde{T}_i) = \int_{\tilde{T}_i}^\infty \hat{f}_{C|X}(c \mid X_i)dc. \tag{4}$$

This procedure, outlined in Algorithm 1, provably leads to a synthetic sample $(X_i, \tilde{T}_i, \hat{C}_i)$ that shares the same distribution as the ideal sample $(X_i, \tilde{T}_i, C_i)$ that would be obtained from (2) under type-I censoring, provided that Assumption 2.1 holds and $\hat{f}_{C|X}$ is equal to $f_{C|X}$.

**Proposition 2.2.** *Under Assumption 2.1, let $P_{X,\tilde{T},C}$ denote the distribution of $(X_i, \tilde{T}_i, C_i)$, for any $i \in [n]$, obtained from (2) under type-I censoring. Then, Algorithm 1 applied with $\hat{f}_{C|X} = f_{C|X}$ outputs independent triplets $(X_i, \tilde{T}_i, \hat{C}_i)$ whose distribution is $P_{X,\tilde{T},C}$.*

---

**Algorithm 1** Imputation of Latent Censoring Times

---

**input** Pre-trained censoring model $\hat{\mathcal{M}}^{\text{cens}}$,
   right-censored calibration data $\{(X_i, \tilde{T}_i, E_i)\}_{i=1}^n$.
 1: Using $\hat{\mathcal{M}}^{\text{cens}}$, compute an estimate $\hat{f}_{C|X}(c \mid x)$ of the
    probability density $f_{C|X}(c \mid x)$ of $C \mid X = x$.
 2: **for** $i = 1$ to $n$ **do**
 3:   **if** $E_i = 0$ **then**
 4:     Set $\hat{C}_i = C_i$.
 5:   **else**
 6:     Randomly generate $\hat{C}_i$ based on (3).
 7:   **end if**
 8: **end for**
**output** Imputed censoring times $(\hat{C}_1, \ldots, \hat{C}_n)$.

---

Implementing Algorithm 1 requires computing the normalization constant in (3), defined by the one-dimensional integral in (4). This can be quickly evaluated either analytically or numerically, depending on the censoring model $\hat{\mathcal{M}}^{\text{cens}}$. Then, sampling $\hat{C}_i$ from (3) can be performed numerically using inverse transform sampling, which is also computationally fast. Further implementation details are provided in the software repository accompanying this paper.

After imputing $(\hat{C}_1, \ldots, \hat{C}_n)$ using Algorithm 1, either the method of Candès et al. (2023) or Gui et al. (2024) can be applied by substituting the imputed values $\hat{C}_i$ for the unobserved $C_i$. The specific steps for each approach are detailed in Sections 2.2.4 and 2.2.5, respectively.

We emphasize that, in practice, Algorithm 1 must be applied using an estimate $\hat{f}_{C|X}$ of $f_{C|X}$. Nonetheless, as long as $\hat{f}_{C|X}$ is reasonably accurate, our two-step method is anticipated to yield approximately valid inferences. This parallels the expected behavior of the approaches proposed by Candès et al. (2023) and Gui et al. (2024), which achieve approximately valid survival LPBs using conformal weights derived from an estimated censoring model. We will formalize this intuition later in Section 3 by establishing double robustness results for our method, which are qualitatively analogous to those derived by Candès et al. (2023) and Gui et al. (2024) under the simpler type-I censoring scenario.

### 2.2.4. DR-COSARC with Fixed Cutoffs

We now describe how to implement our method by integrating Algorithm 1 with the approach of Candès et al. (2023), originally designed for data with type-I censoring, which we apply with $C_i$ replaced by $\hat{C}_i$ for all $i \in [n]$.

The approach of Candès et al. (2023), outlined by Algorithm A1 in Appendix A1.1, entails three main steps. First, the focus is shifted from constructing an LPB for $T_{n+1}$ to constructing an LPB for $(T_{n+1} \wedge c_0) := \min\{T_{n+1}, c_0\}$, where $c_0 > 0$ is a pre-defined cutoff constant. Second, $\mathcal{D}^{\text{t1-c}}$

is filtered to include only samples where $C_i \geq c_0$. The filtered dataset is denoted by $\mathcal{I}_{\text{cal}} = \{i \in [n] : C_i \geq c_0\}$. Third, a standard conformal prediction method for non-censored data is applied to calibrate an LPB for $(T_{n+1} \wedge c_0)$.

In the third step above, the output LPB is obtained by calibrating a *candidate bound* denoted as $\hat{f}_a(X_{n+1}; \hat{\mathcal{M}}^{\text{surv}})$, which depends on the survival model $\hat{\mathcal{M}}^{\text{surv}}$ as well as on a tunable parameter $a$. For example, leveraging conformalized quantile regression (CQR) (Romano et al., 2019), one can use $\hat{f}_a(x; \hat{\mathcal{M}}^{\text{surv}}) = \hat{q}_\alpha(x; \hat{\mathcal{M}}^{\text{surv}}) - a$, for $a \in \mathbb{R}$, where $\hat{q}_\alpha(x; \hat{\mathcal{M}}^{\text{surv}})$ is an estimated $\alpha$-quantile of the conditional distribution of $T \mid X$, given by the model $\hat{\mathcal{M}}^{\text{surv}}$.

The main challenge is that the data $(X_i, \tilde{T}_i \wedge c_0)$ for $i \in \mathcal{I}_{\text{cal}}$ are not exchangeable with the test point $(X_{n+1}, T_{n+1} \wedge c_0)$ due to the condition $C > c_0$, which shifts their distribution. However, Candès et al. (2023) noted that this difference is a *covariate shift*, enabling the use of *weighted conformal inference* (Tibshirani et al., 2019). This adjusts for the distribution shift by re-weighting the samples based on an estimate $\hat{c}(x)$ of the conditional censoring probabilities $c(x) := \mathbb{P}[C > c_0 \mid X = x]$, obtained from $\hat{\mathcal{M}}^{\text{cens}}$.

At first sight, it would seem that the method of Gui et al. (2024) can be directly applied to the imputed dataset $\mathcal{D}^{\text{imputed}}$ output by Algorithm 1. Achieving double robustness, however, requires an additional step. Let $\hat{L}'(X_{n+1})$ denote the survival LPB computed by applying the method of Gui et al. (2024) to the imputed dataset $\mathcal{D}^{\text{imputed}}$. Instead of directly outputting $\hat{L}'(X_{n+1})$, our method takes the minimum of $\hat{L}'(X_{n+1})$ and an *uncalibrated* estimate $\hat{q}_\alpha(X_{n+1})$ of the $\alpha$-quantile of $T \mid X = X_{n+1}$, provided by the survival model $\hat{\mathcal{M}}^{\text{surv}}$. While this adjustment is important in theory to ensure the double robustness of our method, it often has a small impact in practice, as we will show empirically, because it is often true that $\hat{L}'(X_{n+1}) \leq \hat{q}_\alpha(X_{n+1})$.

Algorithm 2 summarizes the main ideas of this implementation of our method. See Appendix A2.1 and Algorithm A4 therein for further implementation details.

### 2.2.5. DR-COSARC with Adaptive Cutoffs

A limitation of the method of Candès et al. (2023), inherited by Algorithm 2, is its sensitivity to the choice of $c_0$. If $c_0$ is too small or too large, the resulting LPBs tend to be too low to be informative, with the optimal choice often depending on the data in a complex manner. While Candès et al. (2023) provide a heuristic for tuning $c_0$, it is not always guaranteed that a suitable value of $c_0$ even exists, as noted by Gui et al. (2024) and confirmed by our numerical experiments.

To address this, Gui et al. (2024) extended the approach of Candès et al. (2023) by allowing $c_0$ to vary across individuals based on their features $X$. Their approach can also use quantile regression (Romano et al., 2019) to compute can-

**Algorithm 2** DR-COSARC with Fixed Cutoffs

**input** Pre-trained censoring model $\hat{\mathcal{M}}^{\text{cens}}$,
    pre-trained survival model $\hat{\mathcal{M}}^{\text{surv}}$,
    right-censored data $\mathcal{D}^{\text{r-c}} = \{(X_i, \tilde{T}_i, E_i)\}_{i=1}^n$,
    significance level $\alpha \in (0, 1)$, test covariates $X_{n+1}$,
    fixed threshold $c_0 > 0$.
1: Impute $(\hat{C}_1, \ldots, \hat{C}_n)$ using $\hat{\mathcal{M}}^{\text{cens}}$ (Algorithm 1).
2: Assemble $\mathcal{D}^{\text{imputed}} := \{(X_i, \tilde{T}_i, \hat{C}_i)\}_{i=1}^n$.
3: Compute an estimate $\hat{c}(x)$ of $c(x)$, using $\hat{\mathcal{M}}^{\text{cens}}$.
4: Apply Algorithm A1 (Appendix A1.1) using $\mathcal{D}^{\text{imputed}}$,
    obtaining a preliminary LPB $\hat{L}'(X_{n+1})$.
5: Using $\hat{\mathcal{M}}^{\text{surv}}$, compute an estimate $\hat{q}_\alpha(x)$ of the $\alpha$-
    quantile of the distribution of $T \mid X = x$.
6: Compute $\hat{L}(X_{n+1}) = \min\{\hat{L}'(X_{n+1}), \hat{q}_\alpha(X_{n+1})\}$.
**output** A $1 - \alpha$ survival LPB $\hat{L}(X_{n+1})$.

---

didate LPBs in the form $\hat{f}_a(x; \hat{\mathcal{M}}^{\text{surv}}) = \hat{q}_\alpha(x; \hat{\mathcal{M}}^{\text{surv}}) - a$, for $a \in \mathbb{R}$, where $\hat{q}_\alpha(x; \hat{\mathcal{M}}^{\text{surv}})$ represents an estimate of the true conditional $\alpha$-quantile of the distribution of $T \mid X$, although this is not the only choice. Their method, described in Algorithm A2 in Appendix A1.2, often leads to much more informative LPBs.

Integrating Algorithm 1 with the approach of Gui et al. (2024) leads to the implementation of our method described by Algorithm 3, which often produces more informative LPBs compared to Algorithm 2. See Appendix A2.2 and Algorithm A5 therein for further implementation details.

---

**Algorithm 3** DR-COSARC with Adaptive Cutoffs

**input** Pre-trained censoring model $\hat{\mathcal{M}}^{\text{cens}}$,
    pre-trained survival model $\hat{\mathcal{M}}^{\text{surv}}$,
    right-censored data $\mathcal{D}^{\text{r-c}} = \{(X_i, \tilde{T}_i, E_i)\}_{i=1}^n$,
    significance level $\alpha \in (0, 1)$, test covariates $X_{n+1}$.
1: Impute $(\hat{C}_1, \ldots, \hat{C}_n)$ using Algorithm 1.
2: Assemble $\mathcal{D}^{\text{imputed}} := \{(X_i, \tilde{T}_i, \hat{C}_i)\}_{i=1}^n$.
3: Apply Algorithm A2 (Appendix A1.2) using $\mathcal{D}^{\text{imputed}}$,
    obtaining a preliminary LPB $\hat{L}'(X_{n+1})$.
4: Using $\hat{\mathcal{M}}^{\text{surv}}$, compute an estimate $\hat{q}_\alpha(x)$ of the $\alpha$-
    quantile of the distribution of $T \mid X = x$.
5: Compute $\hat{L}(X_{n+1}) = \min\{\hat{L}'(X_{n+1}), \hat{q}_\alpha(X_{n+1})\}$.
**output** A $1 - \alpha$ survival LPB $\hat{L}(X_{n+1})$.

---

## 3. Double Robustness

Although it operates on right-censored data, DR-COSARC is asymptotically doubly robust as the training and calibration samples grow, akin to the methods of Candès et al. (2023) and Gui et al. (2024) under type-I censoring.

While we focus on the asymptotic regime here, Appendix A5 also provides finite-sample coverage bounds for both the fixed- and adaptive-cutoff implementations of our

method. Those bounds are valuable in theory to prove the asymptotic double robustness, but they are too loose to be practically useful on their own. Nonetheless, despite the difficulty of finite-sample analyses for this problem, DR-COSARC performs quite well empirically, especially in its adaptive-cutoff implementation, as shown in Section 4.

### 3.1. Double Robustness with Fixed Cutoffs

We begin by studying Algorithm 2. For concreteness, we focus on the implementation detailed by Algorithm A4, which uses quantile regression (Romano et al., 2019) to compute candidate LPBs in the form $\hat{f}_a(x; \hat{\mathcal{M}}^{\text{surv}}) = \hat{q}_\alpha(x; \hat{\mathcal{M}}^{\text{surv}}) - a$, for $a \in \mathbb{R}$, where $\hat{q}_\alpha(x; \hat{\mathcal{M}}^{\text{surv}})$ represents an estimate of $q_\alpha(x)$, the true conditional $\alpha$-quantile of the distribution of $T \mid X$, provided by the pre-trained survival model $\hat{\mathcal{M}}^{\text{surv}}$.

Recall that $\mathcal{D}_{\text{train}}$ denotes the independent training dataset, with size $N = |\mathcal{D}_{\text{train}}|$, used to train $\hat{\mathcal{M}}^{\text{surv}}$ and $\hat{\mathcal{M}}^{\text{cens}}$. To establish double robustness, we assume that at least one of these models consistently estimates the relevant quantities.

**Assumption 3.1.** The following two limits hold:

$$\lim_{N \to \infty} \mathbb{E}\left[\left|\frac{1}{\hat{c}(X)} - \frac{1}{c(X)}\right|\right] = 0,$$

$$\lim_{\substack{N \to \infty \\ n \to \infty}} n \cdot \mathbb{E}\left[\int_{\tilde{T}}^\infty \left|\frac{f_{C|X}(c \mid X)}{A(X, \tilde{T})} - \frac{\hat{f}_{C|X}(c \mid X)}{\hat{A}(X, \tilde{T})}\right| dc\right] = 0.$$

Assumption 3.1 requires that, with increasing training data, the censoring model $\hat{\mathcal{M}}^{\text{cens}}$ consistently estimates the two closely related relevant quantities: the conditional censoring probabilities $c(x)$ used in the approach of Candès et al. (2023), and the conditional censoring density $f_{C|X}(c \mid x)$ used in Algorithm 1. Importantly, $\hat{f}_{C|X}(c \mid x)$ must converge to $f_{C|X}(c \mid x)$ at a rate faster than the growth of the calibration sample size $n$, suggesting that a larger portion of the available data should be allocated for training.

**Assumption 3.2.** The following two conditions hold:

  (i) There exists a constant $b > 0$ such that, for any $\epsilon > 0$, $\mathbb{P}[T \geq q_\alpha(x) + \epsilon \mid X = x] \geq 1 - \alpha - b\epsilon$ almost surely with respect to $P_X$.

  (ii) $\lim_{N \to \infty} \mathbb{E}[|\hat{q}_\alpha(X) - q_\alpha(X)|] = 0$.

Assumption 3.2 requires that the survival model $\hat{\mathcal{M}}^{\text{surv}}$ consistently estimates $q_\alpha(x)$, the conditional $\alpha$-quantile of $T \mid X$, while also assuming that the true distribution of $T \mid X$ satisfies a relatively mild smoothness condition.

**Theorem 3.3.** *Under Assumption 2.1, if either Assumption 3.1 or Assumption 3.2 holds, the survival LPB $\hat{L}(X_{n+1})$ produced by Algorithm 2, implemented as detailed Appendix A2.1, has asymptotically valid marginal coverage:*

$$\lim_{N \to \infty, n \to \infty} \mathbb{P}\left[T_{n+1} \geq \hat{L}(X_{n+1})\right] \geq 1 - \alpha.$$

*Further, under Assumption 3.2, it also has approximate conditional coverage, in the sense that, for any $\epsilon > 0$,*

$$\lim_{N \to \infty} \mathbb{P}\left[\mathbb{P}\left[T_{n+1} \geq \hat{L}(X_{n+1}) \mid X_{n+1}\right] > 1 - \alpha - \epsilon\right] = 1.$$

### 3.2. Double Robustness with Adaptive Cutoffs

We study Algorithm 3, focusing for concreteness on the specific implementation detailed in Appendix A2.2.

Now, Assumption 3.1 is replaced by Assumption 3.4, which is similar. Its first limit says that the function $\hat{c}_a$ should be an accurate estimate (up to a scaling constant) of $c_a$, since in that case $(c_a(x)/\hat{c}_a(x))/\mathbb{E}\left[c_a(X)/\hat{c}_a(X)\right] \approx 1$ for all $x$.

**Assumption 3.4.** The following two limits hold:

$$\lim_{N \to \infty} \sup_{a \in [0,1]} \mathbb{E}\left[\left|\frac{c_a(X_{n+1})/\hat{c}_a(X_{n+1})}{\mathbb{E}\left[c_a(X)/\hat{c}_a(X)\right]} - 1\right|\right] = 0,$$

$$\lim_{\substack{N \to \infty \\ n \to \infty}} n \cdot \mathbb{E}\left[\int_{\tilde{T}}^{\infty} \left|\frac{f_{C|X}(c \mid X)}{A(X, \tilde{T})} - \frac{\hat{f}_{C|X}(c \mid X)}{\hat{A}(X, \tilde{T})}\right| dc\right] = 0.$$

Further, some mild technical conditions are needed.

**Assumption 3.5.** The function $\hat{f}_a(x)$ used to compute *candidate bounds* by the approach of Gui et al. (2024) in Algorithm 3 (see Algorithm A5 for details), is continuous in $a$ for $P_X$-almost all $x$. Further, for any $a$, there exists a constant $\hat{\gamma}_a > 0$ such that $1/\hat{c}_a(x) \leq \hat{\gamma}_a$ for $P_X$-almost all $x$.

Then, we can prove Algorithm 3 is also doubly robust.

**Theorem 3.6.** *Under Assumptions 2.1 and 3.5, if either Assumption 3.4 or Assumption 3.2 holds, the LPB $\hat{L}(X_{n+1})$ produced by Algorithm 3, implemented as detailed in Appendix A2.2, has asymptotically valid marginal coverage:*

$$\lim_{N \to \infty, n \to \infty} \mathbb{P}\left[T_{n+1} \geq \hat{L}(X_{n+1})\right] \geq 1 - \alpha.$$

*Further, under Assumption 3.2, it also has approximate conditional coverage, in the sense that, for any $\epsilon > 0$,*

$$\lim_{N \to \infty} \mathbb{P}\left[\mathbb{P}\left[T_{n+1} \geq \hat{L}(X_{n+1}) \mid X_{n+1}\right] > 1 - \alpha - \epsilon\right] = 1.$$

Next, we will verify that the empirical behavior of our method mirrors this appealing theoretical property.

## 4. Numerical Experiments

### 4.1. Setup

**Synthetic data.** We consider three data-generating distributions, summarized in Table A1 (Appendix A3.1), which span a range of interesting settings, partly inspired by related previous works. In each setting, $p = 100$ covariates $X =$

$(X_1, \ldots, X_p)$ are generated independently, while $T$ and $C$ are sampled independently conditional on $X$, from a log-normal distribution—$\log T \mid X \sim \mathcal{N}(\mu(X), \sigma(X))$—or an exponential distribution—$C \mid X \sim \mathrm{Exp}(\lambda(X))$.

The three settings are ordered by decreasing difficulty. The first two simulate challenging scenarios where accurate survival modeling is difficult, emphasizing the importance of conformal inference. In contrast, the third setting facilitates easier survival model fitting, where raw LPBs from the model already provide approximately valid coverage.

**Design and Performance Metrics.** We generate independent training, calibration, and test datasets, each with 1000 samples. Right-censoring is simulated by replacing the true $T$ and $C$ with $\tilde{T} = \min(T, C)$ and $E = \mathbb{I}(T \leq C)$. The censored data are used to fit survival and censoring models, as specified below. Using these models and the calibration data, we compute 90% survival LPBs for the test set. Performance is evaluated by the average proportion of test points where the true survival time exceeds the LPB (targeting 90%) and the average LPB value, with larger values indicating more informative LPBs. To standardize comparisons across distributions, all LPBs are normalized by dividing by the average *oracle* lower bound in each setting. All experiments are repeated 100 times, and results are averaged.

**Models.** We consider four model families for $\hat{\mathcal{M}}^{\mathrm{cens}}$ and $\hat{\mathcal{M}}^{\mathrm{surv}}$, ensuring consistent comparisons across different calibration methods. The models are as follows: (1) *grf*, a generalized random forest (R package `grf`); (2) *survreg*, an accelerated failure time model (AFT) with a lognormal distribution (R package `survival`); (3) *rf*, a generalized random forest (R package `randomForestSRC`); (4) *cox*, the Cox proportional hazards model (R package `survival`).

**Calibration Methods.** We compare six methods. *Oracle* is an idealized "method" that knows $P_{T|X}$ and directly returns the lower 10% quantile, without using the data. *Uncalibrated* outputs the raw 90% LPB produced by $\hat{\mathcal{M}}^{\mathrm{surv}}$ without any calibration. *Naive CQR* applies CQR using $\tilde{T}_{n+1} = T_{n+1} \wedge C_{n+1}$ as the target of inference instead of $T_{n+1}$, typically leading to very small LPBs (Candès et al., 2023). *KM Decensoring* refers to the method of Qi et al. (2024), reviewed in Appendix A1.3. *DR-COSARC (fixed)* and *DR-COSARC (adaptive)* are our methods, implemented as detailed in Appendix A2.1 and A2.2, respectively.

To simplify comparisons, all calibration methods are applied using the same survival model. While the *Uncalibrated* approach could, in principle, benefit from a survival model trained on a larger dataset, doing so would complicate comparisons and does not affect our main conclusions. As shown in Appendix A3, increasing the training sample size does not resolve the reliability issues of the *Uncalibrated*

method in the challenging settings where this approach fails.

**Leveraging Prior Knowledge on $P_{C|X}$.** To examine the effect of incorporating prior knowledge about the censoring distribution, we fit $\hat{\mathcal{M}}^{\text{cens}}$ using only the first $p_1 \leq p$ covariates, assuming $C$ is independent of $(X_{p_1+1}, \ldots, X_p)$ given $(X_1, \ldots, X_{p_1})$. In the data-generating distributions used in these experiments (Table A1), $C \mid X$ is independent of $X_{11}, \ldots, X_{100}$ in all settings. Therefore, for $10 \leq p_1 \leq p = 100$, this prior knowledge helps improve the censoring model by excluding irrelevant predictors and mitigating overfitting. We start with $p_1 = 10$ and later evaluate the impact of larger $p_1$, representing weaker prior knowledge. The case $p_1 = p$ corresponds to no prior knowledge, where all covariates are used to fit the censoring model.

## 4.2. Results

Figure 1 compares the performance of the six methods in settings 1–3, based on the *grf* models.

In the first setting, the *Uncalibrated* method leads to under-coverage. *KM Decensoring* provides no improvement in this case, as the Kaplan-Meier survival curve it uses to impute $T \mid T > C$ fails to reasonably approximate the true distribution of $T \mid T > C, X$. In contrast, DR-COSARC achieves coverage close to the desired 90% level. However, its coverage is still slightly below the target, and the average value of its lower bounds is noticeably lower than that of the oracle, reflecting the high intrinsic difficulty of this setting.

In the second setting, the *Uncalibrated* method continues to be invalid, as does *KM Decensoring*. However, DR-COSARC performs well, achieving 90% coverage and providing relatively high (more informative) LPBs, approaching the oracle's performance. This success is due to its ability to model the censoring distribution accurately.

In the third setting, all methods except *Naive CQR* perform similarly. In this simpler scenario, $\hat{\mathcal{M}}^{\text{surv}}$ is highly accurate, making conformal calibration less necessary.

## 4.3. Impact of the Censoring Model Quality

Figure 2 presents results from experiments similar to those in Figure 1, using only a subset of the available 1000 training samples to fit the censoring model. The aim is to examine how the quality of the censoring model affects the performance of our method, specifically in the challenging setting 1. The results indicate that when the censoring model is trained using fewer samples—leading to lower-quality imputation—our method fails to provide valid coverage, performing comparably to the approach of Qi et al. (2024). However, as the number of training samples increases and the quality of the censoring model improves, the coverage of our method approaches the desired 90% level, consistent

with its double robustness property. When the censoring model is trained with all 1000 available samples, the coverage reaches the target level, and the experimental setup aligns with that of Figure 1.

## 4.4. Additional Experiments

Appendix A3.2 presents the results of additional experiments examining the impact of varying the number of training samples for the censoring model, similar to Figure 2 but under the relatively easier settings 2 and 3. In those settings, our method consistently achieves valid coverage, and the performance of its predictions appears largely unaffected by the number of training samples for the censoring model.

Appendix A3.3 summarizes the results of experiments similar to those in Figure 2, but with both the survival and censoring models fitted using a varying number of training samples. These results show that our method tends to achieve valid coverage as the sample size increases, even when other approaches either continue to underperform in terms of coverage or produce overly conservative inferences.

Appendix A3.4 investigates the effect of varying numbers of covariates used to fit the censoring model. The results show that our method performs better when the number of covariates used in the censoring model is not too large, highlighting the advantage of leveraging accurate prior information to prevent overfitting.

Appendix A3.5 examines the effect of the calibration sample size. The results indicate that the average performance of all methods is generally not heavily influenced by the number of calibration samples, although larger calibration sizes do tend to reduce variability.

Appendix A3.6 shows empirically that the double robustness adjustment in Algorithms 2 and 3 often has a small effect in practice.

Appendix A3.7 studies the algorithmic stability of our method with respect to the randomness in the imputation of latent censoring times. The results show the adaptive version of our method is more stable than the fixed-cutoff variant, with stability improving as the calibration set grows.

Appendix A3.8 examines the effect of using different survival and censoring models. These results show our method performs robustly across different survival and censoring models, except for the Cox model, which struggles with complex censoring patterns in the most challenging cases.

Appendix A3.9 presents the results of additional experiments conducted under different synthetic data settings, borrowed from Candès et al. (2023) and Gui et al. (2024). These experiments yield similar findings. A complete list of all 10 settings considered is provided in Table A2.

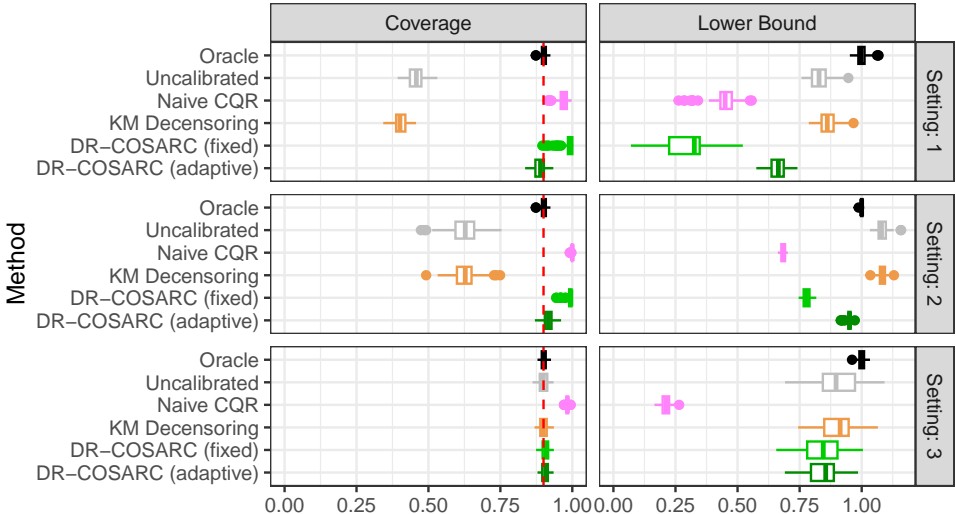

Figure 1. Performance on synthetic data under three different settings of our method for constructing lower confidence bounds on the true survival time of a new individual based on right-censored data, compared to existing benchmark approaches. Performance is measured by empirical coverage, aiming for 90% nominal coverage (dashed red line), and the average value of the lower bound (higher is better, provided the coverage is valid). The number of training samples available to fit the survival and censoring models is 1000. The three settings correspond to situations in which fitting an accurate survival model is increasingly easy, with rigorous conformal calibration being most essential in setting 1 and less crucial in setting 3.

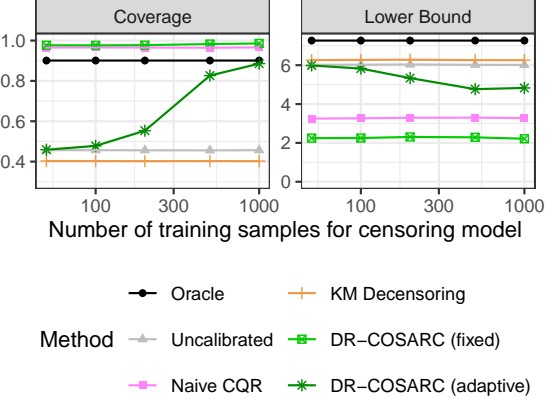

Figure 2. Performance on synthetic data of our method as a function of the number of training samples used to fit the censoring model, compared to other benchmark approaches. The number of training samples available to fit the survival model is fixed equal to 1000. These experiments are conducted under setting 1, where fitting an accurate survival model is most difficult. The results demonstrate the double robustness property of our method, which requires only one of the survival or censoring models to be accurate in order to achieve valid coverage. Other details are as in Figure 1.

## 5. Application to Real Data

We apply our method to seven publicly available datasets: VALCT, PBC, GBSG, METABRIC, COLON, HEART, and RETINOPATHY. These datasets cover a range of study designs and sizes; Table A3 in Appendix A4 provides details on the number of observations, covariates, and data sources.

We apply standard preprocessing to each dataset to handle outliers, missing values, and ensure compatibility with all learning algorithms. Zero survival times are replaced with half the smallest non-zero time in the dataset, missing values are imputed using the median for numeric variables and the mode for categorical variables, and rare factor levels are merged into an "other" category or removed for binary factors. Features with high pairwise correlations are iteratively filtered, and linearly redundant variables are removed. See Appendix A4 for additional details about preprocessing.

We compare our method against the same three benchmark approaches considered in Section 4: *Uncalibrated*, *Naive CQR*, and *KM Decensoring*. Because the ground truth data distribution is unknown for these data, the *Oracle* method cannot be included. Additionally, as the experiments in Section 4 demonstrate that the adaptive-cutoff implementation of our method consistently outperforms the fixed-cutoff implementation, we focus solely on the adaptive version here, referring to it simply as *DR-COSARC* for clarity.

All methods use the same four types of model as in Section 4 to estimate the survival distribution (*grf*, *survreg*, *rf*, and *cox*), with the censoring distribution always estimated using *grf*. The datasets are split into 60% for training, 20% for calibration, and 20% for testing, and each experiment is repeated 100 times using independent random splits.

We evaluate the performance of the survival LPBs produced by each method on the test set in terms of estimated average coverage (targeting the nominal $1 - \alpha$ level) and average LPB value (higher is better). Since the test data are censored, the true survival times for censored individuals are unobserved, making exact coverage evaluation infeasible. Following the approach of Gui et al. (2024), we estimate empirical lower and upper bounds for the average coverage: $\hat{\beta}_{\text{low}} = \hat{\mathbb{P}}[\hat{L}(X_{n+1}) \leq \tilde{T}_{n+1}]$ and $\hat{\beta}_{\text{upp}} = 1 - \hat{\mathbb{P}}[\hat{L}(X_{n+1}) > \tilde{T}_{n+1}, T_{n+1} \leq C_{n+1}]$. These bounds satisfy $\hat{\beta}_{\text{low}} \leq \hat{\mathbb{P}}[\hat{L}(X_{n+1}) \leq T_{n+1}] \leq \hat{\beta}_{\text{upp}}$. To simplify comparisons, we also report a point estimate of the coverage, defined as the midpoint $\hat{\beta}_{\text{mid}} = (\hat{\beta}_{\text{low}} + \hat{\beta}_{\text{upp}})/2$.

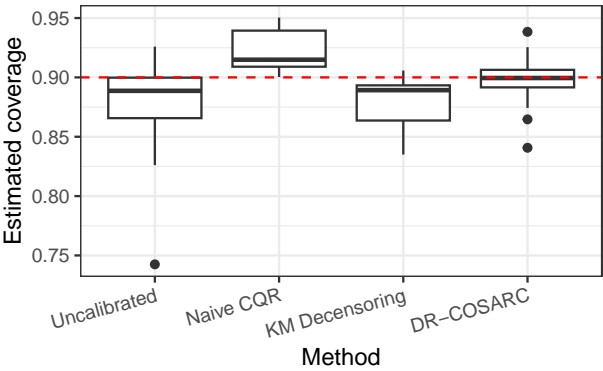

*Figure 3.* Distribution of average (estimated) coverage of survival LPBs computed by different methods, across seven real datasets and four survival models. The nominal coverage level is 90%.

Figure 3 summarizes the distribution of average estimated coverage across the seven datasets and four models, at level $\alpha = 0.1$, with Tables A4–A7 in Appendix A4 reporting the detailed results obtained in each setting. Figure A18 in Appendix A4 summarizes similar results obtained using different values of $\alpha$. Overall, the results indicate that *Uncalibrated* and *KM Decensoring* tend to achieve slightly lower-than-expected coverage, while *Naive CQR* is overly conservative. In contrast, *DR-COSARC* consistently achieves average coverage closer to the desired level.

These findings align with the results on synthetic data presented in Section 4. The relatively modest undercoverage observed with *Uncalibrated* and *KM Decensoring* in Figure 3 reflects the comparatively simpler nature of these datasets, where the survival model seems reasonably well calibrated even without conformal inference, in contrast to the more challenging synthetic scenarios discussed earlier. Of course, in practice one would typically not know whether the fitted survival model is sufficiently accurate, and our conformal inference method is precisely designed to offer an additional layer of protection in those cases.

## 6. Discussion

This paper introduces a novel conformal inference method for constructing lower prediction bounds (LPBs) for survival times from right-censored data, extending recent methods designed for type-I censoring. The proposed approach is asymptotically doubly robust in theory and demonstrates strong empirical performance, producing LPBs that are both informative and robust compared to alternative methods.

Our numerical experiments revealed two key insights. First, the adaptive implementation of our method, inspired by Gui et al. (2024), significantly outperforms the fixed-cutoff version (Candès et al., 2023), and we recommend its use in practice. Second, real data experiments showed that uncalibrated survival models often produce reasonably well-calibrated raw LPBs, though they may fail in more complex scenarios. Our method performs relatively well in these challenging cases, where conformal inference is most critical.

A limitation of our method is its focus on lower prediction bounds, similar to Candès et al. (2023) and Gui et al. (2024). However, Holmes & Marandon (2024) very recently proposed a method for constructing also corresponding upper bounds, suggesting opportunities for combining these approaches. Another promising direction for future work is to extend our method to handle possible data errors, such as inaccuracies in observed times or mislabeled events, building on ideas from Sesia et al. (2024).

Finally, future work could explore strategies to reduce the algorithmic variability of our method, which arises from both random data splitting and stochastic imputation of latent censoring times. Potential directions include the use of e-values (Vovk & Wang, 2021; Bashari et al., 2023) or adopting a full-conformal approach (Vovk et al., 2005).

## Software Availability

A software implementation of the methods described in this paper is available online at `https://github.com/msesia/conformal_survival`.

## Acknowledgements

The authors thank the anonymous referees for their helpful comments on an earlier version of this manuscript.

## Impact Statement

This paper presents work whose goal is to advance the field of Machine Learning. There are many potential societal consequences of our work, none which we feel must be specifically highlighted here.

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

## A1. Review of Existing Conformal Inference Methods

### A1.1. Conformalized Survival Analysis for Type-I Censored Data (fixed)

Algorithm A1 outlines the conformalized survival analysis method proposed by Candès et al. (2023), designed for data subject to type-I censoring. The method requires several key inputs in addition to the censored calibration data: (1) a pre-trained survival model that approximates the conditional distribution of $T \mid X$; (2) a sequence of functions used to compute candidate survival lower bounds; (3) a pre-trained censoring model that approximates the conditional distribution of $C \mid X$ and is utilized to compute the necessary weights to account for covariate shift (Tibshirani et al., 2019); and (4) a fixed threshold $c_0 > 0$ for the censoring times. These inputs are typically trained on a separate dataset, independent of the calibration data (Candès et al., 2023).

---

**Algorithm A1** Conformalized Survival Analysis with Fixed Cutoff (Candès et al., 2023)

**Require:** Pre-trained survival model $\hat{\mathcal{M}}^{\text{surv}}$, sequence of functions $\{\hat{f}_a(x; \hat{\mathcal{M}}^{\text{surv}})\}_{a \in [0,1]}$ to compute candidate survival lower bounds, pre-trained censoring model $\hat{\mathcal{M}}^{\text{cens}}$, fixed cutoff $c_0 > 0$, calibration data with type-I censoring $\{(X_i, \tilde{T}_i, C_i)\}_{i=1}^n$, significance level $\alpha \in (0, 1)$, test covariates $X_{n+1}$.

1: Define the filtered calibration subset $\mathcal{I}'_{\text{cal}} = \{i \in \{1, \ldots, n\} : C_i \geq c_0\}$.
2: **for** each $i \in \mathcal{I}'_{\text{cal}}$ **do**
3:     Compute the conformity score:
$$V_i = \inf\{a \in \mathcal{A} : \hat{f}_a(X_i) \leq \tilde{T}_i \wedge c_0\}.$$
4: **end for**
5: Define the function $\hat{w}(x)$, estimating $1/c(x) := 1/\mathbb{P}(C \geq c_0 \mid X = x)$ using $\hat{\mathcal{M}}^{\text{cens}}$, $\forall x$.
6: For each $i \in \mathcal{I}'_{\text{cal}}$, compute the weight $W_i = \hat{w}(X_i) \in [0, \infty)$.
7: Compute the weights for $X_{n+1}$:
$$\hat{p}_i(X_{n+1}) = \frac{W_i}{\sum_{j \in \mathcal{I}'_{\text{cal}}} W_j + \hat{w}(X_{n+1})}, \quad \hat{p}_\infty(X_{n+1}) = \frac{\hat{w}(X_{n+1})}{\sum_{j \in \mathcal{I}'_{\text{cal}}} W_j + \hat{w}(X_{n+1})}.$$
8: Compute:
$$\eta(X_{n+1}) = \text{Quantile}\left(1 - \alpha; \sum_{i \in \mathcal{I}'_{\text{cal}}} \hat{p}_i(X_{n+1})\delta_{V_i} + \hat{p}_\infty(X_{n+1})\delta_\infty\right).$$
9: Output the calibrated $1 - \alpha$ lower prediction bound:
$$\hat{L}(X_{n+1}) = \hat{f}_{\eta(X_{n+1})}(X_{n+1}) \wedge c_0.$$

---

**Implementation Details.** While Algorithm A1 is quite flexible, allowing the candidate survival lower bounds to be computed using any set of functions $\hat{f}_a(x; \hat{\mathcal{M}}^{\text{surv}})$ indexed by a real-valued calibration parameter $a \in \mathcal{A} \subseteq \mathbb{R}$, in practice, we adopt one of the simpler implementations proposed by Candès et al. (2023). This approach is inspired by the conformalized quantile regression method of Romano et al. (2019) and defines $\hat{f}_a(x; \hat{\mathcal{M}}^{\text{surv}}) = \hat{q}_\alpha(x; \hat{\mathcal{M}}^{\text{surv}}) - a$, for $\mathcal{A} = \mathbb{R}$, where $\hat{q}_\alpha(x; \hat{\mathcal{M}}^{\text{surv}})$ represents the estimated $\alpha$-quantile of the conditional distribution of $T \mid X$, given by the survival model $\hat{\mathcal{M}}^{\text{surv}}$. In this case, the conformity scores are simply given by $V_i = \hat{q}_\alpha(X_i) - (\tilde{T}_i \wedge c_0)$, and the output prediction lower bound is $\hat{L}(X_{n+1}) = (\hat{q}_\alpha(X_{n+1}) - \eta(X_{n+1})) \wedge c_0$.

**Data-Driven Tuning of $c_0$.** Candès et al. (2023) proposed an algorithm for tuning the cutoff parameter $c_0$ adaptively, using the training data set. In this paper, we apply their method using a simpler approach for tuning $c_0$, which we always set equal to the median of the observed censoring times. We found this approach to be relatively more stable in practice.

### A1.2. Conformalized Survival Analysis for Type-I Censored Data (adaptive)

Algorithm A2 presents the conformalized survival analysis method proposed by Gui et al. (2024), a follow-up to the work of Candès et al. (2023). The goal of this more recent approach is to enhance the adaptability of conformal survival analysis for data subject to type-I censoring by incorporating a more flexible, covariate-dependent threshold for the censoring times. As demonstrated in Gui et al. (2024) and corroborated by our numerical experiments, the adaptive strategy of Algorithm A2 often leads to more informative lower prediction bounds compared to the original method proposed by Candès et al. (2023).

Similar to Algorithm A1, this method requires the following inputs in addition to the censored calibration data: (1) a pre-trained survival model that estimates the conditional distribution of $T \mid X$; (2) a sequence of functions used to compute candidate survival lower bounds; and (3) a pre-trained censoring model that approximates the conditional distribution of $C \mid X$, which is used to compute weights for adjusting to covariate shift (Tibshirani et al., 2019). These models are typically trained on a separate dataset, independent of the calibration data (Candès et al., 2023). Unlike Algorithm A1, however, Algorithm A2 does not require the specification of a fixed censoring threshold $c_0 > 0$, and this is its main advantage.

---

**Algorithm A2** Conformalized Survival Analysis with Adaptive Cutoffs (Gui et al., 2024)

---

**Require:** Pre-trained survival model $\hat{\mathcal{M}}^{\text{surv}}$, sequence of functions $\{\hat{f}_a(x; \hat{\mathcal{M}}^{\text{surv}})\}_{a \in [0,1]}$ to compute candidate survival lower bounds, pre-trained censoring model $\hat{\mathcal{M}}^{\text{cens}}$, calibration data with type-I censoring $\{(X_i, \tilde{T}_i, C_i)\}_{i=1}^{n}$, significance level $\alpha \in (0, 1)$, test covariates $X_{n+1}$.

1: Define the function $\hat{w}_a(x)$, estimating $1/c_a(x) := 1/\mathbb{P}(C \geq \hat{f}_a(x) \mid X = x)$ using $\hat{\mathcal{M}}^{\text{cens}}$, $\forall a, x$.
2: Determine $\mathcal{A}$ using the computational shortcut from Gui et al. (2024), as detailed in Equation (A6).
3: **for** each $a \in \mathcal{A}$ **do**
4:   Compute the estimated miscoverage rate:

$$\hat{\alpha}(a) = \frac{\sum_{i=1}^{n} \hat{w}_a(X_i)\mathbb{I}\{\tilde{T}_i < \hat{f}_a(X_i) \leq C_i\}}{\sum_{i=1}^{n} \hat{w}_a(X_i)\mathbb{I}\{\hat{f}_a(X_i) \leq C_i\}}.$$

5: **end for**
6: Compute the threshold:

$$\hat{a} = \sup\{a \in \mathcal{A} : \sup_{a' \leq a, a' \in \mathcal{A}} \hat{\alpha}(a') \leq \alpha\}. \tag{A5}$$

7: Output the calibrated $1 - \alpha$ lower prediction bound:

$$\hat{L}(X_{n+1}) = \hat{f}_{\hat{a}}(X_{n+1}).$$

---

**Computational shortcut.** In general, the threshold $\hat{a}$ in (A5) can be computed efficiently using the following shortcut, originally described in Gui et al. (2024), which can also be utilized to implement our Algorithm A5 efficiently. Note that $\sup_{a' \leq a} \hat{\alpha}(a')$ is a non-decreasing piecewise constant function in $a$, with no more than $2n$ knots—values of $a$ at which the indicators $\mathbb{I}\{T_i < \hat{f}_a(X_i) \leq C_i\}$ or $\mathbb{I}\{\hat{f}_a(X_i) \leq C_i\}$ change signs. Denote $\bar{a}_i = \sup_{a \in [0,1]} \mathbb{I}\{\hat{f}_a(X_i) \leq \tilde{T}_i\}$ and $\tilde{a}_i = \sup_{a \in [0,1]} \mathbb{I}\{\hat{f}_a(X_i) \leq C_i\}$, and let $\mathcal{A}_1 = \{\bar{a}_i : i = 1, \ldots, n\}$ and $\mathcal{A}_2 = \{\tilde{a}_i : i = 1, \ldots, n\}$. Then, by definition, the breakpoints of the piecewise constant map $a \mapsto \hat{\alpha}(a)$ must all lie in $\mathcal{A}_1 \cup \mathcal{A}_2$. Therefore, to compute $\hat{a}$, we only need to search through the finite grids

$$\mathcal{A} = \mathcal{A}_1 \cup \mathcal{A}_2 \cup \{0\}. \tag{A6}$$

**Implementation Details.** Similar to Algorithm A1, Algorithm A2 is quite flexible, allowing the candidate survival lower bounds to be computed using any set of functions $\hat{f}_a(x; \hat{\mathcal{M}}^{\text{surv}})$ indexed by a continous calibration parameter $a \in [0, 1]$. In this paper, we adopt one of the simpler implementations proposed by Gui et al. (2024). This approach defines $\hat{f}_a(x; \hat{\mathcal{M}}^{\text{surv}}) = \hat{q}_a(x; \hat{\mathcal{M}}^{\text{surv}})$, for $\mathcal{A} = [0, 1]$, where $\hat{q}_a(x; \hat{\mathcal{M}}^{\text{surv}})$ represents the estimated $a$-quantile of the conditional distribution of $T \mid X$, given by the survival model $\hat{\mathcal{M}}^{\text{surv}}$.

### A1.3. Conformalized Survival Analysis via KM Decensoring

Algorithm A3 presents the conformalized survival analysis method proposed by Qi et al. (2024), which, similar to our paper, focuses on the analysis of data subject to right censoring.

This approach uses the Kaplan–Meier estimate to impute the latent *survival* times, and is thus very different from our method, which imputes the latent *censoring* times.

---

**Algorithm A3** Conformalized Survival Analysis via KM Decensoring (Qi et al., 2024)

---

**Require:** Level $\alpha$, calibration data $\mathcal{D}_{\text{cal}} = \{(X_i, \tilde{T}_i, C_i)\}_{i=1}^n$, test features $x$, functions $V(x, y; \mathcal{D})$ (conformity score), $\hat{w}(x; \mathcal{D})$ (weight function), $\mathcal{C}(\mathcal{D})$ (threshold selector), right-censored calibration data $\{(X_i, \tilde{T}_i, E_i)\}_{i=1}^n$.

1: **Phase 1: Imputation of Latent Event Times via KM Sampling**
2: Compute the Kaplan-Meier (KM) survival function:

$$S_{\text{KM}}(t) = \prod_{i:\tilde{t}_i \leq t} \left(1 - \frac{d_i}{n_i}\right),$$

based on $\{(\tilde{T}_i, E_i)\}_{i=1}^n$.
3: **for** $i = 1$ to $n$ **do**
4:    **if** $E_i = 0$ **then**
5:       Define the conditional KM survival function:

$$S_{\text{KM}}(t \mid t > C_i) = \min\left\{\frac{S_{\text{KM}}(t)}{S_{\text{KM}}(C_i)}, 1\right\}.$$

6:       Set $T_i'$ equal to a random sample generated with density:

$$S_{\text{KM}}(t \mid t > C_i)\mathbb{I}\{t > C_i\}.$$

7:    **else**
8:       Set $T_i' = \tilde{T}_i$.
9:    **end if**
10: **end for**
11: **Phase 2: Conformal Prediction with De-Censored Data**
12: **for** $i = 1$ to $n$ **do**
13:    Compute the conformity score $V_i = V(X_i, T_i')$.
14: **end for**
15: Compute:

$$\eta(x) = \text{Quantile}\left(1 - \alpha; \sum_{i=1}^n \frac{1}{n+1}\delta_{V_i} + \frac{1}{n+1}\delta_\infty\right).$$

16: Compute the lower prediction bound:

$$\hat{L}(x; \mathcal{D}_{\text{cal}}) = \inf\{y : V(x, y) \leq \eta(x)\}.$$

17: Output the calibrated lower prediction bound:
$$\hat{L}(X_{n+1}; \mathcal{D}_{\text{cal}}).$$

---

## A2. Additional Methodological Details

### A2.1. DR-COSARC with Fixed Cutoffs

Algorithm A4 provides further details on the implementation of our method sketched by Algorithm 2, which integrates Algorithm 1 with Algorithm A1 in Appendix A1.1, the approach of Candès et al. (2023) for conformalized survival analysis under type-I censoring.

---

**Algorithm A4** DR-COSARC with Fixed Cutoffs (detailed implementation)

---

**Require:** Pre-trained survival model $\hat{\mathcal{M}}^{\text{surv}}$, sequence of functions $\{\hat{f}_a(x; \hat{\mathcal{M}}^{\text{surv}})\}_{a \in \mathcal{A}}$ to compute candidate survival lower bounds, pre-trained censoring model $\hat{\mathcal{M}}^{\text{cens}}$, right-censored calibration data $\{(X_i, \tilde{T}_i, E_i)\}_{i=1}^n$, fixed threshold $c_0 > 0$, significance level $\alpha \in (0, 1)$, test covariates $X_{n+1}$.

1: **Phase 1: Imputation of Latent Censoring Times**
2: Generate $(C'_1, \ldots, C'_n)$ using Algorithm 1, and define $\tilde{\mathcal{D}}_{\text{cal}} = \{(X_i, \tilde{T}_i, C'_i)\}_{i=1}^n$.
3: **Phase 2: Conformal Calibration**
4: Define the filtered calibration subset $\mathcal{I}_{\text{cal}} = \{i \in \{1, \ldots, n\} : C'_i \geq c_0\}$.
5: **for** each $i \in \mathcal{I}_{\text{cal}}$ **do**
6:     Compute the conformity score $V_i = \inf\{a \in \mathcal{A} : \hat{f}_a(X_i) \leq \tilde{T}_i \wedge c_0\}$.
7:     Compute the weight $W_i = \hat{w}(X_i)$, where $\hat{w}(x)$ estimates $1/c(x) := 1/\mathbb{P}(C \geq c_0 \mid X = x)$ using $\hat{\mathcal{M}}^{\text{cens}}$.
8: **end for**
9: For each $i \in \mathcal{I}_{\text{cal}}$, compute the weights:

$$\hat{p}_i(X_{n+1}) = \frac{W_i}{\sum_{j \in \mathcal{I}_{\text{cal}}} W_j + \hat{w}(X_{n+1})}, \quad \hat{p}_\infty(X_{n+1}) = \frac{\hat{w}(X_{n+1})}{\sum_{j \in \mathcal{I}_{\text{cal}}} W_j + \hat{w}(X_{n+1})}.$$

10: Compute:

$$\eta(X_{n+1}) = \text{Quantile}\left(1 - \alpha; \sum_{i \in \mathcal{I}_{\text{cal}}} \hat{p}_i(X_{n+1})\delta_{V_i} + \hat{p}_\infty(X_{n+1})\delta_\infty\right).$$

11: Compute $\hat{L}'(X_{n+1}; \tilde{\mathcal{D}}_{\text{cal}}) = \hat{f}_{\eta(X_{n+1})}(X_{n+1}) \wedge c_0$.
12: **Phase 3: Adjustment for Double Robustness**
13: Extract the function $\hat{q}_\alpha(x)$ from $\hat{\mathcal{M}}^{\text{surv}}$, an uncalibrated estimate of the $\alpha$-quantile of $T \mid X = x$.
14: Adjust the lower prediction bound:

$$\hat{L}(X_{n+1}) = \min\{\hat{L}'(X_{n+1}; \tilde{\mathcal{D}}_{\text{cal}}), \hat{q}_\alpha(X_{n+1})\}.$$

15: Output: A calibrated $1 - \alpha$ lower prediction bound $\hat{L}(X_{n+1})$.

---

**Implementation details for Algorithm A4.** While Algorithm A4 is quite flexible, allowing the candidate survival lower bounds to be computed using any set of functions $\hat{f}_a(x; \hat{\mathcal{M}}^{\text{surv}})$ indexed by a real-valued calibration parameter $a \in \mathcal{A} \subseteq \mathbb{R}$, in practice, we adopt one of the simpler implementations proposed by Candès et al. (2023). This approach is inspired by the conformalized quantile regression method of Romano et al. (2019) and defines $\hat{f}_a(x; \hat{\mathcal{M}}^{\text{surv}}) = \hat{q}_\alpha(x; \hat{\mathcal{M}}^{\text{surv}}) - a$, for $\mathcal{A} = \mathbb{R}$, where $\hat{q}_\alpha(x; \hat{\mathcal{M}}^{\text{surv}})$ represents the estimated $\alpha$-quantile of the conditional distribution of $T \mid X$, given by the survival model $\hat{\mathcal{M}}^{\text{surv}}$. In this case, the conformity scores are simply given by $V_i = \hat{q}_\alpha(X_i) - (\tilde{T}_i \wedge c_0)$, and the output prediction lower bound is $\hat{L}(X_{n+1}) = (\hat{q}_\alpha(X_{n+1}) - \eta(X_{n+1})) \wedge c_0$.

## A2.2. DR-COSARC with Adaptive Cutoffs

Algorithm A5 provides further details on the implementation of our method sketched by Algorithm 3, which integrates Algorithm 1 with Algorithm A2 in Appendix A1.2, the approach of Gui et al. (2024) for conformalized survival analysis under type-I censoring.

---

**Algorithm A5** DR-COSARC with Adaptive Cutoffs (detailed implementation)

---

**input** Pre-trained survival model $\hat{\mathcal{M}}^{\text{surv}}$, sequence of functions $\{\hat{f}_a(x; \hat{\mathcal{M}}^{\text{surv}})\}_{a \in [0,1]}$ to compute candidate survival lower bounds, pre-trained censoring model $\hat{\mathcal{M}}^{\text{cens}}$, right-censored calibration data $\{(X_i, \tilde{T}_i, E_i)\}_{i=1}^n$, significance level $\alpha \in (0,1)$, test covariates $X_{n+1}$.

1: **Phase 1: Imputation of Latent Censoring Times**
2: Generate $(C_1', \ldots, C_n')$ using Algorithm 1, and define $\tilde{\mathcal{D}}_{\text{cal}} = \{(X_i, \tilde{T}_i, C_i')\}_{i=1}^n$.
3: **Phase 2: Conformal Calibration**
4: Define the function $\hat{w}_a(x)$, estimating $1/c_a(x) := 1/\mathbb{P}(C \geq \hat{f}_a(x) \mid X = x)$ using $\hat{\mathcal{M}}^{\text{cens}}$, $\forall a, x$.
5: Determine $\mathcal{A}$ using the computational shortcut from Gui et al. (2024), as detailed in Appendix A1.
6: **for** each $a \in \mathcal{A}$ **do**
7:     Compute the estimated miscoverage rate:

$$\hat{\alpha}(a) = \frac{\sum_{i=1}^n \hat{w}_a(X_i)\mathbb{I}\{\tilde{T}_i < \hat{f}_a(X_i) \leq C_i'\}}{\sum_{i=1}^n \hat{w}_a(X_i)\mathbb{I}\{\hat{f}_a(X_i) \leq C_i'\}}.$$

8: **end for**
9: Compute the threshold:

$$\hat{a} = \sup\{a \in \mathcal{A} : \sup_{a' \leq a, a' \in \mathcal{A}} \hat{\alpha}(a') \leq \alpha\}.$$

10: **Phase 3: Adjustment for Double Robustness**
11: Extract the function $\hat{q}_\alpha(x)$ from $\hat{\mathcal{M}}^{\text{surv}}$, an uncalibrated estimate of the $\alpha$-quantile of $T \mid X = x$.
12: Adjust the lower prediction bound:

$$\hat{L}(X_{n+1}) = \min\{\hat{f}_{\hat{a}}(X_{n+1}), \hat{q}_\alpha(X_{n+1})\}.$$

13: Output the calibrated $1 - \alpha$ lower prediction bound $\hat{L}(X_{n+1})$.

---

**Implementation details for Algorithm A5.** In this paper, we implement Algorithm A5 using candidate LPBs in the form $\hat{f}_a(x; \hat{\mathcal{M}}^{\text{surv}}) = \hat{q}_\alpha(x; \hat{\mathcal{M}}^{\text{surv}}) - a$, for $a \in \mathbb{R}$.

The threshold $\hat{a}$ in (4) can be computed efficiently as follows. We note that $\sup_{a' \leq a} \hat{\alpha}(a')$ is a non-decreasing piecewise constant function in $a$, with no more than $2n$ knots—values of $a$ at which the indicators $\mathbb{I}\{T_i < \hat{f}_a(X_i) \leq C_i\}$ or $\mathbb{I}\{\hat{f}_a(X_i) \leq C_i\}$ change signs.

Denote $\bar{a}_i = \sup_{a \in [0,1]} \mathbb{I}\{\hat{f}_a(X_i) \leq \tilde{T}_i\}$ and $\tilde{a}_i = \sup_{a \in [0,1]} \mathbb{I}\{\hat{f}_a(X_i) \leq C_i\}$, and let $A_1 = \{\bar{a}_i : i = 1, \ldots, n\}$ and $A_2 = \{\tilde{a}_i : i = 1, \ldots, n\}$. Then, by definition, the breakpoints of the piecewise constant map $a \mapsto \hat{\alpha}(a)$ must all lie in $A_1 \cup A_2$. In the implementation, in order to obtain $\hat{a}$, we only need to search through the finite grids

$$A = A_1 \cup A_2 \cup \{0\}.$$

## A3. Additional Numerical Results

### A3.1. Additional Details on the Experiments of Section 4

*Table A1.* Summary of three synthetic data generation settings considered in Section 4, listed in decreasing order of intrinsic difficulty.

| Setting | $p$ | Ref. | Covariate, Survival, and Censoring Distributions |
|---|---|---|---|
| 1 | 100 | | $X$ : Unif$([0, 1]^p)$ 
 $T \mid X$: LogNormal, $\mu_{\mathrm{s}}(X) = (X_2 > \frac{1}{2}) + (X_3 < \frac{1}{2}) + (1 - X_1)^{0.25}$, $\sigma_{\mathrm{s}}(X) = \frac{1-X_1}{10}$ 
 $C \mid X$: LogNormal, $\mu_{\mathrm{c}}(X) = (X_2 > \frac{1}{2}) + (X_3 < \frac{1}{2}) + (1 - X_1)^4 + \frac{4}{10}$, $\sigma_{\mathrm{c}}(X) = \frac{X_2}{10}$ |
| 2 | 100 | | $X$ : Unif$([0, 1]^p)$ 
 $T \mid X$: LogNormal, $\mu_{\mathrm{s}}(X) = X_1^{0.25}$, $\sigma_{\mathrm{s}}(X) = 0.1$ 
 $C \mid X$: LogNormal, $\mu_{\mathrm{c}}(X) = X_1^4 + \frac{4}{10}$, $\sigma_{\mathrm{c}}(X) = 0.1$ |
| 3 | 100 | (Candès et al., 2023) | $X$ : Unif$([-1, 1]^p)$ 
 $T \mid X$: LogNormal, $\mu_{\mathrm{s}}(X) = \log(2) + 1 + 0.55(X_1^2 - X_3 X_5)$, $\sigma_{\mathrm{s}}(X) = |X_{10}| + 1$ 
 $C \mid X$: Exponential, $\lambda_{\mathrm{c}}(X) = 0.4$ |

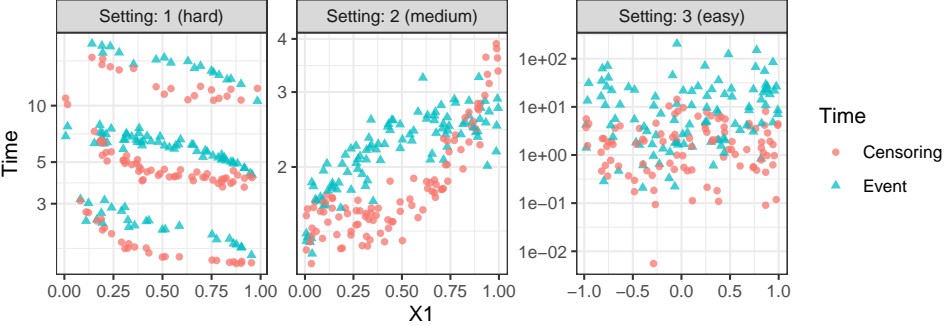

*Figure A1.* Visualization of the synthetic data distribution under the three experimental settings outlined in Table A1. The censoring and event times (in different colors) are plotted against the first covariate, $X_1$.

## A3.2. Impact of the Censoring Model Quality

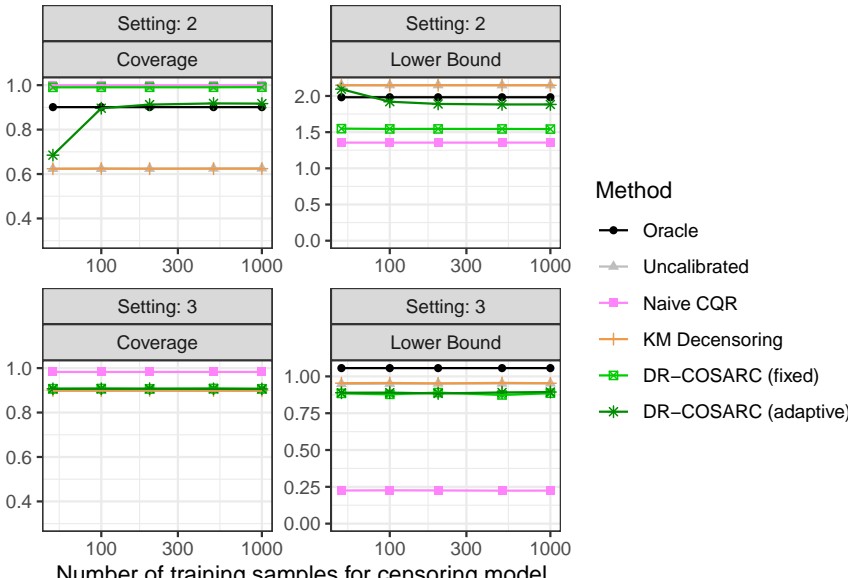

*Figure A2.* Performance on synthetic data of our method as a function of the number of training samples used to fit the censoring model, compared to other approaches. The number of training samples available to fit the survival model is fixed equal to 1000. These experiments are conducted under settings 2 and 3 (see Table A1), where fitting an accurate survival model is easier. Other details are as in Figure 1.

## A3.3. Impact of the Total Number of Training Samples

Figure A3 summarizes the results of experiments similar to those in Figure 2, but with both the survival and censoring models fitted using a varying number of training samples. These results show that our method tends to achieve valid coverage as the sample size increases, even when other approaches either continue to underperform in terms of coverage or produce overly conservative inferences. Figure A4 reports analogous experiments conducted under settings 2 and 3, where our method consistently achieves valid coverage, with its performance remaining largely unaffected by the total number of training samples.

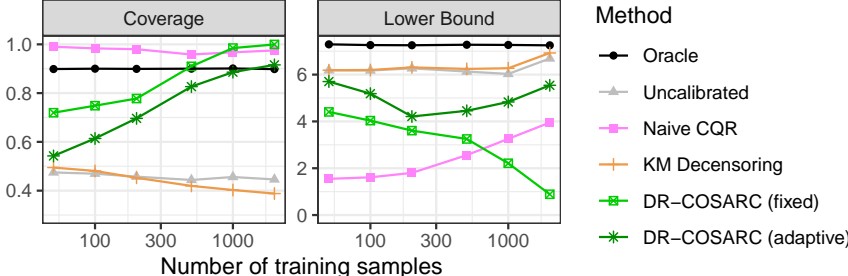

*Figure A3.* Performance of our method (DR-COSARC) as a function of the total number of training samples used to fit the survival and censoring models, in experiments based on synthetic data under setting 1, similar to Figure 2.

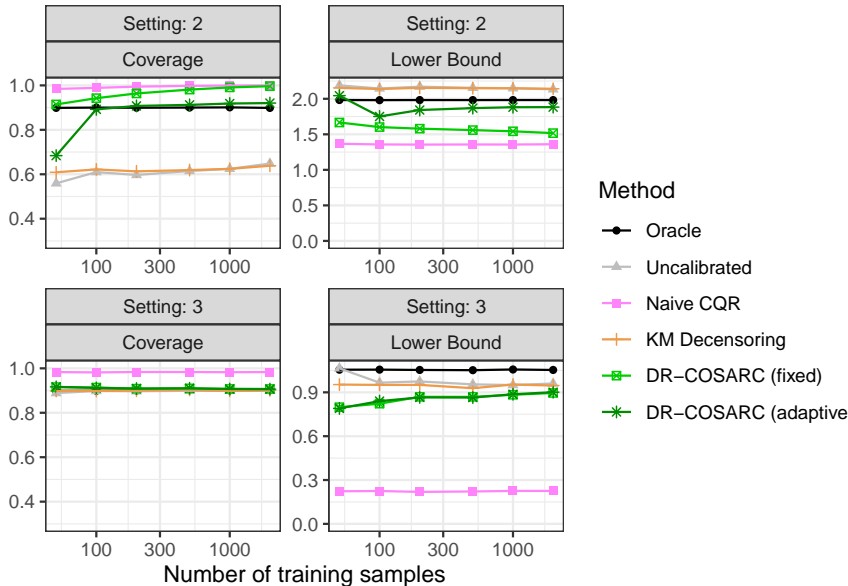

*Figure A4.* Performance of our method as a function of the number of training samples used to fit both the survival and censoring models, in experiments based on synthetic data similar to those of Figure A3. These experiments are conducted under settings 2 and 3 (see Table A1), where fitting an accurate survival model is easier. Other details are as in Figure A3.

### A3.4. Impact of the Number of Covariates in the Censoring Model

Figure A5 presents the results of experiments similar to those in Figure 2, but with varying numbers of covariates used to fit the censoring model, while keeping the number of training and calibration samples fixed at 1000. The results show that our method performs better when the number of covariates used in the censoring model is not too large, highlighting the advantage of leveraging accurate prior information to prevent overfitting. Figure A6 reports similar experiments conducted under the easier settings 2 and 3. In these scenarios, our method consistently achieves valid coverage, and its performance remains largely unaffected by the number of covariates used to fit the censoring model.

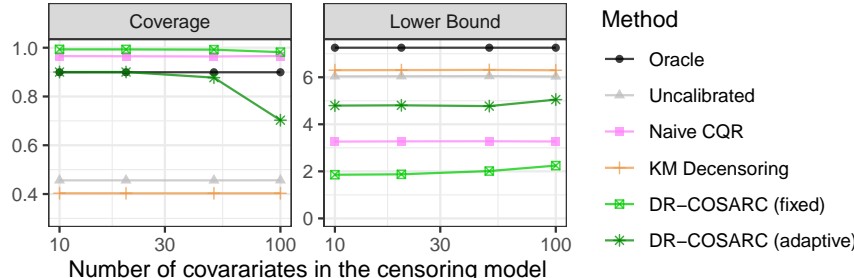

*Figure A5.* Performance on synthetic data of our method and benchmark approaches as a function of the number of covariates used to train the censoring model, with the number of training samples fixed at 1000. The results demonstrate that larger numbers of covariates may increase overfitting, resulting in less accurate censoring models and lower coverage for our method. Other details are as in Figure 2.

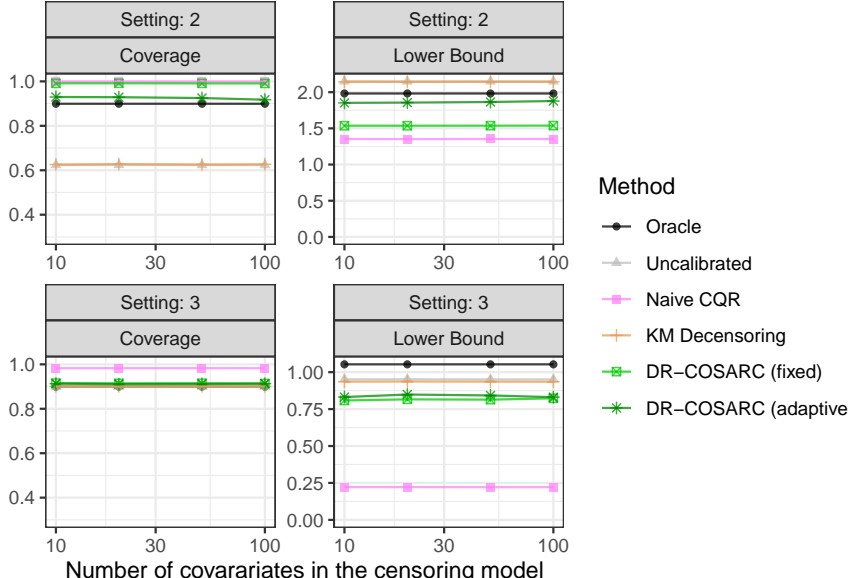

*Figure A6.* Performance on synthetic data of our method and other approaches as a function of the number of covariates used to train the censoring model, with the number of training samples fixed at 1000. These experiments are conducted under settings 2 and 3 (see Table A1), where fitting an accurate survival model is easier. Other details are as in Figure A5.

### A3.5. Robustness to Small Calibration Samples

Figure A7 presents the results of experiments similar to those in Figure 2, but with varying numbers of calibration samples, while keeping the number of training samples fixed at 1000. The results indicate that the average performance of all methods is generally not heavily influenced by the number of calibration samples, although larger calibration sizes do tend to reduce variability across independent repetitions of the experiment. Figure A8 shows similar results for settings 2 and 3.

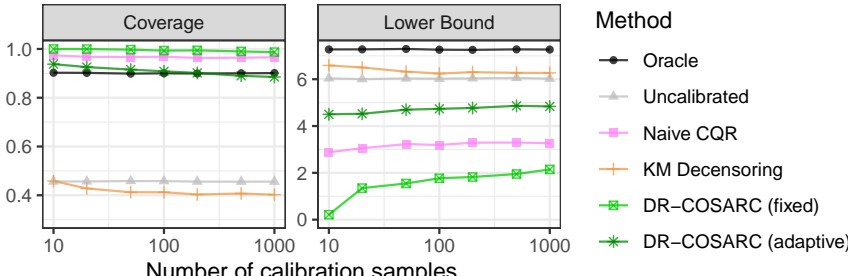

*Figure A7.* Performance on synthetic data of our method (DR-COSARC) and benchmark approaches as a function of the number of calibration samples, with the number of training samples fixed at 1000. The results demonstrate notable robustness to small calibration samples. Other details are as in Figure 2.

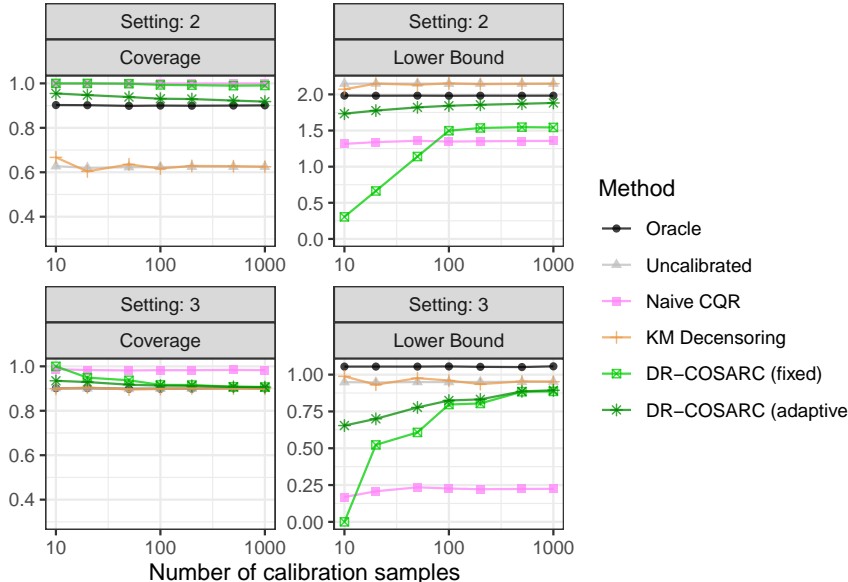

*Figure A8.* Performance on synthetic data of our method and other approaches as a function of the number of calibration samples, with the number of training samples fixed at 1000. These experiments are conducted under settings 2 and 3 (see Table A1), where fitting an accurate survival model is easier. Other details are as in Figure A7.

## A3.6. Impact of the Double Robustness Adjustment

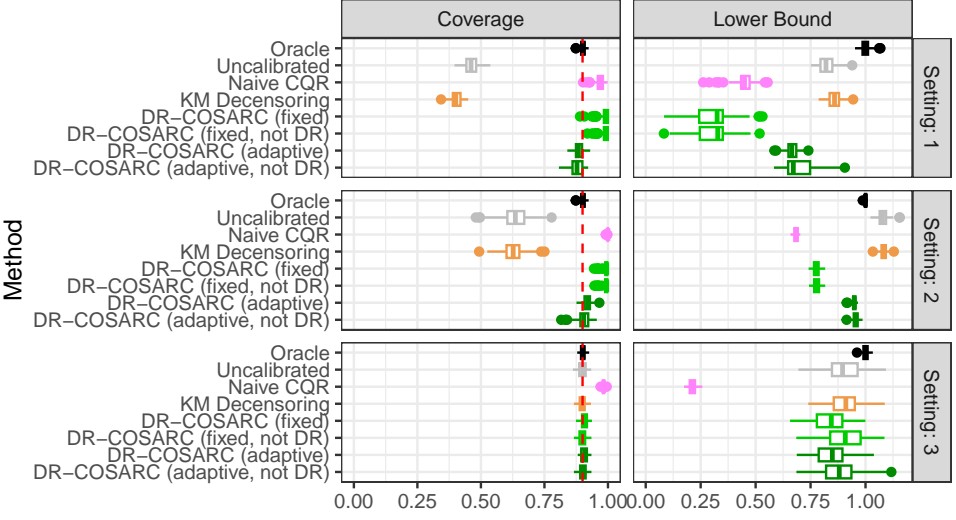

*Figure A9.* Performance on synthetic data under three different settings of our method for constructing lower confidence bounds on the true survival time of a new individual based on right-censored data, compared to existing benchmark approaches. Our method is applied with (default) and without the double robustness adjustment $\hat{L}(X_{n+1}) = \min\{\hat{L}'(X_{n+1}), \hat{q}_\alpha(X_{n+1})\}$, as explained in Algorithms 2 and 3. Other details are as in Figure 1.

### A3.7. Algorithmic Stability

We evaluate the algorithmic stability of our method and the KM Decensoring approach, both of which rely on random imputation of unobserved variables. Our goal is to assess how this randomness impacts the stability of the predicted lower bounds for a fixed dataset and test instance, as a function of the calibration sample size.

To this end, we perform experiments similar to those in Figures A7–A8, varying the calibration set size as a control parameter. For each configuration of the experiment, we independently apply our method (with both fixed and adaptive cutoff strategies) and the KM Decensoring method 10 times per data realization, using different random seeds for the imputation step. Each of these 10 runs uses the same training, calibration, and test sets, and the same trained models. The test set contains 100 points instead of the usual 1000 to reduce the computational cost.

We measure stability by computing the empirical coefficient of variation of the predicted lower bounds across the 10 runs. This coefficient of variation is then averaged across the 100 test points and over 20 independent data realizations.

Figure A10 presents the resulting average coefficient of variations for each method as a function of the calibration sample size. Error bars represent two standard errors of the estimated average coefficient of variation, reflecting the uncertainty in estimating the ideal population coefficient of determination that would be obtained with an infinite number of data realizations (instead of 20).

The results show that the fixed-cutoff implementation of our method tends to have the highest algorithmic variability, while the adaptive-cutoff implementation of our method and the KM Decensoring approach are relatively stable and increasingly become more stable as the calibration sample size grows.

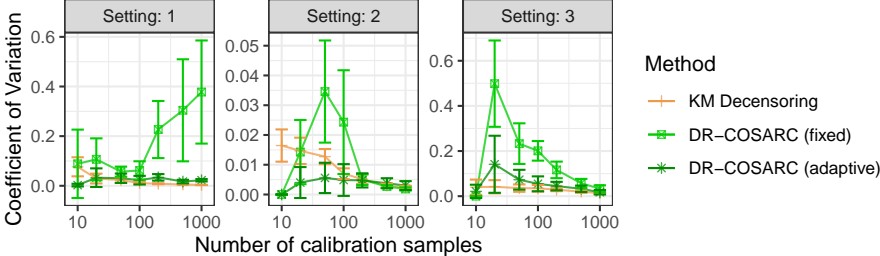

*Figure A10.* Average coefficient of variation of predictive lower bounds obtained with different methods across 10 independent imputations, plotted as a function of the calibration sample size. Results are averaged over 20 independent data realizations. Error bars represent two standard errors. Lower coefficient of variation indicates greater algorithmic stability. Other details are as in Figures A7–A8.

### A3.8. Impact of Different Survival and Censoring Models

We examine here the effect of using different survival and censoring models on the performance of our method and the benchmark approaches. Figures A11 and A12 present results from experiments similar to those in Figure 1, but with different survival and censoring models applied to synthetic data generated under setting 1. Similarly, Figures A13 and A14 report results from setting 2, while Figures A15 and A16 show results from setting 3, where fitting an accurate survival model is relatively easy. Overall, the results are consistent with those presented earlier, empirically demonstrating the double robustness of our method. Notably, Figure A12 reveals that when using a Cox model for the censoring distribution, our method fails to achieve the target 90% coverage, even with a training sample size of 1000, underscoring the limitations of the Cox model in capturing the complex censoring patterns simulated in the particularly challenging setting 1.

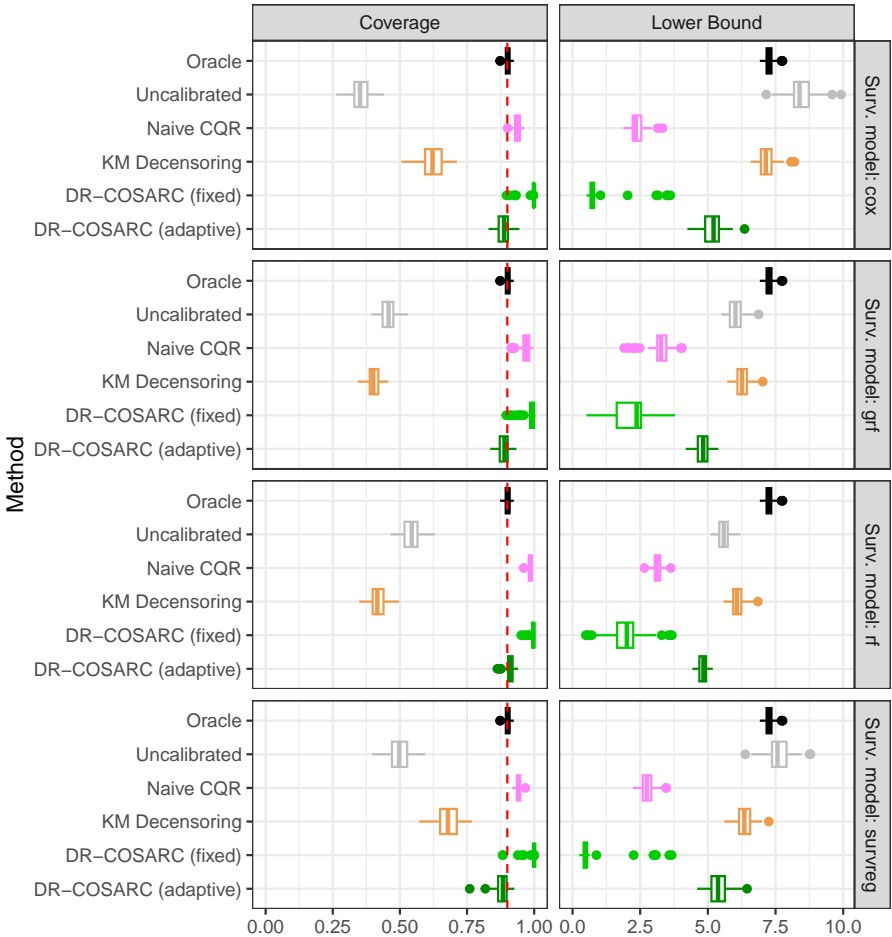

*Figure A11.* Performance on synthetic data of our method for constructing lower confidence bounds on the true survival time of a new individual based on right-censored data, compared to existing approaches. These experiments are conducted under setting 1 (see Table A1), where fitting an accurate survival model is most difficult. Each calibration method is applied with a different type of survival model (fitted using 1000 training samples) and the *grf* censoring model (also fitted using 1000 samples). Other details are as in Figure 1.

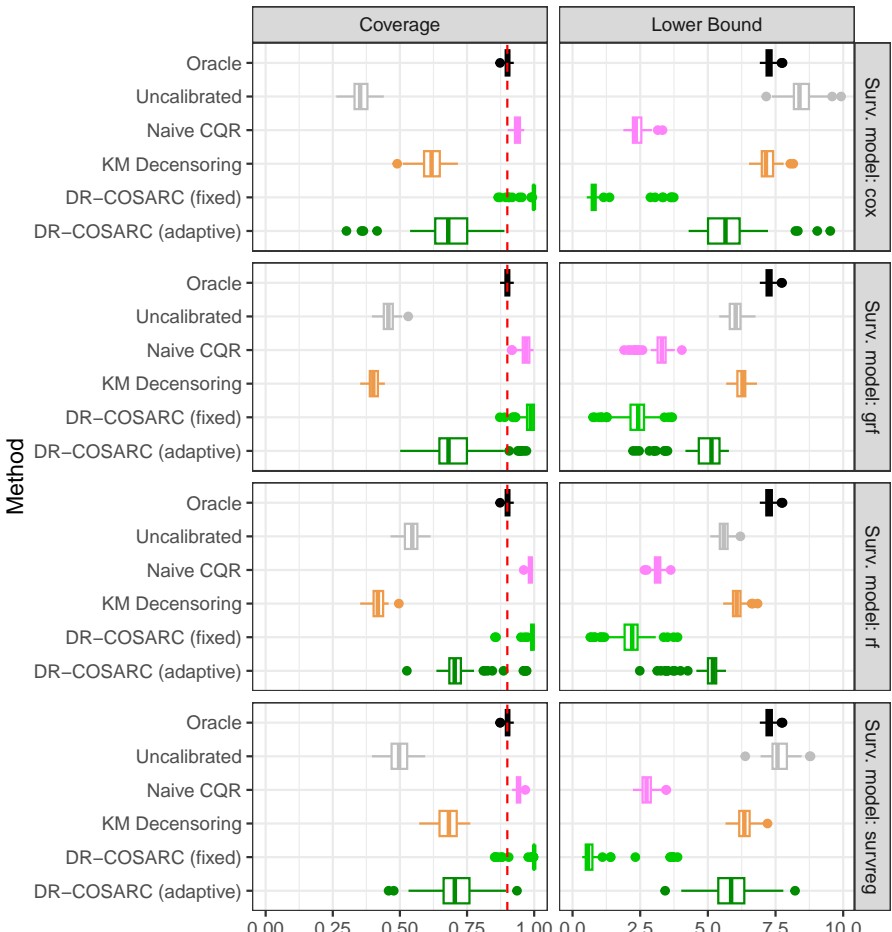

*Figure A12.* Performance on synthetic data of our method for constructing lower confidence bounds on the true survival time of a new individual based on right-censored data, compared to existing approaches. These experiments are conducted under setting 1 (see Table A1), where fitting an accurate survival model is most difficult. Each calibration method is applied with a different type of survival model (fitted using 1000 training samples) and the *cox* censoring model (also fitted using 1000 samples). Other details are as in Figure A11.

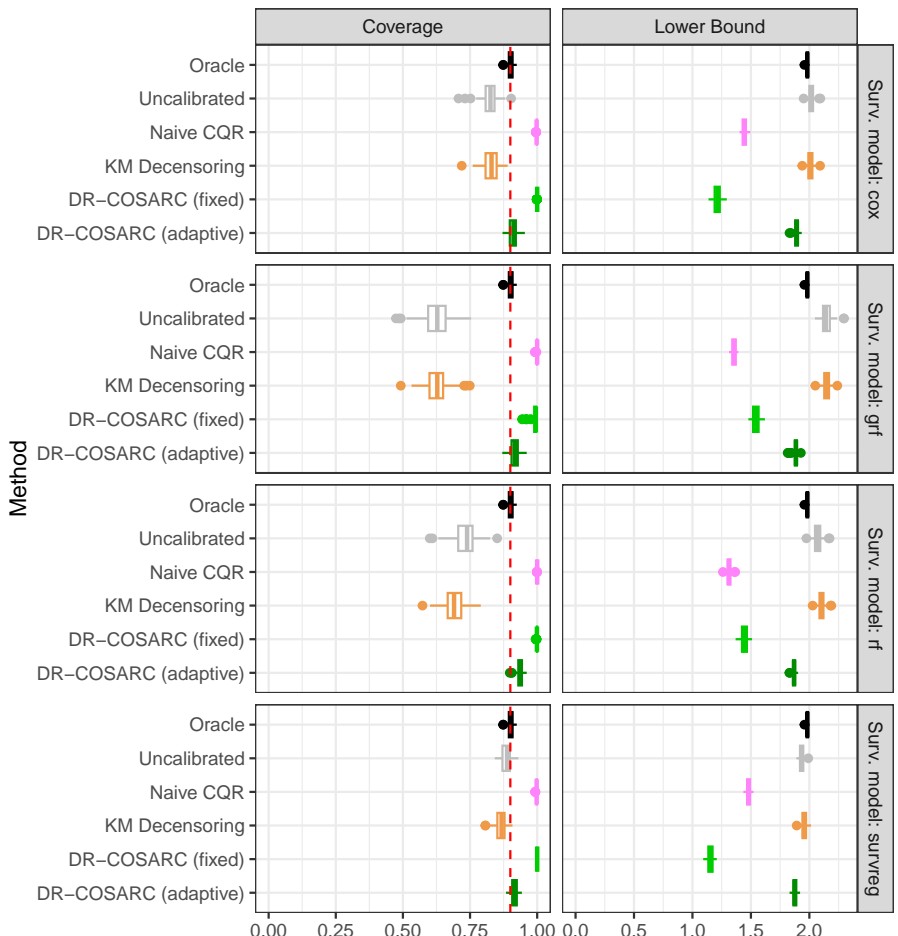

*Figure A13.* Performance on synthetic data of our method for constructing lower confidence bounds on the true survival time of a new individual based on right-censored data, compared to existing approaches. These experiments are conducted under setting 2 (see Table A1), where fitting an accurate survival model is moderately difficult. Each calibration method is applied with a different type of survival model (fitted using 1000 training samples) and the *grf* censoring model (also fitted using 1000 samples). Other details are as in Figure 1.

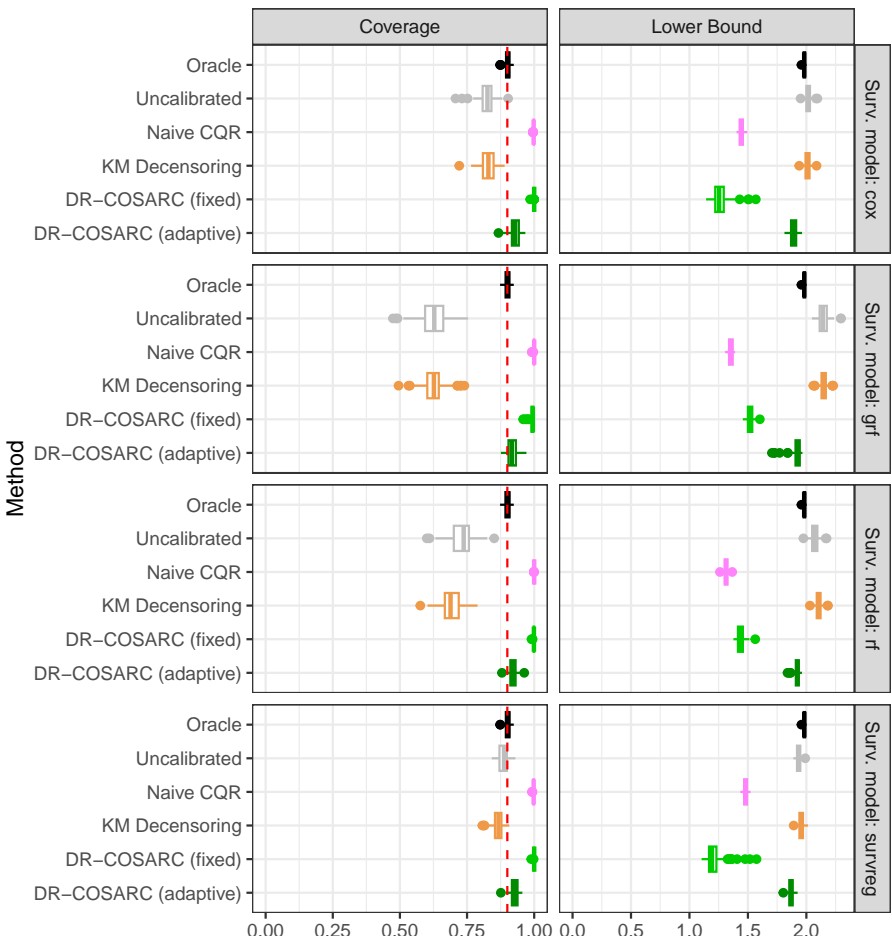

*Figure A14.* Performance on synthetic data of our method for constructing lower confidence bounds on the true survival time of a new individual based on right-censored data, compared to existing approaches. These experiments are conducted under setting 2 (see Table A1), where fitting an accurate survival model is moderately difficult. Each calibration method is applied with a different type of survival model (fitted using 1000 training samples) and the *cox* censoring model (also fitted using 1000 samples). Other details are as in Figure A11.

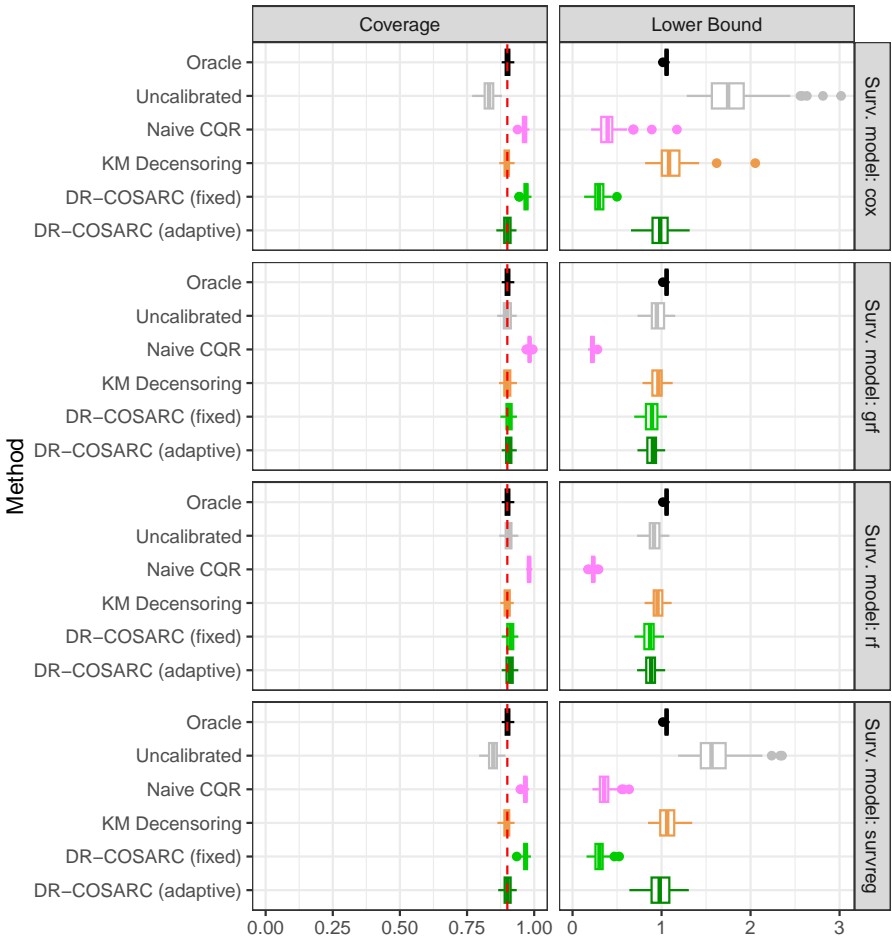

*Figure A15.* Performance on synthetic data of our method for constructing lower confidence bounds on the true survival time of a new individual based on right-censored data, compared to existing approaches. These experiments are conducted under setting 3 (see Table A1), where fitting an accurate survival model is relatively easy and conformal calibration is not crucial. Each calibration method is applied with a different type of survival model (fitted using 1000 training samples) and the *grf* censoring model (also fitted using 1000 samples). Other details are as in Figure A11.

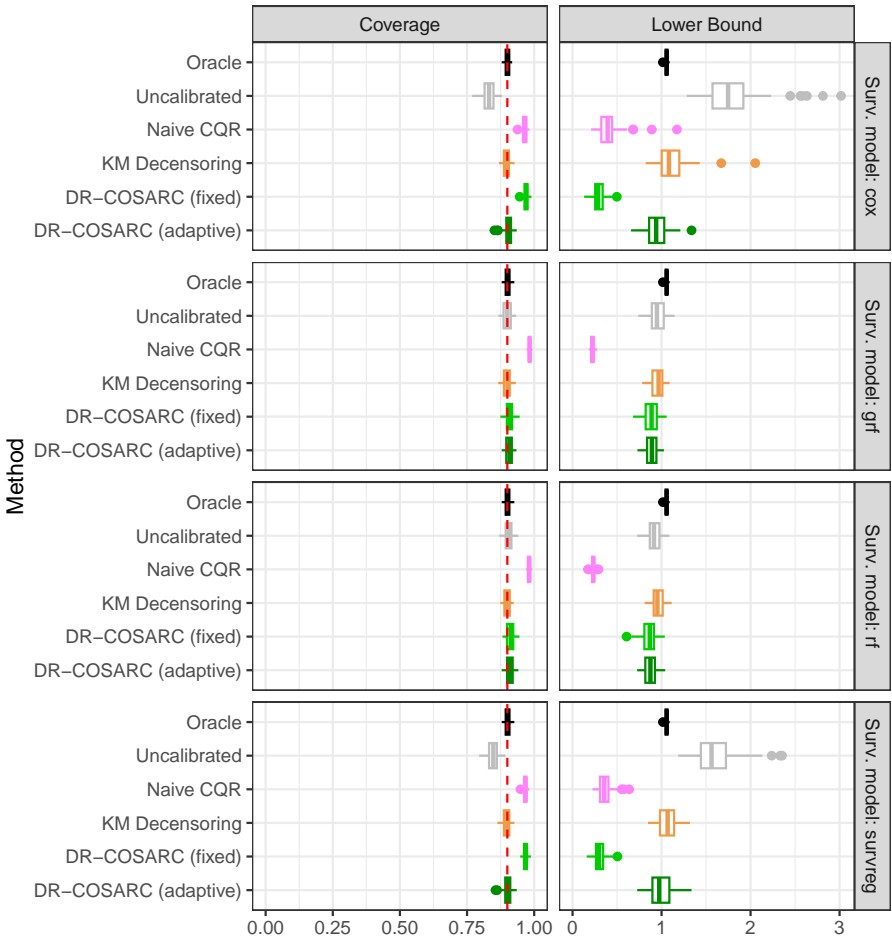

*Figure A16.* Performance on synthetic data of our method for constructing lower confidence bounds on the true survival time of a new individual based on right-censored data, compared to existing approaches. These experiments are conducted under setting 3 (see Table A1), where fitting an accurate survival model is relatively easy and conformal calibration is not crucial. Each calibration method is applied with a different type of survival model (fitted using 1000 training samples) and the *cox* censoring model (also fitted using 1000 samples). Other details are as in Figure A15.

## A3.9. Numerical Experiments under Additional Settings

*Table A2.* Summary of synthetic data generation settings considered in addition to those outlined in Table A1.

| Setting | $p$ | Ref. | Covariate, Survival, and Censoring Distributions |
|---|---|---|---|
| 4 | 100 | (Candès et al., 2023) | $X$ : Unif$([-1, 1]^p)$ 
 $T \mid X$: LogNormal, $\mu_{\mathrm{s}}(X) = \log(2) + 1 + 0.55(X_1^2 - X_3 X_5)$, $\sigma_{\mathrm{s}}(X) = 1$ 
 $C \mid X$: Exponential, $\lambda_{\mathrm{c}}(X) = 0.4$ |
| 5 | 1 | (Gui et al., 2024) | $X$ : Unif$([0, 4]^p)$ 
 $T \mid X$: LogNormal, $\mu_{\mathrm{s}}(X) = 0.632 \times X_1$, $\sigma_{\mathrm{s}}(X) = 2$ 
 $C \mid X$: Exponential, $\lambda_{\mathrm{c}}(X) = 0.1$ |
| 6 | 1 | (Gui et al., 2024) | $X$ : Unif$([0, 4]^p)$ 
 $T \mid X$: LogNormal, $\mu_{\mathrm{s}}(X) = 3(X_1 > 2) + X_1(X_1 < 2)$, $\sigma_{\mathrm{s}}(X) = \frac{1}{2}$ 
 $C \mid X$: Exponential, $\lambda_{\mathrm{c}}(X) = 0.1$ |
| 7 | 1 | (Gui et al., 2024) | $X$ : Unif$([0, 4]^p)$ 
 $T \mid X$: LogNormal, $\mu_{\mathrm{s}}(X) = 2(X_1 > 2) + X_1(X_1 < 2)$, $\sigma_{\mathrm{s}}(X) = \frac{1}{2}$ 
 $C \mid X$: Exponential, $\lambda_{\mathrm{c}}(X) = 0.25 + \frac{X_1 + 6}{100}$ |
| 8 | 1 | (Gui et al., 2024) | $X$ : Unif$([0, 4]^p)$ 
 $T \mid X$: LogNormal, $\mu_{\mathrm{s}}(X) = 3(X_1 > 2) + 1.5X_1(X_1 < 2)$, $\sigma_{\mathrm{s}}(X) = \frac{1}{2}$ 
 $C \mid X$: LogNormal, $\mu_{\mathrm{c}}(X) = 2 + \frac{2 - X_1}{50}$, $\sigma_{\mathrm{c}}(X) = \frac{1}{2}$ |
| 9 | 10 | (Gui et al., 2024) | $X$ : Unif$([0, 4]^p)$ 
 $T \mid X$: LogNormal, $\mu_{\mathrm{s}}(X) = 0.126(X_1 + \sqrt{X_3 X_5}) + 1$, $\sigma_{\mathrm{s}}(X) = \frac{1}{2}$ 
 $C \mid X$: Exponential, $\lambda_{\mathrm{c}}(X) = \frac{X_{10}}{10} + \frac{1}{20}$ |
| 10 | 10 | (Gui et al., 2024) | $X$ : Unif$([0, 4]^p)$ 
 $T \mid X$: LogNormal, $\mu_{\mathrm{s}}(X) = 0.126(X_1 + \sqrt{X_3 X_5}) + 1$, $\sigma_{\mathrm{s}}(X) = \frac{X_2 + 2}{4}$ 
 $C \mid X$: Exponential, $\lambda_{\mathrm{c}}(X) = \frac{X_{10}}{10} + \frac{1}{20}$ |

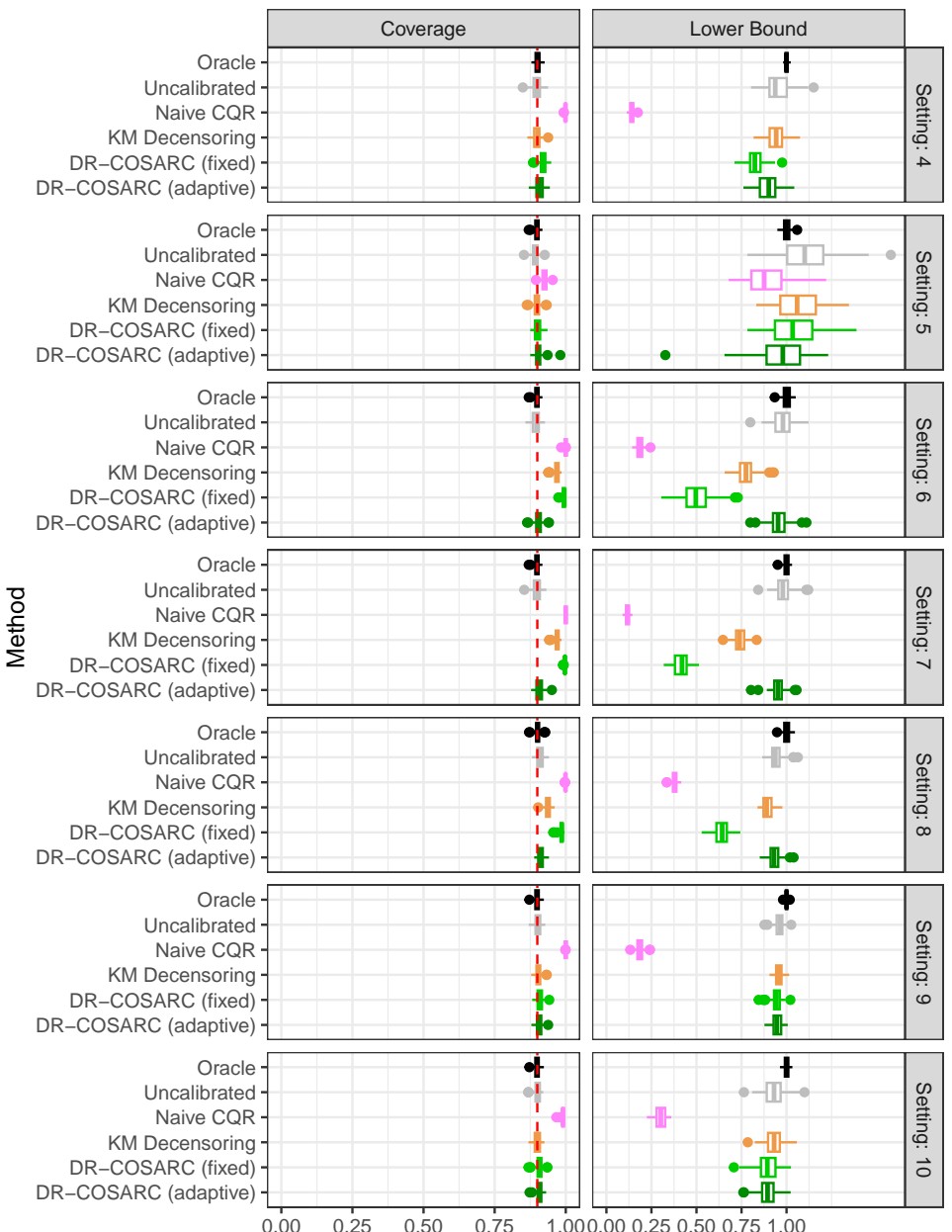

*Figure A17.* Performance on synthetic data under seven additional settings of our method for constructing lower confidence bounds on the true survival time of a new individual based on right-censored data, compared to existing benchmark approaches. These seven settings are borrowed from Candès et al. (2023) and Gui et al. (2024), as outlined in Table A2, and correspond to situations in which fitting an accurate survival model is relatively easy, with rigorous conformal calibration not being as crucial as in settings 1 and 2 of Figure 1. Other details are as in Figure 1.

## A4. Additional Results from Real Data Applications

### A4.1. Data and Pre-Processing

We apply our method to seven datasets: the Veterans' Administration Lung Cancer Trial (VALCT); the Primary Biliary Cirrhosis (PBC) dataset; the German Breast Cancer Study Group (GBSG) dataset; the Molecular Taxonomy of Breast Cancer International Consortium (METABRIC) dataset; the Colon Cancer Chemotherapy (COLON) dataset; the Stanford Heart Transplant Study (HEART); and the Diabetic Retinopathy Study (RETINOPATHY). Table A3 provides details on the number of observations, covariates, and data sources.

The datasets were obtained from various publicly available sources. VALCT, PBC, COLON, HEART, and RETINOPA-THY are included in the `survival` R package. GBSG was sourced from GitHub: `https://github.com/jaredleekatzman/DeepSurv/`. METABRIC was accessed via `https://www.cbioportal.org/study/summary?id=brca_metabric`.

Each dataset underwent a pre-processing pipeline to ensure consistency and prepare the data for analysis. Survival times equal to zero were replaced with half the smallest observed non-zero time. Missing values were imputed using the median for numeric variables and the mode for categorical variables. Factor variables were processed to merge rare levels (frequency below 2%) into an "Other" category, while binary factors with one rare level were removed entirely. Dummy variables were created for all factors, and redundant features were identified and removed using an alias check. Additionally, highly correlated features (correlation above 0.75) were iteratively filtered.

| Dataset | Observations ($n$) | Variables ($p$) | Source | Citation |
|---|---|---|---|---|
| VALCT | 137 | 6 | survival | (Kalbfleisch & Prentice, 2002) |
| PBC | 418 | 17 | survival | (Therneau et al., 2000) |
| GBSG | 2232 | 6 | github.com | (Katzman et al., 2018) |
| METABRIC | 1981 | 41 | cbioportal.org | (Curtis et al., 2012) |
| COLON | 1858 | 11 | survival | (Moertel et al., 1990) |
| HEART | 172 | 4 | survival | (Crowley & Hu, 1977) |
| RETINOPATHY | 394 | 5 | survival | (Blair et al., 1980) |

*Table A3.* Summary of the publicly available survival analysis datasets used in Section 5 of this paper.

### A4.2. Additional Results

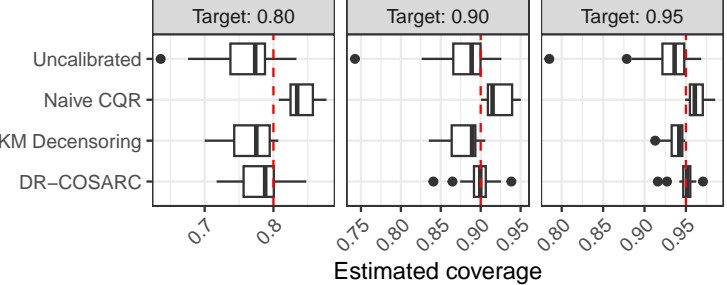

*Figure A18.* Distribution of average (estimated) coverage of survival LPBs computed by different methods, across seven real datasets and four survival models, using different nominal levels $\alpha$. Other details are as in Figure 3.

*Table A4.* Performance on seven publicly available data sets of different methods for constructing survival LPBs. All methods utilize the same survival model (grf) and aim for 90% coverage. Since the performance is evaluated on a censored test set, the coverage cannot be evaluated exactly. Instead, we report empirical lower and upper bounds for the true coverage. Values in parentheses represent twice the standard error. Coverage point estimates that are more than two standard errors below the nominal level are highlighted in red.

| | Estimated Coverage | | | |
| Method | Point | Lower bound | Upper bound | LPB |
|---|---|---|---|---|
| **COLON** | | | | |
| Uncalibrated | 0.91 (0.00) | 0.90 (0.00) | 0.91 (0.01) | 299.15 (3.89) |
| Naive CQR | 0.90 (0.01) | 0.90 (0.01) | 0.91 (0.01) | 296.97 (8.86) |
| KM Decensoring | 0.90 (0.01) | 0.90 (0.01) | 0.90 (0.01) | 306.27 (9.12) |
| DR-COSARC | 0.91 (0.00) | 0.91 (0.01) | 0.91 (0.00) | 286.03 (5.06) |
| **GBSG** | | | | |
| Uncalibrated | 0.90 (0.00) | 0.90 (0.00) | 0.91 (0.00) | 13.10 (0.11) |
| Naive CQR | 0.91 (0.00) | 0.90 (0.01) | 0.92 (0.01) | 12.80 (0.25) |
| KM Decensoring | 0.89 (0.00) | 0.89 (0.01) | 0.90 (0.01) | 13.70 (0.23) |
| DR-COSARC | 0.91 (0.00) | 0.90 (0.00) | 0.91 (0.00) | 12.98 (0.13) |
| **HEART** | | | | |
| Uncalibrated | 0.84 (0.02) | 0.78 (0.03) | 0.90 (0.02) | 14.27 (1.40) |
| Naive CQR | 0.94 (0.01) | 0.92 (0.02) | 0.97 (0.01) | 3.58 (0.81) |
| KM Decensoring | 0.84 (0.02) | 0.77 (0.04) | 0.91 (0.02) | 15.06 (2.28) |
| DR-COSARC | 0.88 (0.02) | 0.83 (0.03) | 0.92 (0.02) | 9.74 (1.43) |
| **METABRIC** | | | | |
| Uncalibrated | 0.90 (0.00) | 0.89 (0.01) | 0.91 (0.01) | 40.86 (0.53) |
| Naive CQR | 0.92 (0.00) | 0.91 (0.01) | 0.93 (0.01) | 37.84 (0.69) |
| KM Decensoring | 0.89 (0.00) | 0.88 (0.01) | 0.90 (0.01) | 42.55 (0.83) |
| DR-COSARC | 0.90 (0.00) | 0.89 (0.01) | 0.91 (0.01) | 40.10 (0.56) |
| **PBC** | | | | |
| Uncalibrated | 0.92 (0.01) | 0.89 (0.01) | 0.95 (0.01) | 853.89 (24.37) |
| Naive CQR | 0.93 (0.01) | 0.91 (0.01) | 0.95 (0.01) | 784.62 (38.83) |
| KM Decensoring | 0.85 (0.01) | 0.81 (0.02) | 0.90 (0.01) | 1035.49 (32.76) |
| DR-COSARC | 0.92 (0.01) | 0.89 (0.01) | 0.95 (0.01) | 852.27 (24.16) |
| **RETINOPATHY** | | | | |
| Uncalibrated | 0.88 (0.01) | 0.87 (0.01) | 0.89 (0.01) | 7.58 (0.38) |
| Naive CQR | 0.90 (0.01) | 0.89 (0.01) | 0.91 (0.01) | 6.66 (0.56) |
| KM Decensoring | 0.88 (0.01) | 0.86 (0.02) | 0.89 (0.01) | 8.03 (0.68) |
| DR-COSARC | 0.90 (0.01) | 0.89 (0.01) | 0.91 (0.01) | 6.19 (0.56) |
| **VALCT** | | | | |
| Uncalibrated | 0.93 (0.01) | 0.93 (0.01) | 0.93 (0.01) | 12.35 (0.69) |
| Naive CQR | 0.94 (0.01) | 0.94 (0.02) | 0.94 (0.02) | 10.66 (1.50) |
| KM Decensoring | 0.90 (0.02) | 0.90 (0.02) | 0.90 (0.02) | 14.47 (1.78) |
| DR-COSARC | 0.94 (0.01) | 0.94 (0.01) | 0.94 (0.01) | 10.17 (0.80) |

*Table A5*. Performance on seven publicly available data sets of different methods for constructing survival LPBs. All methods utilize the same survival model (cox) and aim for 90% coverage. Since the performance is evaluated on a censored test set, the coverage cannot be evaluated exactly. Instead, we report empirical lower and upper bounds for the true coverage. Values in parentheses represent twice the standard error. Coverage point estimates that are more than two standard errors below the nominal level are highlighted in red.

| Method | Estimated Coverage | | | LPB |
|---|---|---|---|---|
| | Point | Lower bound | Upper bound | |
| **COLON** | | | | |
| Uncalibrated | 0.89 (0.00) | 0.89 (0.01) | 0.89 (0.01) | 320.96 (4.88) |
| Naive CQR | 0.90 (0.01) | 0.90 (0.01) | 0.91 (0.01) | 299.68 (9.27) |
| KM Decensoring | 0.90 (0.01) | 0.90 (0.01) | 0.90 (0.01) | 308.78 (9.56) |
| DR-COSARC | 0.91 (0.00) | 0.90 (0.01) | 0.91 (0.01) | 288.30 (9.46) |
| **GBSG** | | | | |
| Uncalibrated | 0.89 (0.00) | 0.89 (0.00) | 0.90 (0.00) | 12.36 (0.09) |
| Naive CQR | 0.91 (0.00) | 0.90 (0.01) | 0.92 (0.01) | 11.38 (0.20) |
| KM Decensoring | 0.90 (0.00) | 0.89 (0.01) | 0.90 (0.01) | 12.23 (0.18) |
| DR-COSARC | 0.90 (0.00) | 0.89 (0.01) | 0.91 (0.00) | 11.87 (0.17) |
| **HEART** | | | | |
| Uncalibrated | 0.74 (0.02) | 0.64 (0.03) | 0.84 (0.02) | 28.52 (1.81) |
| Naive CQR | 0.94 (0.01) | 0.91 (0.02) | 0.97 (0.01) | 6.97 (1.17) |
| KM Decensoring | 0.84 (0.02) | 0.77 (0.04) | 0.91 (0.02) | 17.65 (2.54) |
| DR-COSARC | 0.84 (0.02) | 0.78 (0.03) | 0.90 (0.02) | 13.19 (2.40) |
| **METABRIC** | | | | |
| Uncalibrated | 0.89 (0.00) | 0.87 (0.01) | 0.90 (0.01) | 42.41 (0.50) |
| Naive CQR | 0.91 (0.00) | 0.90 (0.01) | 0.92 (0.01) | 35.02 (1.02) |
| KM Decensoring | 0.89 (0.00) | 0.88 (0.01) | 0.91 (0.01) | 40.55 (1.01) |
| DR-COSARC | 0.90 (0.00) | 0.88 (0.01) | 0.91 (0.01) | 38.79 (1.08) |
| **PBC** | | | | |
| Uncalibrated | 0.85 (0.01) | 0.79 (0.02) | 0.90 (0.01) | 1198.29 (37.23) |
| Naive CQR | 0.94 (0.01) | 0.91 (0.02) | 0.96 (0.01) | 768.98 (56.89) |
| KM Decensoring | 0.87 (0.01) | 0.82 (0.02) | 0.91 (0.02) | 1110.57 (51.64) |
| DR-COSARC | 0.88 (0.01) | 0.85 (0.02) | 0.92 (0.01) | 1020.75 (66.27) |
| **RETINOPATHY** | | | | |
| Uncalibrated | 0.87 (0.01) | 0.85 (0.01) | 0.88 (0.01) | 8.77 (0.47) |
| Naive CQR | 0.91 (0.01) | 0.90 (0.02) | 0.92 (0.01) | 6.55 (0.62) |
| KM Decensoring | 0.88 (0.01) | 0.86 (0.02) | 0.90 (0.02) | 8.24 (0.80) |
| DR-COSARC | 0.89 (0.01) | 0.88 (0.01) | 0.91 (0.01) | 6.57 (0.54) |
| **VALCT** | | | | |
| Uncalibrated | 0.88 (0.01) | 0.87 (0.02) | 0.88 (0.02) | 24.64 (5.55) |
| Naive CQR | 0.94 (0.01) | 0.94 (0.02) | 0.94 (0.02) | 16.36 (5.63) |
| KM Decensoring | 0.89 (0.02) | 0.89 (0.03) | 0.90 (0.02) | 21.93 (5.45) |
| DR-COSARC | 0.91 (0.01) | 0.90 (0.02) | 0.91 (0.02) | 18.05 (4.41) |

*Table A6.* Performance on seven publicly available data sets of different methods for constructing survival LPBs. All methods utilize the same survival model (survreg) and aim for 90% coverage. Since the performance is evaluated on a censored test set, the coverage cannot be evaluated exactly. Instead, we report empirical lower and upper bounds for the true coverage. Values in parentheses represent twice the standard error. Coverage point estimates that are more than two standard errors below the nominal level are highlighted in red.

| Method | Estimated Coverage | | | LPB |
|---|---|---|---|---|
| | Point | Lower bound | Upper bound | |
| **COLON** | | | | |
| Uncalibrated | 0.90 (0.00) | 0.90 (0.01) | 0.90 (0.01) | 369.05 (7.39) |
| Naive CQR | 0.91 (0.00) | 0.91 (0.01) | 0.91 (0.01) | 351.31 (10.46) |
| KM Decensoring | 0.90 (0.01) | 0.90 (0.01) | 0.90 (0.01) | 361.87 (10.75) |
| DR-COSARC | 0.91 (0.00) | 0.91 (0.01) | 0.91 (0.01) | 339.93 (10.70) |
| **GBSG** | | | | |
| Uncalibrated | 0.90 (0.00) | 0.89 (0.00) | 0.91 (0.00) | 13.43 (0.16) |
| Naive CQR | 0.91 (0.00) | 0.90 (0.01) | 0.92 (0.00) | 12.69 (0.21) |
| KM Decensoring | 0.89 (0.00) | 0.88 (0.01) | 0.90 (0.01) | 13.80 (0.24) |
| DR-COSARC | 0.90 (0.00) | 0.89 (0.00) | 0.91 (0.00) | 13.17 (0.14) |
| **HEART** | | | | |
| Uncalibrated | 0.83 (0.02) | 0.75 (0.03) | 0.90 (0.02) | 22.72 (2.74) |
| Naive CQR | 0.94 (0.01) | 0.91 (0.02) | 0.97 (0.01) | 8.91 (1.69) |
| KM Decensoring | 0.84 (0.02) | 0.77 (0.03) | 0.91 (0.02) | 20.75 (2.53) |
| DR-COSARC | 0.86 (0.02) | 0.81 (0.03) | 0.92 (0.02) | 14.22 (2.32) |
| **METABRIC** | | | | |
| Uncalibrated | 0.88 (0.00) | 0.87 (0.01) | 0.90 (0.01) | 46.04 (0.61) |
| Naive CQR | 0.92 (0.00) | 0.91 (0.01) | 0.93 (0.00) | 38.98 (0.86) |
| KM Decensoring | 0.89 (0.00) | 0.87 (0.01) | 0.91 (0.01) | 44.98 (0.94) |
| DR-COSARC | 0.89 (0.00) | 0.88 (0.01) | 0.91 (0.01) | 43.66 (0.80) |
| **PBC** | | | | |
| Uncalibrated | 0.85 (0.01) | 0.79 (0.02) | 0.92 (0.01) | 1197.47 (41.55) |
| Naive CQR | 0.95 (0.01) | 0.92 (0.01) | 0.98 (0.01) | 738.45 (54.18) |
| KM Decensoring | 0.86 (0.01) | 0.79 (0.02) | 0.92 (0.01) | 1167.58 (40.41) |
| DR-COSARC | 0.87 (0.01) | 0.82 (0.02) | 0.93 (0.01) | 1098.76 (43.67) |
| **RETINOPATHY** | | | | |
| Uncalibrated | 0.87 (0.01) | 0.85 (0.01) | 0.88 (0.01) | 9.22 (0.49) |
| Naive CQR | 0.91 (0.01) | 0.89 (0.02) | 0.92 (0.01) | 6.93 (0.65) |
| KM Decensoring | 0.87 (0.01) | 0.86 (0.02) | 0.89 (0.02) | 8.79 (0.78) |
| DR-COSARC | 0.89 (0.01) | 0.88 (0.02) | 0.91 (0.01) | 7.13 (0.59) |
| **VALCT** | | | | |
| Uncalibrated | 0.89 (0.01) | 0.89 (0.02) | 0.90 (0.02) | 21.47 (0.97) |
| Naive CQR | 0.94 (0.01) | 0.93 (0.02) | 0.94 (0.02) | 16.92 (1.92) |
| KM Decensoring | 0.89 (0.02) | 0.88 (0.02) | 0.89 (0.02) | 22.24 (2.15) |
| DR-COSARC | 0.92 (0.01) | 0.91 (0.02) | 0.92 (0.02) | 17.27 (1.67) |

*Table A7*. Performance on seven publicly available data sets of different methods for constructing survival LPBs. All methods utilize the same survival model (rf) and aim for 90% coverage. Since the performance is evaluated on a censored test set, the coverage cannot be evaluated exactly. Instead, we report empirical lower and upper bounds for the true coverage. Values in parentheses represent twice the standard error. Coverage point estimates that are more than two standard errors below the nominal level are highlighted in red.

| Method | Estimated Coverage | | | LPB |
|---|---|---|---|---|
| | Point | Lower bound | Upper bound | |
| **COLON** | | | | |
| Uncalibrated | 0.90 (0.00) | 0.89 (0.01) | 0.91 (0.01) | 403.71 (9.60) |
| Naive CQR | 0.91 (0.01) | 0.90 (0.01) | 0.91 (0.01) | 379.64 (9.90) |
| KM Decensoring | 0.89 (0.01) | 0.89 (0.01) | 0.90 (0.01) | 406.86 (10.83) |
| DR-COSARC | 0.91 (0.00) | 0.90 (0.01) | 0.91 (0.01) | 379.60 (12.36) |
| **GBSG** | | | | |
| Uncalibrated | 0.89 (0.00) | 0.88 (0.01) | 0.90 (0.00) | 14.48 (0.13) |
| Naive CQR | 0.91 (0.00) | 0.90 (0.01) | 0.92 (0.01) | 13.43 (0.26) |
| KM Decensoring | 0.89 (0.00) | 0.88 (0.01) | 0.90 (0.01) | 14.55 (0.25) |
| DR-COSARC | 0.90 (0.00) | 0.89 (0.01) | 0.91 (0.00) | 14.05 (0.19) |
| **HEART** | | | | |
| Uncalibrated | 0.83 (0.02) | 0.76 (0.03) | 0.90 (0.02) | 18.26 (1.93) |
| Naive CQR | 0.95 (0.01) | 0.93 (0.02) | 0.97 (0.01) | 4.55 (1.08) |
| KM Decensoring | 0.83 (0.02) | 0.77 (0.04) | 0.90 (0.02) | 17.58 (2.45) |
| DR-COSARC | 0.87 (0.02) | 0.82 (0.03) | 0.93 (0.02) | 11.45 (1.69) |
| **METABRIC** | | | | |
| Uncalibrated | 0.89 (0.00) | 0.88 (0.01) | 0.91 (0.00) | 43.21 (0.59) |
| Naive CQR | 0.91 (0.00) | 0.90 (0.01) | 0.93 (0.00) | 39.45 (0.75) |
| KM Decensoring | 0.89 (0.00) | 0.87 (0.01) | 0.90 (0.01) | 44.67 (0.81) |
| DR-COSARC | 0.90 (0.00) | 0.88 (0.01) | 0.91 (0.01) | 42.22 (0.70) |
| **PBC** | | | | |
| Uncalibrated | 0.88 (0.01) | 0.82 (0.01) | 0.94 (0.01) | 1057.31 (31.32) |
| Naive CQR | 0.95 (0.01) | 0.91 (0.01) | 0.98 (0.01) | 662.76 (54.81) |
| KM Decensoring | 0.86 (0.01) | 0.79 (0.02) | 0.92 (0.01) | 1117.82 (42.77) |
| DR-COSARC | 0.88 (0.01) | 0.83 (0.02) | 0.94 (0.01) | 1012.21 (49.52) |
| **RETINOPATHY** | | | | |
| Uncalibrated | 0.88 (0.01) | 0.86 (0.02) | 0.89 (0.01) | 9.75 (0.62) |
| Naive CQR | 0.91 (0.01) | 0.89 (0.01) | 0.92 (0.01) | 7.90 (0.63) |
| KM Decensoring | 0.88 (0.01) | 0.86 (0.02) | 0.90 (0.01) | 9.78 (0.67) |
| DR-COSARC | 0.89 (0.01) | 0.88 (0.01) | 0.91 (0.01) | 8.19 (0.61) |
| **VALCT** | | | | |
| Uncalibrated | 0.90 (0.01) | 0.90 (0.02) | 0.90 (0.02) | 33.90 (9.67) |
| Naive CQR | 0.94 (0.01) | 0.94 (0.02) | 0.94 (0.02) | 19.12 (6.61) |
| KM Decensoring | 0.91 (0.02) | 0.91 (0.02) | 0.91 (0.02) | 29.50 (8.73) |
| DR-COSARC | 0.93 (0.01) | 0.93 (0.01) | 0.93 (0.01) | 21.33 (6.13) |

## A5. Mathematical Proofs

### A5.1. Proof of Proposition 2.2

*Proof of Proposition 2.2.* Since $P_{X,\tilde{T},C} = P_{X,\tilde{T}} \cdot P_{C|X,\tilde{T}}$, it suffices to demonstrate that $\hat{C}$ is a random sample from $P_{C|X,\tilde{T}}$. This is immediately true when $\tilde{T} > C$, as in those cases, we have $\hat{C} = C$. Therefore, it remains to show that $\hat{C}$ is a random sample from $P_{C|X,C>\tilde{T},\tilde{T}}$ when $\hat{C} > \tilde{T}$.

To see this, we apply the definition of conditional probability along with the assumption of conditionally independent censoring, $T \perp\!\!\!\perp C \mid X$. The distribution of $C$ given $(X = x, T < C, T = t)$ for any $x$ and $t$ can be written as:

$$
\begin{aligned}
p_{C|X,C>\tilde{T},\tilde{T}}(c \mid x,t) &= p(C = c \mid X = x, T < C, T = t) \\
&= p(C = c \mid X = x, C > t, T = t) \\
&\propto p(C = c, C > t \mid X = x, T = t) \\
&\propto p(C = c, C > t \mid X = x) \qquad \text{(since } T \perp\!\!\!\perp C \mid X) \\
&= \frac{f_{C|X}(c \mid x)}{A(x,t)} \mathbb{I}(c > t),
\end{aligned}
$$

where $A(x,t) = \int_t^\infty f_{C|X}(c \mid x)dc$ is a normalizing constant.

If $\hat{f}_{C|X} = f_{C|X}$, this matches the procedure used in Algorithm A4 to sample $\hat{C}$ when $C > \tilde{T}$, completing the proof. For clarity, note that the subscripts for probability density functions $p$ have been omitted where possible, to simplify the notation without introducing ambiguity. □

### A5.2. Auxiliary Theoretical Results

**Lemma A1.** *Let $\{(X_i, T_i, C_i)\}_{i=1}^n$ be i.i.d. random samples from some distribution $P_{X,T,C} = P_X \cdot P_{T|X} \cdot P_{C|X}$, under Assumption 2.1. For each $i \in [n]$, define $\tilde{T}_i = T_i \wedge C_i$ and $E_i = \mathbb{I}(T_i < C_i)$. Let $C_1, \ldots, C_n$ denote the set of imputed censoring times output by Algorithm 1, based on the data $\{(X_i, \tilde{T}_i, E_i)\}_{i=1}^n$ and any fixed censoring model $\hat{\mathcal{M}}^{cens}$. Then, the total variation distance between the distributions of $\{(X_i, \tilde{T}_i, C_i)\}_{i=1}^n$ and $\{(X_i, T_i, C_i')\}_{i=1}^n$, denoted as $d_{TV}(P_{(X,\tilde{T},C)^n}, P_{(X,\tilde{T},C')^n})$, satisfies*

$$
d_{TV}(P_{(X,\tilde{T},C)^n}, P_{(X,\tilde{T},C')^n}) \leq \frac{n}{2} \mathbb{E}_{(X,\tilde{T}) \sim P_{X,\tilde{T}}} \left[ \int_{\tilde{T}}^\infty \left| \frac{f_{C|X}(c \mid X)}{A(X,\tilde{T})} - \frac{\hat{f}_{C|X}(c \mid X)}{\hat{A}(X,\tilde{T})} \right| dc \right],
$$

*where $P_{X,\tilde{T}}$ denote the joint distribution of $(X, \tilde{T})$, for $(X, T, C) \sim P_{X,T,C}$ and $\tilde{T} = T \wedge C$.*

*Proof of Lemma A1.* Note that

$$
\begin{aligned}
d_{TV}(P_{(X,\tilde{T},C)^n}, P_{(X,\tilde{T},C')^n}) &\leq n \cdot d_{TV}(P_{X,\tilde{T},C}, P_{X,\tilde{T},C'}) \\
&= n \cdot \mathbb{E}_{(X,\tilde{T}) \sim P_{X,\tilde{T}}} \left[ d_{TV}(P_{C|X,\tilde{T}}, P_{C'|X,\tilde{T}}) \right] \\
&= n \cdot \mathbb{E}_{(X,T) \sim P_{X,T}} \left[ d_{TV}(P_{C|X,\tilde{T},C>\tilde{T}}, P_{C'|X,\tilde{T},C>\tilde{T}}) \right] \\
&= \frac{n}{2} \cdot \mathbb{E}_{(X,\tilde{T}) \sim P_{X,\tilde{T}}} \left[ \int_{\tilde{T}}^\infty \left| \frac{f_{C|X}(c \mid X)}{A(X,\tilde{T})} - \frac{\hat{f}_{C|X}(c \mid X)}{\hat{A}(X,\tilde{T})} \right| dc \right],
\end{aligned}
$$

where the first equality follows directly from the definition of the total variation distance, and the second equality follows from the fact that $C' = C$ almost surely if $C < T$. □

**Theorem A2.** *Consider a random sample $(X, T)$ from some distribution $P_{X,T} = P_X \cdot P_{T|X}$. For any $x \in \mathcal{X}$, let $\hat{q}_\alpha(x)$ denote an estimate of the $\alpha$ quantile of the conditional distribution of $T \mid X = x$. Assume the function $\hat{q}_\alpha : \mathcal{X} \mapsto \mathbb{R}_+$ depends on a training data set independent of $(X, T)$. Assume also that*

*(A.A2.1) There exist a constant $b > 0$, and a function $q_\alpha : \mathcal{X} \mapsto \mathbb{R}_+$, such that, for any $\epsilon > 0$,*

$$\mathbb{P}\left[T \geq q_\alpha(x) + \epsilon \mid X = x\right] \geq 1 - \alpha - b\epsilon, \text{ almost surely with respect to } P_X.$$

*For any $x \in \mathcal{X}$, define $\mathcal{E}(x) := |\hat{q}_\alpha(x) - q_\alpha(x)|$. Then, for any constant $\ell > 0$,*

$$\mathbb{P}\left[T \geq \hat{q}_\alpha(X)\right] \geq 1 - \alpha - (b+1)\left(\mathbb{E}\left[\mathcal{E}^\ell(X)\right]\right)^{1/(1+\ell)}, \tag{A7}$$

*and, for any $\beta \in (0,1)$, with probability at least $1 - \beta$,*

$$\mathbb{P}\left[T \geq \hat{q}_\alpha(X) \mid X\right] \geq 1 - \alpha - b\frac{\left(\mathbb{E}\left[\mathcal{E}^\ell(X)\right]\right)^{1/\ell}}{\beta^{1/\ell}}. \tag{A8}$$

*Proof of Theorem A2.* Define $\mathcal{E}(x) = |\hat{q}_\alpha(x) - q_\alpha(x)|$. Note that

$$\begin{aligned}
\mathbb{P}&\left[T \geq \hat{q}_\alpha(x) \mid X = x\right] \\
&\geq \mathbb{P}\left[T \geq q_\alpha(x) + \epsilon + \hat{q}_\alpha(x) - q_\alpha(x) - \epsilon \mid X = x\right] \\
&\geq \mathbb{P}\left[T \geq q_\alpha(x) + \epsilon + |\hat{q}_\alpha(x) - q_\alpha(x)| - \epsilon \mid X = x\right] \\
&\geq \mathbb{P}\left[T \geq q_\alpha(x) + \epsilon \mid X = x\right] - \mathbb{I}\left[\mathcal{E}(x) > \epsilon\right] \\
&\geq 1 - \alpha - b\epsilon - \mathbb{I}\left[\mathcal{E}(x) > \epsilon\right],
\end{aligned} \tag{A9}$$

where the last inequality follows from Assumption A.A2.1. Therefore, by Markov's inequality, for any $\ell > 0$,

$$\mathbb{P}\left[T \geq \hat{q}_\alpha(X)\right] \geq 1 - \alpha - b\epsilon - \frac{\mathbb{E}\left[\mathcal{E}^\ell(X)\right]}{\epsilon^\ell}.$$

Consider now the choice

$$\epsilon = \left(\mathbb{E}\left[\mathcal{E}^\ell(X)\right]\right)^{1/(1+\ell)},$$

which leads to:

$$\mathbb{P}\left[T \geq \hat{q}_\alpha(X; \tilde{\mathcal{D}}_{\text{cal}})\right] \geq 1 - \alpha - (b+1)\left(\mathbb{E}\left[\mathcal{E}^\ell(X)\right]\right)^{1/(1+\ell)},$$

This completes the proof of the unconditional result (A7).

We will now prove the conditional result (A8). For any $\beta \in (0,1)$, consider

$$\epsilon = \frac{\left(\mathbb{E}\left[\mathcal{E}^\ell(X)\right]\right)^{1/\ell}}{\beta^{1/\ell}},$$

which, through Markov's inequality, leads to

$$\mathbb{P}\left[\mathcal{E}^\ell(X) > \epsilon\right] \leq \frac{\mathbb{E}\left[\mathcal{E}^\ell(X)\right]}{\epsilon^\ell} = \beta.$$

Combined with Equation (A9), this implies that, with probability at least $1 - \beta$,

$$\mathbb{P}\left[T \geq \hat{q}_\alpha(x) \mid X\right] \geq 1 - \alpha - b\frac{\left(\mathbb{E}\left[\mathcal{E}^\ell(X)\right]\right)^{1/\ell}}{\beta^{1/\ell}}.$$

$\square$

## A5.3. Double Robustness of Algorithm A4

### A5.3.1. NON-ASYMPTOTIC THEORY

We begin by establishing a non-asymptotic theory for the double robustness of Algorithm A4. To achieve this, we present two key results: Theorem A3 and Theorem A4, each addressing one side of double robustness. The first result focuses on the validity of Algorithm A4 when the censoring model is accurately estimated, while the second addresses its validity when the survival model is accurately estimated, leveraging the auxiliary result stated in Theorem A2.

**Robustness when the censoring model is accurate**

**Theorem A3.** *Let $\{(X_i, T_i, C_i)\}_{i=1}^n$ be i.i.d. random samples from some distribution $P_{X,T,C} = P_X \cdot P_{T|X} \cdot P_{C|X}$, under Assumption 2.1. Let $f_{C|X}(c \mid x)$ be the probability density of $P_{C|X}$. Consider an independent random test sample $(X_{n+1}, T_{n+1}) \sim P_X \cdot P_{T|X}$. Let $\hat{L}(X_{n+1}; \tilde{\mathcal{D}}_{cal})$ indicate the lower bound output by Algorithm A4, based on input calibration data $\{(X_i, \tilde{T}_i, E_i)\}_{i=1}^n$ with $\tilde{T}_i = T_i \wedge C_i$ and $E_i = \mathbb{I}(T_i < C_i)$. Then, this lower bound satisfies:*

$$\mathbb{P}\left[T_{n+1} \wedge c_0 \geq \hat{L}(X_{n+1}; \tilde{\mathcal{D}}_{cal})\right] \geq 1 - \alpha - \frac{1}{2}\mathbb{E}\left[\left|\frac{1}{\hat{c}(X)} - \frac{1}{c(X)}\right|\right] - \psi\left(f_{C|X}, \hat{f}_{C|X}\right),$$

*where*

$$\psi\left(f_{C|X}, \hat{f}_{C|X}\right) = \frac{n}{2} \cdot \mathbb{E}\left[\int_{\tilde{T}}^{\infty}\left|\frac{f_{C|X}(c \mid X)}{A(X, \tilde{T})} - \frac{\hat{f}_{C|X}(c \mid X)}{\hat{A}(X, \tilde{T})}\right| dc\right].$$

*Above, the expectation is taken with respect to a random sample $(X, \tilde{T})$, for $(X, T, C) \sim P_{X,T,C}$ and $\tilde{T} = T \wedge C$.*

Intuitively, Theorem A3 tells us that the finite-sample coverage achieved by our method depends on how well we can estimate the censoring distribution, $P_{C|X}$. In the special case where $f_{C|X}(c \mid x) = \hat{f}_{C|X}(c \mid x)$ for all $c, x$, which corresponds to statistically exact imputation, then $\psi(f_{C|X}, \hat{f}_{C|X}) = 0$ and the finite-sample bound given by Theorem A3 becomes the same as the bound obtained by Candès et al. (2023) under type-I censoring.

*Proof of Theorem A3.* The high-level idea of this proof is to connect the lower bound output by Algorithm A4 to the imaginary lower bound which would be obtained by applying Algorithm A1, the approach originally proposed by Candès et al. (2023), to an imaginary data set sampled from the same distribution but subject to type-I instead of right censoring. We know from Proposition 2.2 that, if the imputation phase of Algorithm A4 utilizes an accurate survival model, the two aforementioned methods become equivalent, and thus the desired result should follow from Theorem B.1 in Candès et al. (2023). We will now make this intuition precise.

Let $\mathcal{D}_{\text{cal}}$ denote the ideal calibration data set containing the true censoring times $C_i$ along with the corresponding values of $X_i$ and $\tilde{T}_i$; i.e.,

$$\mathcal{D}_{\text{cal}}^* = \left\{(X_i, \tilde{T}_i, C_i)\right\}_{i=1}^n.$$

Let $\hat{L}^*(X_{n+1}; \mathcal{D}_{\text{cal}}^*)$ denote the imaginary output that one would obtain by applying Algorithm A4 using $\mathcal{D}_{\text{cal}}^*$ instead of $\tilde{\mathcal{D}}_{\text{cal}}$ in the calibration phase. Equivalently, $\hat{L}^*(X_{n+1}; \mathcal{D}_{\text{cal}}^*)$ is the lower bound that would be produced under type-I censoring by

Algorithm A1, the approach originally proposed by Candès et al. (2023). Then,

$$
\mathbb{P}\left[T_{n+1} \wedge c_0 < \hat{L}(X_{n+1}; \tilde{\mathcal{D}}_{\text{cal}})\right]
$$

$$
\leq \mathbb{P}\left[T_{n+1} \wedge c_0 < \hat{L}^*(X_{n+1}; \mathcal{D}_{\text{cal}}^*)\right]
$$

$$
+ \mathbb{P}\left[T_{n+1} \wedge c_0 < \hat{L}(X_{n+1}; \tilde{\mathcal{D}}_{\text{cal}})\right] - \mathbb{P}\left[T_{n+1} \wedge c_0 < \hat{L}^*(X_{n+1}; \mathcal{D}_{\text{cal}}^*)\right]
$$

$$
\leq \mathbb{P}\left[T_{n+1} \wedge c_0 < \hat{L}^*(X_{n+1}; \mathcal{D}_{\text{cal}}^*)\right] + d_{\text{TV}}(P_{(X,\tilde{T},C)^n}, P_{(X,\tilde{T},C')^n})
$$

$$
\overset{(1)}{\leq} \alpha + \frac{1}{2}\mathbb{E}_{X \sim P_X}\left[\left|\frac{1}{\hat{c}(X)} - \frac{1}{c(X)}\right|\right] + d_{\text{TV}}(P_{(X,\tilde{T},C)^n}, P_{(X,\tilde{T},C')^n})
$$

$$
\overset{(2)}{\leq} \alpha + \frac{1}{2}\mathbb{E}_{X \sim P_X}\left[\left|\frac{1}{\hat{c}(X)} - \frac{1}{c(X)}\right|\right] + \frac{n}{2} \cdot \mathbb{E}_{(X,\tilde{T}) \sim P_{X,\tilde{T}}}\left[\int_{\tilde{T}}^{\infty}\left|\frac{f_{C|X}(c \mid X)}{A(X,\tilde{T})} - \frac{\hat{f}_{C|X}(c \mid X)}{\hat{A}(X,\tilde{T})}\right| dc\right],
$$

where $d_{\text{TV}}(P_{(X,\tilde{T},C)^n}, P_{(X,\tilde{T},C')^n})$ denotes the total variation distance between the distributions of $\mathcal{D}_{\text{cal}}$ and $\mathcal{D}_{\text{cal}}^*$, the inequality (1) follows directly from Theorem B.1 in Candès et al. (2023) applied with $w(x) = 1/c(x)$ and $\hat{w}(x) = 1/\hat{c}(x)$, and the inequality (2) follows from Lemma A1. $\square$

### Robustness when the survival model is accurate

**Theorem A4.** *Consider a random sample $(X, T)$ from some distribution $P_{X,T} = P_X \cdot P_{T|X}$. Let $\hat{L}(X; \tilde{\mathcal{D}}_{cal})$ indicate the corresponding survival lower bound output by Algorithm A4. For any $x \in \mathcal{X}$, let $\hat{q}_\alpha(x)$ denote the estimate of the $\alpha$ quantile of the conditional distribution of $T \mid X = x$ utilized in the final phase of Algorithm A4, with the function $\hat{q}_\alpha : \mathcal{X} \mapsto \mathbb{R}_+$ depending only on $\hat{\mathcal{M}}^{surv}$. Assume also that there exist a constant $b > 0$, and a function $q_\alpha : \mathcal{X} \mapsto \mathbb{R}_+$, such that, for any $\epsilon > 0$,*

$$
\mathbb{P}\left[T \geq q_\alpha(x) + \epsilon \mid X = x\right] \geq 1 - \alpha - b\epsilon, \text{ almost surely with respect to } P_X.
$$

*Then, for any constant $\ell > 0$,*

$$
\mathbb{P}\left[T \geq \hat{L}(X; \tilde{\mathcal{D}}_{cal})\right] \geq 1 - \alpha - (b+1)\left(\mathbb{E}\left[\mathcal{E}^\ell(X)\right]\right)^{1/(1+\ell)}. \tag{A10}
$$

*and, for any $\beta \in (0,1)$, with probability at least $1 - \beta$,*

$$
\mathbb{P}\left[T \geq \hat{L}(X; \tilde{\mathcal{D}}_{cal}) \mid X\right] \geq 1 - \alpha - b\frac{\left(\mathbb{E}\left[\mathcal{E}^\ell(X)\right]\right)^{1/\ell}}{\beta^{1/\ell}}. \tag{A11}
$$

*Proof of Theorem A4.* The proof of this result is an immediate consequence of Theorem A2, since by design Algorithm A4 leads to $\hat{L}(x; \tilde{\mathcal{D}}_{cal}) \leq \hat{q}_\alpha(x)$ almost surely for any $x \in \mathcal{X}$. $\square$

A5.3.2. ASYMPTOTIC THEORY

*Proof of Theorem 3.3.* We begin by proving the unconditional result. For this, we consider two cases separately, relying on Theorems A3 and A4 respectively.

- Suppose Assumption 3.1 holds. In this case, the result follows immediately from Theorem A3, which tells us that, for any fixed $n$ and $N$,

$$
\mathbb{P}\left[T_{n+1} \geq \hat{L}(X_{n+1}; \tilde{\mathcal{D}}_{\text{cal}})\right]
$$

$$
\geq \mathbb{P}\left[T_{n+1} \wedge c_0 \geq \hat{L}(X_{n+1}; \tilde{\mathcal{D}}_{\text{cal}})\right]
$$

$$
\geq 1 - \alpha - \frac{1}{2}\mathbb{E}\left[\left|\frac{1}{\hat{c}(X)} - \frac{1}{c(X)}\right|\right] - \frac{n}{2} \cdot \mathbb{E}\left[\int_{\tilde{T}}^{\infty}\left|\frac{f_{C|X}(c \mid X)}{A(X,\tilde{T})} - \frac{\hat{f}_{C|X}(c \mid X)}{\hat{A}(X,\tilde{T})}\right| dc\right].
$$

The proof is then completed by noting that the asymptotic limit of the right-hand-side term above, for $N, n \to \infty$ is $1 - \alpha$ under Assumption 3.1.

- Suppose Assumption 3.2 holds. In this case, we can obtain the desired result by applying Theorem A4 with $\ell = 1$, which tells us that

$$\mathbb{P}\left[T_{n+1} \geq \hat{L}(X; \tilde{\mathcal{D}}_{\text{cal}})\right] \geq 1 - \alpha - (b+1)\left(\mathbb{E}\left[\mathcal{E}(X)\right]\right)^{1/2}.$$

The proof is then completed by noting that the asymptotic limit of the right-hand-side term above, for $N \to \infty$ is $1 - \alpha$ under Assumption 3.2.

Let us now turn to proving the conditional result. Under Assumption 3.2, applying Theorem A4 with $\ell = 1$ tells us that, for fixed $N$ and any $\beta \in (0, 1)$,

$$\mathbb{P}\left[\mathbb{P}\left[T \geq \hat{L}(X; \tilde{\mathcal{D}}_{\text{cal}}) \mid X\right] \geq 1 - \alpha - b\frac{\mathbb{E}\left[\mathcal{E}(X)\right]}{\beta}\right] \geq 1 - \beta.$$

In particular, choosing $\beta = \sqrt{\mathbb{E}\left[\mathcal{E}(X)\right]}$ completes the proof, because

$$\mathbb{P}\left[\mathbb{P}\left[T \geq \hat{L}(X; \tilde{\mathcal{D}}_{\text{cal}}) \mid X\right] \geq 1 - \alpha - b\sqrt{\mathbb{E}\left[\mathcal{E}(X)\right]}\right] \geq 1 - \sqrt{\mathbb{E}\left[\mathcal{E}(X)\right]}.$$

$\square$

### A5.4. Double Robustness of Algorithm A5

A5.4.1. NON-ASYMPTOTIC THEORY

**Robustness when the censoring model is accurate**

**Theorem A5** (Adapted from Theorem 3 in Gui et al. (2024)). *Let $\{(X_i, T_i, C_i)\}_{i=1}^n$ be i.i.d. random samples from some distribution $P_{X,T,C} = P_X \cdot P_{T|X} \cdot P_{C|X}$, under Assumption 2.1. Assume that $\hat{f}_a(x)$ is continuous in $a$ and that there exists some constant $\hat{\gamma}_a > 0$ such that $1/\hat{c}_a(x) \leq \hat{\gamma}_a$ for $P_X$-almost all $x$. Consider an independent random test sample $(X_{n+1}, T_{n+1}) \sim P_X \cdot P_{T|X}$. Let $\hat{L}(X_{n+1}; \mathcal{D}^*_{cal})$ indicate the lower bound output by Algorithm A2, based on input calibration data $\mathcal{D}^*_{cal} = \{(X_i, \tilde{T}_i, C_i)\}_{i=1}^n$ with $\tilde{T}_i = T_i \wedge C_i$. Then, this lower bound satisfies:*

$$\mathbb{P}\left[T_{n+1} \geq \hat{L}(X_{n+1}; \mathcal{D}^*_{cal})\right]$$

$$\geq \left(1 - \frac{1}{n}\right)\left[1 - \alpha - \sup_{a \in [0,1]}\left(\mathbb{E}\left[\left|\frac{c_a(X_{n+1})}{\hat{c}_a(X_{n+1})\hat{\pi}_a} - 1\right|\right] + \sqrt{\frac{1 + \frac{\hat{\gamma}_a^2}{\hat{\pi}_a^2} + \max\left(1, \frac{\hat{\gamma}_a}{\hat{\pi}_a} - 1\right)^2}{n}\log(n)}\right)\right],$$

*where, for any $a \in [0, 1]$, we define*

$$\hat{\pi}_a = \mathbb{E}\left[\frac{c_a(X)}{\hat{c}_a(X)}\right].$$

*Proof of Theorem A5.* Under the same assumptions, Theorem 3 in Gui et al. (2024) says that for any $\delta \in (0, 1)$, with probability at least $1 - \delta$ over $\mathcal{D}^*_{\text{cal}}$,

$$\mathbb{P}\left[T_{n+1} \geq \hat{L}(X_{n+1}; \mathcal{D}^*_{\text{cal}}) \mid \mathcal{D}^*_{\text{cal}}\right]$$

$$\geq 1 - \alpha - \sup_{a \in [0,1]}\left(\mathbb{E}\left[\left|\frac{c_a(X_{n+1})}{\hat{c}_a(X_{n+1})\hat{\pi}_a} - 1\right|\right] + \sqrt{\frac{1 + \frac{\hat{\gamma}_a^2}{\hat{\pi}_a^2} + \max\left(1, \frac{\hat{\gamma}_a}{\hat{\pi}_a} - 1\right)^2}{n}\log\left(\frac{1}{\delta}\right)}\right).$$

Our result is then simply obtained by setting $\delta = 1/n$ and marginalizing over $\mathcal{D}^*_{\text{cal}}$. $\square$

**Theorem A6.** *Let $\{(X_i, T_i, C_i)\}_{i=1}^n$ be i.i.d. random samples from some distribution $P_{X,T,C} = P_X \cdot P_{T|X} \cdot P_{C|X}$, under Assumption 2.1. Let $f_{C|X}(c \mid x)$ be the probability density of $P_{C|X}$. Assume that $\hat{f}_a(x)$ is continuous in $a$ and that there exists some constant $\hat{\gamma}_a > 0$ such that $1/\hat{c}_a(x) \leq \hat{\gamma}_a$ for $P_X$-almost all $x$. Consider an independent random test sample $(X_{n+1}, T_{n+1}) \sim P_X \cdot P_{T|X}$. Let $\hat{L}(X_{n+1}; \tilde{\mathcal{D}}_{cal})$ indicate the lower bound output by Algorithm A5, based on input calibration data $\{(X_i, \tilde{T}_i, E_i)\}_{i=1}^n$ with $\tilde{T}_i = T_i \wedge C_i$ and $E_i = \mathbb{I}(T_i < C_i)$. Then, this lower bound satisfies:*

$$\mathbb{P}\left[T_{n+1} \geq \hat{L}(X_{n+1}; \tilde{\mathcal{D}}_{cal})\right]$$

$$\geq \left(1 - \frac{1}{n}\right) \left[1 - \alpha - \sup_{a \in [0,1]} \left(\mathbb{E}\left[\left|\frac{c_a(X_{n+1})}{\hat{c}_a(X_{n+1})\hat{\pi}_a} - 1\right|\right] + \sqrt{\frac{1 + \frac{\hat{\gamma}_a^2}{\hat{\pi}_a^2} + \max\left(1, \frac{\hat{\gamma}_a}{\hat{\pi}_a} - 1\right)^2}{n}\log(n)}\right)\right]$$

$$- \frac{n}{2} \cdot \mathbb{E}\left[\int_{\tilde{T}}^{\infty} \left|\frac{f_{C|X}(c \mid X)}{A(X, \tilde{T})} - \frac{\hat{f}_{C|X}(c \mid X)}{\hat{A}(X, \tilde{T})}\right| dc\right].$$

*Above, the expectation is taken with respect to a random sample $(X, \tilde{T})$, for $(X, T, C) \sim P_{X,T,C}$ and $\tilde{T} = T \wedge C$.*

Intuitively, Theorem A6 tells us that the finite-sample coverage achieved by our method depends on how well we can estimate the censoring distribution, $P_{C|X}$. In the special case where $f_{C|X}(c \mid x) = \hat{f}_{C|X}(c \mid x)$ for all $c, x$, which corresponds to statistically exact imputation, the finite-sample bound given by Theorem A6 becomes the same as the bound obtained by Gui et al. (2024) under type-I censoring, as reported in Theorem A5.

*Proof of Theorem A6.* The high-level idea of this proof is similar to that of Theorem A3: we connect the lower bound output by Algorithm A5 to the imaginary lower bound which would be obtained by applying Algorithm A2, the approach originally proposed by Gui et al. (2024), to an imaginary data set sampled from the same distribution but subject to type-I instead of right censoring.

Let $\mathcal{D}_{cal}$ denote the ideal calibration data set containing the true censoring times $C_i$ along with the corresponding values of $X_i$ and $\tilde{T}_i$; i.e.,

$$\mathcal{D}_{cal}^* = \left\{(X_i, \tilde{T}_i, C_i)\right\}_{i=1}^n.$$

Let $\hat{L}^*(X_{n+1}; \mathcal{D}_{cal}^*)$ denote the imaginary output that one would obtain by applying Algorithm A5 using $\mathcal{D}_{cal}^*$ instead of $\tilde{\mathcal{D}}_{cal}$ in the calibration phase. Equivalently, $\hat{L}^*(X_{n+1}; \mathcal{D}_{cal}^*)$ is the lower bound that would be produced under type-I censoring by Algorithm A2, the approach originally proposed by Gui et al. (2024). Then,

$$\mathbb{P}\left[T_{n+1} < \hat{L}(X_{n+1}; \tilde{\mathcal{D}}_{cal})\right] \leq \mathbb{P}\left[T_{n+1} < \hat{L}^*(X_{n+1}; \mathcal{D}_{cal}^*)\right]$$

$$+ \mathbb{P}\left[T_{n+1} < \hat{L}(X_{n+1}; \tilde{\mathcal{D}}_{cal})\right] - \mathbb{P}\left[T_{n+1} < \hat{L}^*(X_{n+1}; \mathcal{D}_{cal}^*)\right]$$

$$\leq \mathbb{P}\left[T_{n+1} < \hat{L}^*(X_{n+1}; \mathcal{D}_{cal}^*)\right] + d_{\mathrm{TV}}(P_{(X,\tilde{T},C)^n}, P_{(X,\tilde{T},C')^n}),$$

where $d_{\mathrm{TV}}(P_{(X,\tilde{T},C)^n}, P_{(X,\tilde{T},C')^n})$ denotes the total variation distance between the distributions of $\mathcal{D}_{cal}$ and $\mathcal{D}_{cal}^*$. Finally, the proof is completed by bounding $\mathbb{P}\left[T_{n+1} < \hat{L}^*(X_{n+1}; \mathcal{D}_{cal}^*)\right]$ with Theorem A5 and $d_{\mathrm{TV}}(P_{(X,\tilde{T},C)^n}, P_{(X,\tilde{T},C')^n})$ with Lemma A1.

$\square$

## Robustness when the survival model is accurate

**Theorem A7.** *Consider a random sample $(X, T)$ from some distribution $P_{X,T} = P_X \cdot P_{T|X}$. Let $\hat{L}(X; \tilde{\mathcal{D}}_{cal})$ indicate the corresponding survival lower bound output by Algorithm A5. For any $x \in \mathcal{X}$, let $\hat{q}_\alpha(x)$ denote the estimate of the $\alpha$ quantile of the conditional distribution of $T \mid X = x$ utilized in the final phase of Algorithm A4, with the function $\hat{q}_\alpha : \mathcal{X} \mapsto \mathbb{R}_+$*

*depending only on $\hat{\mathcal{M}}^{surv}$. Assume also that there exist a constant $b > 0$, and a function $q_\alpha : \mathcal{X} \mapsto \mathbb{R}_+$, such that, for any $\epsilon > 0$,*

$$\mathbb{P}\left[T \geq q_\alpha(x) + \epsilon \mid X = x\right] \geq 1 - \alpha - b\epsilon, \text{ almost surely with respect to } P_X.$$

*Then, for any constant $\ell > 0$,*

$$\mathbb{P}\left[T \geq \hat{L}(X; \tilde{\mathcal{D}}_{cal})\right] \geq 1 - \alpha - (b + 1) \left(\mathbb{E}\left[\mathcal{E}^\ell(X)\right]\right)^{1/(1+\ell)}. \tag{A12}$$

*and, for any $\beta \in (0, 1)$, with probability at least $1 - \beta$,*

$$\mathbb{P}\left[T \geq \hat{L}(X; \tilde{\mathcal{D}}_{cal}) \mid X\right] \geq 1 - \alpha - b\frac{\left(\mathbb{E}\left[\mathcal{E}^\ell(X)\right]\right)^{1/\ell}}{\beta^{1/\ell}}. \tag{A13}$$

*Proof of Theorem A7.* The proof of this result is an immediate consequence of Theorem A2, since by design Algorithm A5 leads to $\hat{L}(x; \tilde{\mathcal{D}}_{cal}) \leq \hat{q}_\alpha(x)$ almost surely for any $x \in \mathcal{X}$. $\qquad\square$

## A5.4.2. ASYMPTOTIC THEORY

*Proof of Theorem 3.6.* This proof follows the same strategy as that of Theorem 3.3, but relying on Theorems A6 and A7 instead of Theorems A3 and A4.

We begin by proving the unconditional result. For this, we consider two cases separately, relying on Theorems A6 and A7 respectively.

- Suppose Assumption 3.4 holds. In this case, the result follows immediately from Theorem A6, which tells us that, for any fixed $n$ and $N$,

$$\mathbb{P}\left[T_{n+1} \geq \hat{L}(X_{n+1}; \tilde{\mathcal{D}}_{cal})\right]$$

$$\geq \left(1 - \frac{1}{n}\right)\left[1 - \alpha - \sup_{a \in [0,1]}\left(\mathbb{E}\left[\left|\frac{c_a(X_{n+1})}{\hat{c}_a(X_{n+1})\hat{\pi}_a} - 1\right|\right] + \sqrt{\frac{1 + \frac{\hat{\gamma}_a^2}{\hat{\pi}_a^2} + \max\left(1, \frac{\hat{\gamma}_a}{\hat{\pi}_a} - 1\right)^2}{n}\log(n)}\right)\right].$$

  The proof is then completed by noting that the asymptotic limit of the right-hand-side term above, for $N, n \to \infty$ is $1 - \alpha$ under Assumption 3.4.

- Suppose Assumption 3.2 holds. In this case, we can obtain the desired result by applying Theorem A7 with $\ell = 1$, which tells us that

$$\mathbb{P}\left[T_{n+1} \geq \hat{L}(X; \tilde{\mathcal{D}}_{cal})\right] \geq 1 - \alpha - (b + 1)\left(\mathbb{E}\left[\mathcal{E}(X)\right]\right)^{1/2}.$$

  The proof is then completed by noting that the asymptotic limit of the right-hand-side term above, for $N \to \infty$ is $1 - \alpha$ under Assumption 3.2.

Let us now turn to proving the conditional result. Under Assumption 3.2, applying Theorem A7 with $\ell = 1$ tells us that, for fixed $N$ and any $\beta \in (0, 1)$,

$$\mathbb{P}\left[\mathbb{P}\left[T \geq \hat{L}(X; \tilde{\mathcal{D}}_{cal}) \mid X\right] \geq 1 - \alpha - b\frac{\mathbb{E}\left[\mathcal{E}(X)\right]}{\beta}\right] \geq 1 - \beta.$$

In particular, choosing $\beta = \sqrt{\mathbb{E}\left[\mathcal{E}(X)\right]}$ completes the proof, because

$$\mathbb{P}\left[\mathbb{P}\left[T \geq \hat{L}(X; \tilde{\mathcal{D}}_{cal}) \mid X\right] \geq 1 - \alpha - b\sqrt{\mathbb{E}\left[\mathcal{E}(X)\right]}\right] \geq 1 - \sqrt{\mathbb{E}\left[\mathcal{E}(X)\right]}.$$

$\qquad\square$

