# OpenReview forum: "Doubly Robust Conformalized Survival Analysis with Right-Censored Data"
_ICML.cc/2025/Conference — ICML 2025 spotlightposter_

### Official Review · Reviewer_MQ29 · 2025-02-26

**Overall Recommendation:** 4

**Summary:**

This work presents a novel conformal inference framework for survival analysis with right-censored data, motivated by the limitations of existing conformal methods for construction lower prediction bounds (LPBs) for general right-censored data. The core idea is to fit a censoring distribution and sample from the corresponding truncated distribution (“Decensoring”), changing the right-censored dataset into a semi-synthetic “decensored” dataset, where the existing methods apply. This paper proved in Theorem 3.3 and 3.6 that the marginal coverage of constructed LPBs on ideal assumption has asymptotic double robustness via a fixed or adaptive censoring-time filter technique. Experiments results have shown that this framework, leveraging an adaptive cutoff, yields the best LPB, compared to other calibration methods.

**Claims And Evidence:**

Mostly. See below.

**Essential References Not Discussed:**

An important contemporary work is founded here
https://openreview.net/forum?id=JQtuCumAFD
They introduced two methods: a “focused” method as dropout and “fused” method as a similar imputation method. According to their claim, their LPBs are finite-sample valid in section 1.3, also approximately valid in section 3.1.

**Experimental Designs Or Analyses:**

1.	Ablation of “decensoring” in model evaluations:
The novelty in methodology of this work mainly relies on the “decensoring” of calibration dataset added to the established fixed/adaptive cut-off strategies. The claim that “as long as $f{c|x}$ is reasonably accurate reasonably accurate, our two-step method is anticipated to yield approximately valid inferences” in section 2.3.3 is not backed with information about how accurate they are in experiments. Even if double robustness does not require perfect censoring distribution theoretically, the discussion on how the performance of pre-trained censoring distribution would impact the conformal LPB is meaningful in practice.

**Methods And Evaluation Criteria:**

Yes

**Other Comments Or Suggestions:**

1.	The authors only mentioned DeepSurv (Katzman et al., 2018) and Random Survival Forest (Ishwaran et al., 2008) in the introduction and they only evaluated Random Survival Forest in Section 4.1.
2.	The implementation of random survival forests is not consistent. “e.g., implemented via the R package ranger” in section 2.3.3, while it becomes “randomForestSRC” in section 4.1.
3.	Typo in line 181.
4.	Typo in the title of Algorithm A5.

**Other Strengths And Weaknesses:**

Strength: Detailed information with the major conformal methods in terms of pseudo-codes is nice.

Weaknesses：
If I understand Table A7 correctly, for SOTA models (random survival forests) in real-world observation datasets, DR-COSARC is not as competitive as Naive CQR.

The writing of this paper needs to be improved.
1.	Related work should be expanded and reference to censoring time imputation should be added, while repeated citations of Candes et al. (2023) and Gui et al. (2024) in Section 2 should be simplified. Use a preliminary section if necessary.

2.	While the computational cost of conformal prediction methods can be justified by the need for informed decision-making in fields like healthcare and finance, the potential computation cost in the imputation step and the conformal step should be mentioned as potential limitations of scalability in the Discussion.

**Questions For Authors:**

I have some practical questions:

1.	Is it fair to hold out 20% calibration data in the uncalibrated method? How are pretrained models obtained?

2.	As a main takeaway of this work, this framework is competitive in simulated cases where both pre-train models are badly fitted: How do we know they fit badly?

The censoring distribution modeling would flip the event indicator. How does the censoring rate in the training dataset affect the performance of pre-train models (and thus the proposed framework)?

**Relation To Broader Scientific Literature:**

“Decensoring” or, more broadly, modelling both censoring and time-to-event is not a new idea. The key contribution of this work is to establish an integrated theory and methodology with the established conformal survival analysis.

**Theoretical Claims:**

Math proofs seem valid but are not fully checked.

---

> ### Author Rebuttal · Authors · 2025-03-26
>
> Thank you for your thoughtful review and encouraging evaluation.
> ## Accuracy of the Censoring Model
> We appreciate your interest in how the censoring model’s accuracy affects performance. However, we believe there may be some misunderstanding about how best to assess this in practice.
> Rather than reporting standard imputation error metrics—which would be both difficult to interpret in our context and unobservable in real applications—we evaluate performance by varying key simulation parameters that directly affect model quality, such as training sample size (Figures A2, A3, A6, A7) and the number of irrelevant features (Figures A5, A9). These factors offer a more meaningful and observable way to assess the impact of censoring model accuracy.
> Collectively, these experiments support our central empirical claim: the method achieves valid inference as long as either the survival or censoring model is reasonably accurate. This double robustness is the core strength of our approach and is clearly demonstrated in our results.
> ## Discussion of Concurrent Work
> The paper by Davidov et al. (2025) is concurrent with ours. It appeared on OpenReview one week before the ICML submission deadline—after our submission—and prior to that, only an anonymous version was available, without author names. This made it both too recent and too ambiguous to cite appropriately. Moreover, even the anonymous version was first posted within four months of the ICML deadline, and we became aware of it only when our paper was already essentially finalized. While the two papers address related goals, the methods are conceptually very different. It would be interesting to see a detailed empirical comparison in future work.
> ## Empirical Performance Relative to Naive-CQR
> This seems to be a misunderstanding. Table A7 shows our method yields higher (more informative) LPBs than Naive-CQR on most datasets—except COLON, where performance is similar. This aligns with Figure 2 and expectations: Naive-CQR does not properly handle censoring and tends to be too conservative (although not always).
> ## Writing of Sections 1.3 and 2
> Thank you for the suggestion—we will expand the related work section in the camera-ready version. In Section 2, we intentionally highlight Candès et al. (2023) and Gui et al. (2024), as our method directly builds on and generalizes their work. While their names appear multiple times, each instance clarifies this connection. We believe the structure is appropriate but would welcome specific suggestions if further streamlining is needed.
> ## Computational Cost
> We agree this deserves mention. The imputation step adds negligible overhead—once the censoring model is fitted (as also required by existing methods), sampling is extremely fast. Compared to model fitting, the additional cost is minimal. We will clarify this in the revised version.
> ## Other Comments or Suggestions
> While Section 4.1 focuses on experiments using a generalized random forest (grf) for the survival model, we do evaluate other models in the appendix, as described in Section 4.3. We will clarify this point in the main text and fix the minor inconsistencies and typos you pointed out.
> ## Fairness of Holding Out Calibration Data for Uncalibrated Method
> This is a fair question. While the uncalibrated method could use more training data, we applied all calibration techniques to the same pre-trained model to ensure a clean comparison. Using different training sets would introduce some confounding.
> In any case, as shown in Figures A3 and A7, the uncalibrated method performs poorly in harder settings regardless of the training sample size, so using more data would not meaningfully improve its results. We’ll clarify this in the final version.
> ## Model Misfit and the Motivation for Conformal Inference
> In practice we wouldn’t know if the pre-trained survival model is accurate enough, and that’s exactly the motivation for conformal inference. Our goal is to provide robust, distribution-free guarantees even when the survival model is unreliable. Our framework is designed to achieve valid inference as long as either the survival or censoring model is reasonably accurate. The simulations were intentionally constructed to reflect this uncertainty and demonstrate our method’s double robustness.
> ## Varying the Censoring Proportion
> As the censoring rate increases, the censoring model becomes easier to estimate, while the survival model becomes harder to learn—highlighting the value of our method’s double robustness. Since only one model needs to be accurate, coverage should be maintained across a range of censoring levels.
> That said, extreme censoring can make our survival bounds less informative (i.e., more conservative), even if their coverage remains valid. This is a general challenge in survival analysis, not specific to our approach. We appreciate this question, and space permitting, will include this discussion and potentially an additional supporting figure in the final version.

---

### Official Review · Reviewer_5akg · 2025-03-13

**Overall Recommendation:** 4

**Summary:**

The paper proposes a doubly robust conformal inference method for constructing lower prediction bounds (LPBs) for survival times under right-censored data. By imputing unobserved censoring times using a machine learning model and calibrating survival models via weighted conformal inference, the method theoretically guarantees asymptotic validity if either the censoring or survival model is correctly specified. Extensive experiments on synthetic and real datasets demonstrate robustness in challenging scenarios where existing methods (e.g., Kaplan-Meier decensoring) underperform. The approach extends prior work on type-I censoring to handle more practical right-censoring settings.

## update after rebuttal
I updated the overall recommendation from 3 to 4 as my concerns have been adequately addressed.

**Claims And Evidence:**

Overall, the claims made in the submission can be supported by theoretical proofs or experiments.

**Essential References Not Discussed:**

N/A

**Experimental Designs Or Analyses:**

Synthetic experiments convincingly validate robustness, but lack exploration of high-dimensional covariates (p≫1000).
Real-data preprocessing (e.g., merging rare factor levels) is not rigorously justified and may introduce bias.

**Methods And Evaluation Criteria:**

Comprehensive evaluation across 10 synthetic settings and 7 real datasets.

**Other Comments Or Suggestions:**

N/A

**Other Strengths And Weaknesses:**

Strengths:
First method to extend doubly robust conformal inference to right-censored data.
Clear theoretical-empirical synergy.

Weaknesses:
Limited discussion of computational costs for imputation and calibration.
Real-data experiments lack domain-specific evaluation (e.g., clinical utility of LPBs).

**Questions For Authors:**

Why use split-conformal instead of full-conformal inference? Could cross-validation improve small-sample performance?

**Relation To Broader Scientific Literature:**

The work builds on conformal inference for survival analysis and connects to double robustness in causal inference.

**Theoretical Claims:**

Theorems 3.3 and 3.6 are well-constructed, leveraging double robustness principles from causal inference. However, the proofs assume asymptotic consistency of model estimates, which may not hold with finite training data.

---

> ### Author Rebuttal · Authors · 2025-03-26
>
> Thank you for your thoughtful and generally positive review. We appreciate your recognition of the novelty and strength of our contributions and are happy to respond to your comments below.
> ## Asymptotic vs Finite-Sample Theoretical Results
> You're right that Theorems 3.3 and 3.6 provide asymptotic double-robustness guarantees. However, as shown in Theorems A3 and A5 (Appendix), we also establish finite-sample coverage results for both the fixed- and adaptive-cutoff versions of our method. While these bounds are somewhat loose and not directly useful for practitioners, they rely on weaker assumptions and are key to establishing asymptotic validity, offering some theoretical support for our method’s strong empirical performance.
> The challenge of deriving tighter finite-sample guarantees reflects the inherent complexity of conformal inference under censoring—a challenge also noted in Candès et al. (2023) and Gui et al. (2024), even in the simpler type-I setting. In short, censoring makes theoretical analysis harder.
> Importantly, our experiments show that the adaptive version of our method consistently outperforms the fixed-cutoff version, despite requiring somewhat stronger assumptions for double robustness. This illustrates a broader pattern in modern data science: practical performance sometimes exceeds what theory can rigorously explain. We’ll clarify this point and make the finite-sample results more visible in the revised version.
> ## Experiments with High-Dimensional Data
> Our synthetic experiments are designed to evaluate the performance of our conformal inference method under the most relevant practical challenges that may affect the quality of the survival and censoring models—including model misspecification, limited sample sizes, and increasing numbers of covariates. These challenges commonly arise in high-dimensional settings, but in our context they do not require using extremely large feature spaces in simulation.
> Figures A5 and A9 (referenced in Section A.3) vary the number of irrelevant covariates used to train the censoring model and show that performance degrades as more noise features are added—due to overfitting. This is consistent with double robustness and precisely represents the kind of failure mode that our framework is designed to mitigate. While a sparse learner could further improve performance in such settings (since the true censoring model is sparse), our focus is on imputation and conformal inference, not sparse modeling. Our method is designed to be broadly applicable and can be used with any underlying model, including sparse ones. We will highlight these experiments more clearly in the revised version.
> ## Data Pre-Processing
> The preprocessing steps in Appendix A4.1—such as merging rare categorical levels and removing extreme outliers—are standard procedures to ensure model compatibility and stability. We are not aware of any way these steps would introduce “bias” in our experiments. That said, we will clarify and briefly justify them in the final version to avoid confusion.
> ## Computational Cost
> Thanks for this suggestion. The imputation step is a key conceptual contribution of our method, but computationally it is very light. Once the censoring model is trained, sampling requires evaluating a one-dimensional integral (Eq. 4), which is either analytical or quickly approximated. This step is not a bottleneck. As with all comparable methods, the main cost lies in training the survival and censoring models. We’ll clarify this in the revised paper.
> ## Practical Relevance
> We agree that domain-specific evaluation—such as assessing the clinical value of lower predictive bounds—is important future work. Our goal in this paper was to develop and validate the core methodology, which is why we focused on benchmark evaluations.
> That said, we believe our survival LPBs have the potential to be useful for practitioners. For example, in healthcare they may inform treatment prioritization based on survival likelihood under resource constraints (as noted by Candès et al., 2023). We will expand this motivation in our introduction to make the practical relevance more self-contained.
> ## Split vs. Full Conformal Inference, and Possible Extensions using Cross-Validation
> We chose split conformal inference for its simplicity, efficiency, and ability to support double-robustness guarantees under relatively mild assumptions. These same considerations motivated prior work on conformal survival analysis under type-I censoring (Candès et al., 2023; Gui et al., 2024).
> Extending our method to full conformal inference, cross-validation, or jackknife+ could potentially improve small-sample performance, but each would involve substantial computational and theoretical challenges—especially due to censoring. We agree these are potentially interesting directions for future work and will mention them in the revised discussion. Thank you for the suggestion!

---

### Official Review · Reviewer_Udgo · 2025-03-13

**Overall Recommendation:** 4

**Summary:**

This paper studies conformal inference for right-censored data, in which it generalizes prior works beyond the type-I censoring setting (i.e., when all the censoring times are observed). Its main idea is to impute the  censoring times by sampling from the estimated censoring mechanism, and obtain a "synthetic" data set whose censoring times are fully observed, on which prior methods can be applied to
produce calibrated lower prediction bounds.

It is shown that the proposed method enjoys a double-robustness property, achieving asymptotic coverage guarantees as long as either the distribution of $T|X$ or $C|X$ is well estimated. The proposed method is evaluated in extensive simulation and real-data studies, in comparison with existing methods.

**Claims And Evidence:**

The main claim of the paper is that it provides calibrated lower prediction bounds for right-censored data, generalizing prior works beyond the type-I censoring setting. The method is said to be asymptotically valid if either the censoring mechanism or the survival distribution can be well estimated. The claim is supported by theoretical results (Theorems 3.3 and 3.6), as well as simulation results.

**Essential References Not Discussed:**

NA.

**Experimental Designs Or Analyses:**

The experimental design follows from settings in existing works; it also provides comprehensive comparison and sensitivity analysis.

**Methods And Evaluation Criteria:**

The proposed method makes it possible to construct LPB when the censoring times are not fully observed. This is achieved by imputing the missing censoring times by sampling from the estimated distribution of $C|X$. I wonder, however, how stable this algorithm is,
considering that the missing censoring times are imputed with random variables.

**Other Comments Or Suggestions:**

Page 4, line 181, $\hat C i$ -> $\hat C_i$

**Other Strengths And Weaknesses:**

NA.

**Questions For Authors:**

1. Since the missing censoring times are imputed by sampling from the estimated $C|X$, the results depend on the realization of the random variables. I wonder how stable this algorithm is to different realizations of censoring times.
2. (If I understand correctly) The proposed method (fixed version) does have finite-sample guarantees when $C|X$ is known, which is perhaps worth emphasizing.

**Relation To Broader Scientific Literature:**

The main claim of the paper is that it provides calibrated lower prediction bounds for right-censored data, generalizing prior works beyond the type-I censoring setting. This greatly improves the applicability of the framework.

**Theoretical Claims:**

I went over the proof outline, and did not spot any significant issue.

---

> ### Author Rebuttal · Authors · 2025-03-25
>
> Thank you for your thoughtful review and for your positive evaluation of our paper. We appreciate your careful reading, and we are happy to answer your two very insightful questions.
>
> ## Imputation randomness
>
> *"Since the missing censoring times are imputed by sampling from the estimated $C \mid X$, the results depend on the realization of the random variables. I wonder how stable this algorithm is to different realizations of censoring times."*
>
> This is an excellent question. Conceptually, we believe statisticians should not view randomness as inherently problematic. Imputing missing censoring times by sampling from their estimated conditional distribution given observed data is a natural and principled strategy. Even if the censoring times were observed directly, they would still be treated as realizations from a random process—our method simply mirrors that generative view using model-based imputation. This allows us to reduce a more complex model-free problem (survival analysis under general right censoring) to a more tractable one: survival analysis under type-I censoring, for which conformal methods are more easily applicable.
>
> While data randomness is fundamental to statistical inference in general, many conformal inference methods in particular introduce additional randomness into the observed data—most notably through random sample splitting between training and calibration sets. The random imputation step in our method is thus fully consistent with the broader conformal inference framework and, in fact, this randomness plays a key role in enabling formal coverage guarantees.
>
> That said, your point about stability is well taken. To address it concretely, we will include a figure in the camera-ready version of the paper that empirically assesses the variability of the results obtained by applying our method to the same observed data set across different imputations of the latent censoring times. These experiments will show that, as long as the calibration sample size is at least moderately large (on the order of a few hundred observations), the sampling variability introduced by our method is minimal and does not meaningfully impact the informativeness of our predictive inferences. Moreover, as expected, this variability decreases as the calibration sample size increases.
>
> Prompted by your suggestion, we will also mention in the discussion that one could explore possible extensions of our method that further reduce this variability by leveraging recent developments in e-value-based methods. We expect that such extensions may reduce variance at the cost of increased conservativeness—an interesting tradeoff that we agree is worth flagging as a possible direction for future work.
>
> ## Finite-sample theory
>
> *“(If I understand correctly) The proposed method (fixed version) does have finite-sample guarantees when $C \mid X$ is known, which is perhaps worth emphasizing.”*
>
> Yes, you are absolutely correct. Indeed, Theorem A3 in Appendix A5.3 provides a finite-sample guarantee for our method under the fixed-cutoff implementation, which becomes tighter if the censoring mechanism is known. We will highlight this more clearly in the main text.
>
> Additionally, we have also established finite-sample validity for the adaptive-cutoff implementation in Theorem A5 (Appendix A5.4). These results are somewhat more conservative—not because the method is less effective in practice, but because the theoretical analysis is more technically involved. This is similar to what Gui et al. (2024) encountered in their analysis under type-I censoring.
>
> We believe this distinction is worth clarifying in the camera-ready version: while the adaptive version of our method performs better empirically (as our results show), it is more challenging to analyze theoretically. This situation is not uncommon in modern statistical methodology, where practical performance can sometimes outpace what current theory can tightly capture. We’ll make sure to emphasize that the theory remains sound in both cases and the gap lies primarily in the sharpness of the finite-sample bounds, not in the validity or robustness of the approach itself.
>
> Thank you again for your helpful comments. We believe your suggestions will help improve the clarity and impact of the final version.

---

### Official Review · Reviewer_qnsR · 2025-03-13

**Overall Recommendation:** 4

**Summary:**

The paper addresses the problem of constructing lower prediction bound (LPB) for survival time under conditional independent right censoring. In particular, the paper extends the approaches in Candes et al. (2023) and Gui et al. (2024) for type I censored data, which assumes that the right censoring time is always observed, to the censoring scenario that are more commonly encountered in practice where the right censoring time is only observed for censored subjects. The proposed approaches first impute the missing censoring times from the estimated conditional censoring distribution, and then apply the methods in Candes et al. (2023) and Gui et al. (2024) for type I censored data. With an additional adjustment, the proposed LPBs enjoy a doubly robust (DR) property: the obtained LPB has asymptotically valid marginal coverage when either (i) the conditional censoring distribution is estimated sufficently well, or (ii) the conditional quantile of event time is consistently estimated and the true conditional event time distribution satisfies a smoothness condition. In addition, when (ii) is true, the proposed LPB also also has valid approximate conditional coverage. Simulation studies are done to compare the performance of the proposed LPBs with existing apoproaches. The results show that the proposed LPBs is robust in challenging settings. The proposed approaches are applied to construct LPB for survival times in seven publicly available datasets.

## update after rebuttal
I updated the overall recommendation from 2 to 4. Previously I choose 2 because there are errors in a math expression and the algorithm in the main text missing the important DR adjustment step which caused confusion. The authrors clarified that the error in the math expression is only a typo and they will revise the algorithms and writing of the paper. These resolves the above concerns. As for the assumption in the theory being a strong assumption, I agree that it is okay to keep it as it is for the ICML paper, without making the paper more technical. Also the authors mentioned that they improve on addressing the simulation results in the main text.

**Claims And Evidence:**

Overall the claims made in the submission is supported by clear and convincing evidence, but I found their statement of the algorithms and the claims for DR results potentially confusing and misleading.

Their algorithm first impute the missing censoring times and then applied the existing approaches for handling type-I censored data, and they showed in Proposition 2.2 that if the true conditional censoring distribution is known, the data with the imputed censoring times follows the same distribution as the hypothetical type-I censored data if all the censoring times were observed. This justifies the use of the imputed censoring times and the application of the existing approaches for type-I censored data. While reading, that makes me think that the proposed approach will rely on the conditional cenosring distribution to be estimated resonably well, so it is kind of surprising that their approach enjoys the DR property. It turns out that their algorithms also has a DR adjustment step, which takes the minimum of (i) the LPB from the existing approaches under type-I censoring and (ii) the estimated quantile function of the conditional event time distribution. Since (ii) has approximately valid marginal coverage if the quantile function of the conditional event time distribution is estimated sufficently well, by definition, the adjusted LPB has the DR property claimed in the paper. However, this key step of adjustment is barely mentioned in the main text and it is burried in Algorithm A4 and A5 in the appendix (except mentioned in algorithm 3 Step 4 and 5, but it replicates the DR adjustment step of Algorithm A5 in Step 3, and it is not higlighted to be an adjustment step for DR). So I think their current writing is potentially confusing. More discussion about this key step and the consequence and potential limitations of taking the minimum of two LPB is needed for clarity.

**Essential References Not Discussed:**

I’m not aware.

**Experimental Designs Or Analyses:**

I read the simulation in the main text and briefly browsed the additional results in the appendix.

The simulation studies is done under various situations, but few discussions are made on how the results relate to their theoretical results of double robustness. I think the paper would benefit from discussing under which scenarios, the proposed methods are (1) expected to have good estimate for the censoring model and/or event time model, (2) expected to show good performance according to their theory, and (3) expected to outperform the comparison methods, etc. For example, their paragraph on “Leveraging Prior knowledge on P_{C|X}” makes me think that this probably relate to how well the conditional censoring distribution can be estimated under these models.

I’m also curious about how the performance of the proposed approaches compares with their counterparts without the DR adjustment step.

In simulation results, the lower bound for the proposed LPB is lower than the naïve CQR, which is not surprising as the DR adjustment step in the proposed algorithms takes the minimum of the two LPB where one of them is the naïve CQR. I suggest commenting on this result as it relate to the potential limitation of the proposed approach.

For application, the event time distribution is estimated under different models but the censoring distribution is always estimated using grf. Why only one censoring model is considered instead of multiple ones like for the event time distribution?

The discussion of the application results can be more informative. For example, in line 414 (left), “the comparatively simpler nature of survival analysis with these datasets”. Do you mean the true distribution of the T|X or C|X follow simplier models, or there are fewer covariates, or the model can estimate the conditional distributions T|X, C|X well so that the error rate condition required in your Assumptions are more likely to hold?

For the data analysis results in the appendix (Table A4, etc.), what is the unit of the LPB. I saw the numbers varied a lot across datasets. Does it mean there is more uncertainty in some than the other, or they are due to different units of the time-to-event in the datasets? How can we tell if they are informative?

**Methods And Evaluation Criteria:**

Yes, the proposed methods and/or evaluation criteria (e.g., benchmark datasets) make sense for the problem or application at hand.

**Other Comments Or Suggestions:**

In line 181, Ci, i should be subscript.

**Other Strengths And Weaknesses:**

Strength:

Originality for addressing the problem of conformal inference for right censored data under a more realistic censoring setting than the type-I censoring that has been studied in the literature.

Comprehensive simulations and real data analysis.

Weakness:

The writing and flow of the paper can be improved. For example, perhaps Section 2.2 can be folded in 1.3.  The last paragraph in the right column in line 157 may be simplified by saying that (3) is the conditional density of C|X,\tilde T, E=1.

The clarity of the writing may be improved if the authors use detailed and accurate wording. For example, what does “robust” and “delicated” mean in “tend to be more robust in more delicated scenarios” (line 088 left).

**Questions For Authors:**

1.	How the two proposed algorithms and the writing of the paper can be revised to highlight the key adjustment step for DR property and acknowledge the limitation that this adjustment step may cause?

2.	What is the cost of achieving the DR property with the adjustment step? Are there any justification that with this adjustment step the LPB is still informative enough? Does it worth to achieve DR at the cost of this adjustment step?

3.	Can the second condition of Assumptions 3.1 and 3.4 can be relaxed?

4.	How can the simulation studies be better designed or how can the current simulation results to be better intepreted to show the advantage of the DR property of the proposed approaches?

I think addressing the above questions will improve the clarity of the paper and make the paper much stronger.

**Relation To Broader Scientific Literature:**

The paper contribute to the conformal inference literature by extending the existing approach for constructing LPB for time-to-event data under type-I censoring to the more general type of right censoring encountered in practice where the censoring times are not always obseved. The paper extend both the algorithms with fixed cutoffs and with adaptive cutoff for type-I censored data.

**Theoretical Claims:**

I checked their theoretical results in Proposition 2.2, Theorem 3.3 and Theorem 3.6 and the corresponding assumptions, as well as some of the proofs in the appendix.  Besides the concerns about their statement of the algorithms and the DR results mentioned in the section of “Claims and Evidence”, I’m also concerned with the second condition in their Assumptions 3.1 and Assumotion 3.4, which incolves the sample size n for the calibration set.

The statement is about the limit of a quantity related to the estimation error of the conditional censoring distribution converges to zero faster than 1/n rate, as the sample sizes n and N goes to infinity. I guess N denotes the sample size of the data used to train the censoring model, but the definition of N seems to be not mentioned in the paper.

My concern for this assumption is how strict it is. In practice, the integral term in their assumption that involves the estimation error of the conditional censoring distribution usually converges at no faster than N^{-1/2} rate, which is the parametric rate of convergence. So it seems that this condition requires that n is of order at least N^2, which is pretty large. The choice in the simulation section with sample size 1000 for each set seems to not follow this requirement.I think more discussion and examples of when this condition can be satisfies will enhance the theory part.

Also more discussion about how the Assumptions 3.1, 3.2, 3.4, 3.5 compare with the assumptions in the literature will make the theory part stronger.

There is an error in equation (4). The limits of the integral should be from \tilde T_i to infinity?

Discussion from line 180 – 184 (left) is inaccurate. The synthetic sample shares the same distribution as the ideal sample only if \hat f_{C|X} is exactly equal the truth.

---

> ### Author Rebuttal · Authors · 2025-03-26
>
> Thank you for your detailed and thoughtful review. We find your feedback very helpful. While some presentation issues may have caused confusion, they are easily resolved and do not reflect inherent flaws. We respond below and will incorporate your suggestions into the revised paper.
> ## DR Adjustment
> You're right that the DR adjustment is essential for establishing double robustness. This is discussed in Section 2 and shown in line 5 of Algorithm 3. The confusion arose because it is also part of the fixed-cutoff method but was accidentally omitted from Algorithm 2 (though present in Algorithm A4). We will fix this omission and also correct the outdated sentence comparing Algorithms 2 and 3. Empirically, the DR adjustment typically has a small effect, as the preliminary LPB very often is already lower than the uncalibrated quantile. Based on your comment, we will include in the appendix an additional figure showing this explicitly.
> ## Convergence Rate of Censoring Model
> We agree the convergence rate assumed for the censoring model in the theoretical analysis is quite strong, but we highlight this directly after Assumption 3.1, noting it suggests more data should be allocated to training than calibration. We view this as a reasonable and useful recommendation.
> That said, it may be possible to relax the converge rate under stronger assumptions or with more technical effort. However, we do not see a compelling need nor an obvious way of doing so without making the paper far more technical than is appropriate for ICML.
> Our main goal is to establish the soundness of the method and highlight its double robustness—a key property not shared by existing alternatives. Whether the convergence rate assumption can be weakened seems secondary. In practice, what matters is whether the method performs well, and our results show that it does. (After all, it is unclear what it would even mean for a convergence rate to “hold” for a single finite dataset.)
> We would of course welcome future work that seeks to tighten or relax this theoretical condition.
> ## Censoring Time Imputation
> Thank you—you are right about the typographical error in Equation (4); we will correct the integration limits. We will also revise the vague phrase “accurately estimates” to clarify that the synthetic sample matches the target distribution only if $\hat{f}\_{C \mid X} = f\_{C \mid X}$.
> ## Comparison with Naive CQR
> There appears to be a misunderstanding. As stated on page 6, Naive CQR calibrates CQR to predict $\tilde{T} = T \land C$ instead of $T$, which leads to overly conservative bounds. This is very different from our DR correction, which uses the survival model’s uncalibrated quantile. We do not take the minimum with the Naive CQR bound, and doing so would be unjustifiably conservative. Indeed, our method produces more informative bounds than Naive CQR, as shown in Figures 1 and 2.
> ## Design of Empirical Experiments
> We conducted a thorough empirical evaluation, much of it in Appendix A3–A4. We will better reference these results in the main text and will add a new figure to further highlight double robustness.
> Our experiments explore the performance of our method in the face of key challenges like model misspecification, limited training data, and covariate dimensionality. For example:
> - Figures A2, A6: Vary censoring model training size; performance improves as quality increases.
> - Figures A5, A9: Add irrelevant covariates in censoring model; overfitting degrades performance.
> - Figures A3, A7: Vary both models’ training size; highlight double robustness.
>
> Figure A2 is particularly illustrative: with a poor survival model, performance improves rapidly as the censoring model improves. We will complement this with a similar figure where the survival model is more accurate and our method remains robust even if the censoring model is poor.
> ## Different Censoring Models
> While the main text uses GRF for simplicity, Appendix A3.2 includes additional experiments with other censoring models (see p.18). We limited real-data results to GRF to avoid overwhelming the appendix, which already contains 16 supplementary figures and 7 tables. We believe these experiments sufficiently support our claims.
> ## Survival Modeling in Benchmark Data Sets
> We will clarify that our comment about “simpler” datasets refers to the fact that the uncalibrated survival model performs reasonably well, suggesting $T|X$ is easier to estimate in these benchmarks. However, our synthetic experiments show that when survival modeling is hard, alternative methods fail—whereas ours maintains valid coverage as long as either model is accurate. If the survival model is perfect, conformal inference may not be needed—but in practice, we don’t know if it is, and our method offers stronger protection than alternative approaches.
> ## LPB Units
> LPBs are in different units across datasets and should not be directly compared across them.

---

### Official Review · Reviewer_FTCz · 2025-03-16

**Overall Recommendation:** 4

**Summary:**

This paper proposes a new conformal inference approach specifically for right-censored data, aimed at constructing lower prediction bounds (LPBs) for survival times. The method is theoretically asymptotically doubly robust and demonstrates strong empirical results, offering more informative and reliable LPBs compared to existing alternatives.

**Claims And Evidence:**

The paper proposal of doubly robust conformal inference method for survival analysis is supported by their theory and strong empirical results.

**Essential References Not Discussed:**

N/A

**Experimental Designs Or Analyses:**

While the experiments in Fig. 1 demonstrate that the proposed method performs well, its success is attributed to the framework’s ability to model the censoring distribution accurately. However, this aspect is not clearly demonstrated or fully substantiated.

**Methods And Evaluation Criteria:**

The paper presents DR-COSARC (Doubly Robust Conformalized Survival Analysis under Right Censoring), which builds on recent work in conformalized survival analysis by focusing on more practical right-censored cases. It further incorporates doubly robust properties, making the proposed framework more reliable. DR-COSARC is designed with both fixed and dynamic cutoffs, enhancing its generalizability. The method is evaluated on both simulated and real-world datasets, and compared against several existing frameworks.

**Other Comments Or Suggestions:**

N/A

**Other Strengths And Weaknesses:**

N/A

**Questions For Authors:**

N/A

**Relation To Broader Scientific Literature:**

The contributions of the paper is related to the broader scientific literature for conformal inference.

**Theoretical Claims:**

The paper theoretically proves (under the provided assumption) that the proposed DR-COSARC is doubly robust for both constant cutoffs (Theorem 3.3) and Adaptive custoffs (Theorem 3.6).

---

> ### Author Rebuttal · Authors · 2025-03-25
>
> Thank you for your review and your positive evaluation of our paper. We’re grateful for the opportunity to clarify some aspects of our empirical analysis, particularly since many important results are presented in the appendix and may have been unintentionally overlooked.
>
> The appendix contains an extensive and carefully structured set of numerical experiments specifically designed to study how the performance of our method depends on the accuracy of the survival and censoring models. These results provide strong empirical support for the theoretical double robustness of our method—namely, that valid and informative predictive bounds can be obtained when either distribution is modeled accurately. We believe that a more detailed look at these experiments will confirm that our claims are fully substantiated by the data.
>
> The main paper (Figure 1 and the discussion in Section 4) focuses on settings that vary the difficulty of estimating the survival model. These experiments show that our method approaches oracle performance when the survival distribution is easier to learn, while still outperforming existing alternatives in more challenging settings. These results are complemented by a broad set of additional experiments detailed in the appendix and only briefly summarized in Section 4 due to space constraints.
>
> For example:
> - Figures A2 and A6 vary the number of training samples used to fit the censoring model, showing that coverage improves as the censoring model becomes more accurate, especially in difficult settings where the survival model may be unreliable.
> - Figures A5 and A9 vary the number of irrelevant covariates used in the censoring model, demonstrating the impact of overfitting on performance.
> - Figures A3 and A7 examine the joint effect of training sample size on both models, further illustrating our method’s double robustness.
>
> These and many other results presented in Appendices A3 and A4 offer what we believe is a thorough empirical validation of our approach. While Section 4.3 of the main paper already points readers to these results, we now realize that some of this material may not have been sufficiently highlighted, and that even attentive readers could miss it.
>
> We will take full advantage of the extra page allowed in the camera-ready version to address this. In particular, we plan to improve how these experiments are referenced and summarized in the main text, and will likely move an important figure (such as A2 or A3) into the main body to make the role of the censoring model even more visible. This will ensure that all readers, regardless of how closely they examine the appendix, clearly see the breadth and depth of the empirical results supporting this paper. Thank you again for this helpful feedback!

---

### Decision · Program_Chairs · 2025-05-01

**Decision:**

Accept (spotlight poster)

**Comment:**

After a very fruitful discussion between the authors and the reviewers, the reviewers unanimously favored accepting this paper (all five of the reviewers gave a score of a 4). I am thus also recommending this paper for acceptance.